# Defining the condensate landscape of fusion oncoproteins

Swarnendu Tripathi[1,20], Hazheen K. Shirnekhi [1,20], Scott D. Gorman [1,16,20], Bappaditya Chandra[1], David W. Baggett[1], Cheon-Gil Park[1], Ramiz Somjee [1,2,17], Benjamin Lang [1,3], Seyed Mohammad Hadi Hosseini[1,3], Brittany J. Pioso [1], Yongsheng Li [4], Ilaria Iacobucci [5], Qingsong Gao[5], Michael N. Edmonson [6], Stephen V. Rice[6], Xin Zhou[6], John Bollinger[1], Diana M. Mitrea[1,18], Michael R. White[1,19], Daniel J. McGrail [7,8], Daniel F. Jarosz [9,10], S. Stephen Yi [4,11,21], M. Madan Babu [1,3,21], Charles G. Mulligan [5,21], Jinghui Zhang [6,21], Nidhi Sahni [12,13,14,21] & Richard W. Kriwacki [1,15,21] ✉

Fusion oncoproteins (FOs) arise from chromosomal translocations in ~17% of cancers and are often oncogenic drivers. Although some FOs can promote oncogenesis by undergoing liquid-liquid phase separation (LLPS) to form aberrant biomolecular condensates, the generality of this phenomenon is unknown. We explored this question by testing 166 FOs in HeLa cells and found that 58% formed condensates. The condensate-forming FOs displayed physicochemical features distinct from those of condensate-negative FOs and segregated into distinct feature-based groups that aligned with their sub-cellular localization and biological function. Using Machine Learning, we developed a predictor of FO condensation behavior, and discovered that 67% of ~3000 additional FOs likely form condensates, with 35% of those predicted to function by altering gene expression. 47% of the predicted condensate-negative FOs were associated with cell signaling functions, suggesting a functional dichotomy between condensate-positive and -negative FOs. Our Datasets and reagents are rich resources to interrogate FO condensation in the future.

Fusion oncoproteins (FOs) arise from chromosomal translocations and are observed in ~17% of all cancers[1]. FOs commonly drive cell transformation and oncogenesis, and are prevalent in pediatric cancers, especially those with poor clinical prognosis[2]. The advent of cancer genome and transcriptome sequence databases has led to the identification of thousands of FOs across diverse cancer types[1,3,4], increasing the need to understand the molecular mechanisms underlying their roles in cancer. Most FOs fall into one of two general classes, determined by the parent genes involved in translocation events. The first class incorporates enzymatic kinase domains that become misregulated due to elimination of allosteric regulatory regions and/or sequence motifs that govern degradation in the fusion proteins[5]. The

second encodes aberrant transcription factors that commonly join an intrinsically disordered region (IDR) of one parent to a folded chromatin- or DNA-binding domain of the other; FOs in this class recruit components of the transcriptional and/or chromatin remodeling machinery and drive aberrant gene expression[6–13]. While the molecular mechanisms underlying oncogenesis by several FOs are understood in some detail[14,15], this knowledge is lacking for hundreds of other FOs with known cancer associations.

In the first class, two FOs, EML4-ALK and CCDC6-RET, incorporate enzymatic domains from receptor tyrosine kinases and promote oncogenic Ras signaling by undergoing liquid–liquid phase separation (LLPS) to form cytoplasmic condensates that exhibit elevated kinase

---

activity[16–18]. These FOs display folded domains that mediate multivalent interactions, promoting formation of round condensates, or puncta, with heterogeneous material properties in cells. In the second class, several FOs that function as aberrant transcription factors, such as EWS-FLI1[7], FUS-CHOP[10,11], and several NUP98 FOs[6,8,12,19], undergo LLPS to form aberrant transcriptional condensates that mis-regulate gene expression. These condensates appear as round puncta in cells and the FOs that form them display IDRs that promote multivalent interactions. These observations implicate LLPS in the oncogenic mechanisms of both major classes of FOs. However, because only a few FOs have been studied through the lens of LLPS, it is not currently known whether condensate formation is a general property of FOs.

Here we address this knowledge gap by asking whether formation of aberrant cellular condensates is a common property of the thousands of FOs associated with diverse human cancers. To investigate this question, we assembled a database of the amino acid sequences of several thousand FOs and tested 166 of them for condensate formation in cells. We note that while the observation of FO-induced condensates is suggestive of formation through phase separation, other tests beyond the scope of the current investigations would be needed to establish this assembly mechanism. Therefore, our use of the term "condensates" is agnostic as to formation mechanism. Using fluorescence imaging, we found that 58% of 166 fluorescently labeled FOs form round condensates (puncta) in cells. To understand the factors that drive condensate formation, we computed amino acid sequence-derived physicochemical features for all tested FOs and identified distinct patterns of features associated with condensate-positive and negative FOs. Remarkably, for many FOs, feature patterns also correlated with their sub-cellular localization and function, including regulation of gene expression for nuclear FOs and regulation of cell signaling for cytoplasmic FOs, respectively. Surprisingly, most of the cytoplasmic FOs with cell signaling functional terms did not form condensates. Using Machine Learning, we leveraged physicochemical features to predict condensate formation by 2999 additional, untested FOs associated with diverse cancers. Amongst these, 1999 (67%) were predicted to form condensates in cells, with 35% of those further predicted to localize within the nucleus and regulate gene expression. 1000 (33%) FOs were predicted to not form condensates, with 47% of those predicted to localize within the cytoplasm and regulate cell signaling. The databases, imaging datasets, computational tools, and cellular expression reagents, we have generated for FOs will serve as a resource for testing the roles and molecular mechanisms of condensate formation by FOs in cancer biology in the future.

## Results

### FOs are significantly enriched in physicochemical features associated with phase separation

We assembled a database of 4540 FO protein sequences derived from a combination of patient genomic and transcriptomic data, termed the "FOdb", from sources at St. Jude Children's Research Hospital (SJCRH) and from The Cancer Genome Atlas (TCGA) (Fig. 1a, Supplementary Dataset 1; see "Methods"). For a subset of 3174 FOs, we could obtain verified data on cancer type and number of patient occurrences (termed "FOdb-II"; Supplementary Dataset 2). Those FOs were derived from diverse cancers, ranging from B-cell acute lymphoblastic leukemia to solid tumors such as breast invasive carcinoma, osteosarcoma, prostate adenocarcinoma, lung carcinomas, and others (Supplementary Dataset 3, Fig. 1b). Many FOs were observed in multiple patients, although the majority of FOs were seen in a single patient (Fig. 1c).

To assess the potential for condensate formation by FOs in FOdb-II, we analyzed their enrichment (or depletion) in a few physicochemical features associated with phase separation[20,21]. In comparison to the human proteome, the amino acid sequences in the FOdb-II were significantly enriched (two-sided $t$-test, $p \leq 10^{-4}$) in predicted intrinsic protein disorder (quantified as the fraction of residues in predicted disordered regions); potential for pi-pi and pi-cation interactions (quantified as the PScore value[22]), and for prion-like domains (quantified as the PLAAC NLLR prion propensity score[23]); and depleted in hydrophobic residues (quantified as the fraction of hydrophobic residues using CIDER[24]; Fig. 1d). Further, we found that the parent proteins of FOs in FOdb were significantly more likely to undergo LLPS than proteins across the human proteome (odds ratio, 2.06 and $p$-value, 1.8e−23; Fig. 1e). Together, these observations showed that FOs are enriched in features associated with phase separation and led us to hypothesize that many form condensates in cells.

### Many mEGFP-tagged FOs form condensates in HeLa cells

To test our hypothesis, we expressed monomeric, enhanced green fluorescent protein (mEGFP)-tagged forms of 166 FOs (termed the Expressed FOs, Supplementary Dataset 4; we note that Supplementary Dataset 3 provides a glossary describing the various groups of FOs presented in this work) in live HeLa cells and used confocal fluorescence microscopy to evaluate them for condensate formation (Fig. 2a). The 166 FOs spanned both adult and pediatric cancers and included 77 of the 110 FOs with a patient count of ≥3 and 36 FOs previously demonstrated to drive oncogenic phenotypes in cancer-relevant cell types (Supplementary Dataset 4).

Live HeLa cells were transfected with mEGFP-tagged FOs under identical conditions (e.g., protocol, DNA concentration, number of cells) and imaged 24 h post transfection to test for condensate formation. Due to the nature of transient transfections, the mEGFP-tagged FOs were expressed at variable levels, both for a given FO within a cell population and between different FOs, and we recorded fluorescence microscopy images of ≥50 cells for each FO to broadly sample their expression profiles. We observed that 96 (58%) of the Expressed FOs formed round condensates in HeLa cells, and those were termed puncta(+) FOs (see "Methods" for criteria for scoring an FO as puncta(+); Fig. 2b, left panel; Supplementary Dataset 4). We excluded from this puncta(+) group eight FOs that localized within nucleoli ("Methods"; Supplementary Fig. 1A, Supplementary Dataset 4; termed the "Nucleolar" category) and nine FOs that formed organized structures in the cytoplasm that were not round (termed the "Other" category, 9 FOs; see "Methods"; Supplementary Fig. 1B, Supplementary Dataset 4). Puncta(+) FOs formed puncta that localized exclusively within nuclei (50%), exclusively within the cytoplasm (33%), or within both compartments (17%) (Fig. 2b, middle panel; Supplementary Dataset 4). The abundance and sizes of puncta formed by FOs varied widely, in the nucleus (Fig. 2c), cytoplasm (Fig. 2d), and in both the nucleus and cytoplasm (Fig. 2e). Their morphologies were similar to those of several proteins shown previously to form condensates through phase separation and used here as positive controls[7,8,25–30] (Supplementary Fig. 1C). We reiterate, however, that our observation of condensate formation by a large proportion of the Expressed FOs is insufficient to establish whether formation occurs through phase separation. 53 FOs did not form puncta (termed puncta(-) FOs) and exhibited varied sub-cellular localization (Fig. 2b, right panel; Supplementary Dataset 4) and a diffuse fluorescence appearance consistent with an mEGFP empty vector negative control (Fig. 2f, g).

### Puncta(+) FOs cluster into distinct groups based on physicochemical features

We examined the 96 puncta(+) and 53 puncta(-) FOs to identify patterns of LLPS-associated sequence-derived physicochemical features. For each sequence, we calculated the values of 39 LLPS-relevant physicochemical features, a few drawn from the analyses discussed above (Fig. 1d), which included those known to influence multivalent interactions (Fig. 3a; Supplementary Dataset 5), and reported those as Z-scores with respect to those of human protein sequences in the Protein Data Bank (PDB)[31]. Human protein sequences in the PDB were used as a reference set due to their folded nature and generally

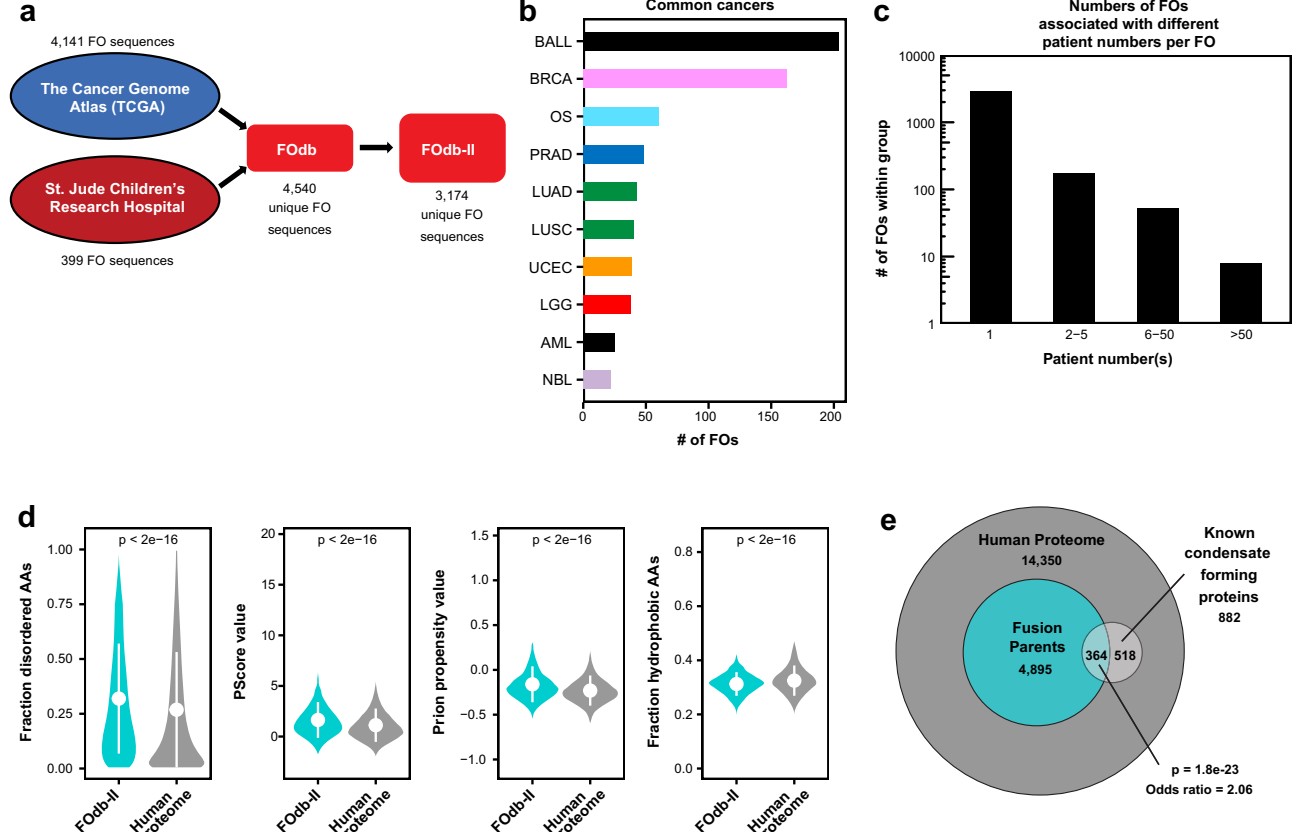

**Fig. 1 | Overview of the fusion oncoprotein (FO) database (FOdb). a** Schematic representation of sequence sources for the FOdb. Information on cancer type and number of patient occurrences was obtained for 3174 FO sequences; these are reported in FOdb-II. **b** Bar graph representation of the most frequently observed cancer types in which FOs were observed, based on analysis of FOs in the FOdb-II (BALL B-cell acute lymphoblastic leukemia, BRCA breast invasive carcinoma, OS Osteosarcoma, PRAD prostate adenocarcinoma, LUAD lung adenocarcinoma, LUSC lung squamous cell carcinoma, UCEC uterine corpus endometrial carcinoma, LGG low grade glioma, AML acute myeloid leukemia, NBL neuroblastoma). **c** Bar graph representation of the number of FOs associated with certain ranges of patient number(s) (number of patients in which the FO was observed) in FOdb-II.

**d** Comparison of the fraction of disordered amino acids, PScore values, Prion propensity values, and fractions of hydrophobic amino acids in the sequences in FOdb-II ($n = 3174$) to these values for the human proteome (using the Swiss-Prot database) ($n = 20,373$). Average values ± standard deviations of the mean are reported; significance was assessed using the two-sided t-test and no adjustments were made for multiple comparisons. **e** Euler diagram showing the overlap between the 4540 FOdb fusion oncoproteins' parent proteins and known condensate-forming proteins, as portions of the human proteome. The statistical significance of the overlap was assessed using Fisher's exact test (two-sided), and the log-odds ratio reflects the increased probability of fusion parents to be known condensate-forming proteins. All source data are provided as a Source Data File.

reduced propensity for phase separation and condensate formation[32]. To reduce redundancy between features, we excluded 14 features that had a mutual information value of > 0.5 with at least one other feature (Fig. 3b; Supplementary Dataset 5). From the remaining 25 features, we identified 12 that were discriminatory, i.e., those with Z-score values that varied the most between the puncta(+) and puncta(-) FO datasets (two-sided t-test, $p \leq 0.05$; Fig. 3c and Supplementary Dataset 5). Those 12 features captured charged residue content and patterning (Fig. 3c), disorder content, prion-like domain content, and enrichment in amino acids that promote pi-pi and pi-cation interactions (PScore)[22]. They also probe diverse physicochemical properties of groups of amino acids within sequences, including tracts of charged residues as probed by the Acidic/Basic Tract (ABT) algorithm[33], which suggests that different types of molecular forces driving multivalent interactions may underlie condensate formation by the puncta(+) FOs.

We next performed two-dimensional (2D) hierarchical clustering of puncta(+) FOs based on the 12 discriminatory features, which revealed four groups displaying different patterns of feature enrichments (Fig. 4a). Group membership was assessed for significance using the Pvclust method[34], with most members placed within groups with >90% confidence (Supplementary Fig. 2A). Importantly, the FO group clustering was not simply a result of high sequence identities amongst

FOs within each group: average sequence identity (SI_av) was <17% for Groups 1,3, and 4, and 23% for Group 2, due to the presence of multiple KMT2A and NUTM1 FOs (Fig. 4b).

Group 1, comprised of 19 FOs (Fig. 4c), showed enrichment (Z-score values > 2.0) in two prion domain-associated features [Prion propensity 1 and Fraction polar amino acids (AAs)], suggesting those types of interactions contribute to condensate formation in this group. Also enriched was a feature associated with pi-pi and pi-cation interactions (PScore). Two additional features, the number of amino acids in disordered regions (# Disorder AAs) and charged (chrg.) residue/ proline patterning (Ω, chrg. Pro pattern), were also weakly enriched (Z-score values > 1.0). These feature profiles derived from enrichments (>25% above background) of five amino acids: glycine, glutamine, serine, proline, and methionine (Fig. 4d), some of which are known to be involved in multivalent interactions underlying phase separation[22].

Group 2, comprised of 8 FOs (Fig. 4c), showed enrichment of disorder and charge-related features [# Disorder AAs, Acidic/Basic Tract (ABT) valence, # Positive (Pos.) AAs] and weak enrichment of two additional features, PScore and Prion propensity 1 (Fig. 4c, Group 2). These feature enrichments were correlated with enrichment of three amino acids: lysine, serine, and proline (Fig. 4d). The findings above suggest that different types of molecular forces driving multivalent

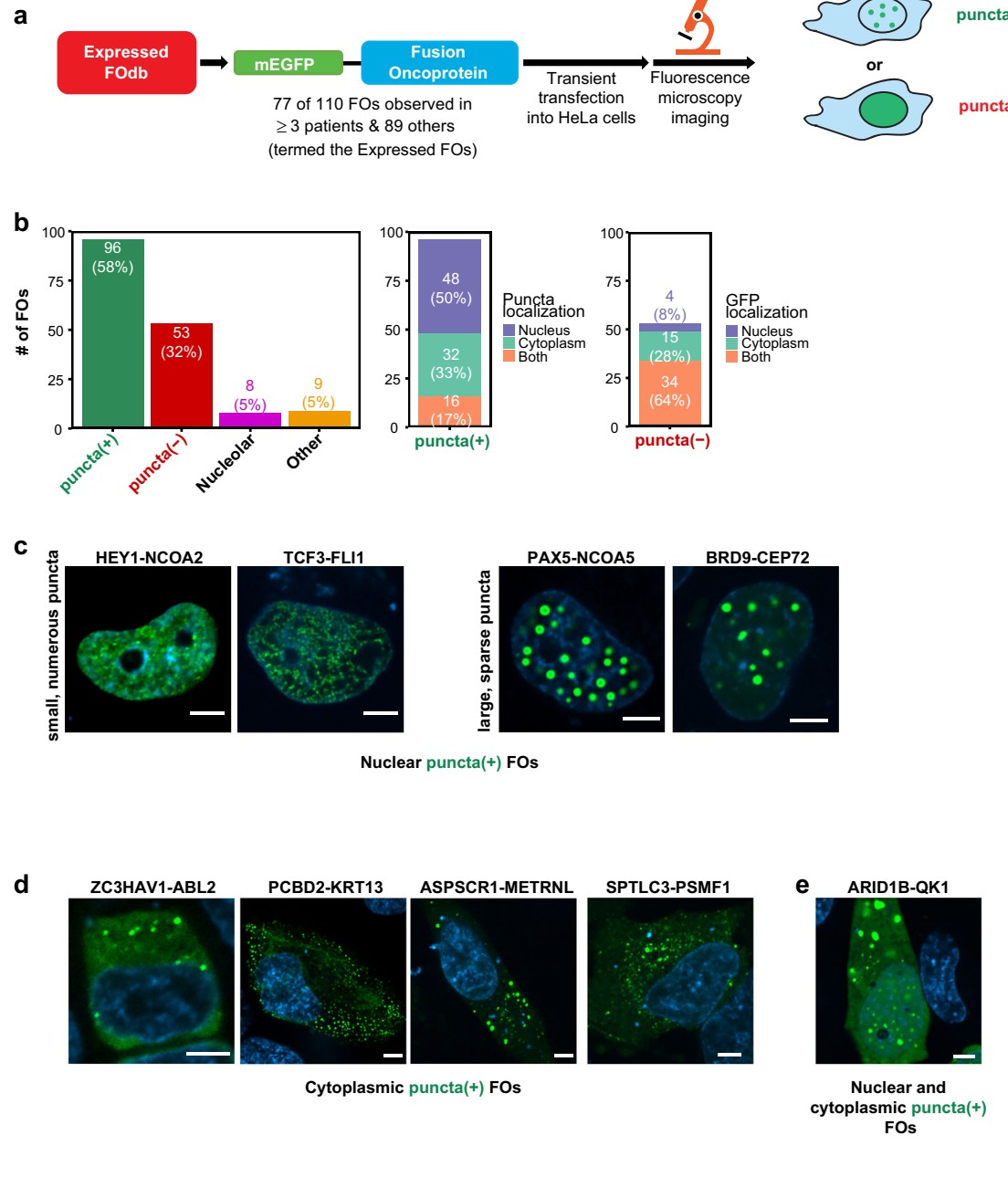

interactions promote cellular condensate formation by FOs in Groups 1 and 2.

FOs in Group 3 (26 FOs, Fig. 4c) exhibited feature enrichments similar to but weaker than those observed in Group 1 (e.g., smaller average values of the absolute average Z-score value, $|Z_{av}|_{av}$) and enrichment of histidine, serine, and proline amino acids (Fig. 4d). While many of the feature enrichments observed in Groups 1 and 3 FOs echo properties associated with phase separation by proteins (e.g., prion domain-like features and enrichment in pi-pi and pi-cation interactions)[22,35], the lack of enrichment of aromatic amino acids (and depletion in most cases, Fig. 4d) in these two FO groups is notable.

**Fig. 2 | Results of live cell imaging of mEGFP-tagged FOs from diverse human cancers. a** Schematic representation of the FO imaging workflow. A total of 166 FOs were analyzed for condensate formation in HeLa cells, termed the Expressed FOs. **b** Quantification of the number of FOs classified as puncta(+), puncta(-), nucleolar, or other (left). Within the puncta(+) and puncta(-) FOs, the number of FOs localized to either the nucleus, cytoplasm, or both was quantified based on puncta [for puncta(+) FOs] (middle) or diffuse GFP localization [for puncta(-) FOs] (right). Percentages are reported in parentheses. See "Methods" for details of these classifications. (C-E) Representative confocal microscopy images of live HeLa cells expressing mEGFP-tagged puncta(+) FOs localized to the nucleus (**c**), cytoplasm (**d**) or both compartments (**e**) based upon two biological replicates. **f** Representative confocal microscopy images of live HeLa cells expressing mEGFP-tagged puncta(-) FOs localized to the nucleus (left), cytoplasm (middle) or both (right) based upon two biological replicates. **g** Representative confocal microscopy images of live HeLa cells expressing mEGFP empty vector as a negative control based upon two biological replicates. In all images, the FO signal (green) is overlaid with the DNA signal (Hoechst dye, blue). All scale bars are 5 μm. All source data are provided as a Source Data File.

Group 4, the largest with 43 FOs, displayed heterogeneous, weak feature enrichments (Z-scores < 1.0), with only ABT density, ABT valence, and ABT balance enriched significantly with respect to the human PDB reference dataset (*p*-value ≤ 0.05; Fig. 4c). Amino acid enrichments for this group were correspondingly weak (Fig. 4d, Group 4). These observations on Group 4 indicate that our current feature set may not capture the physicochemical features underlying multivalent interactions associated with condensation by these FOs.

Interestingly, the sub-cellular localization of condensates formed by puncta(+) FOs was associated with the four physicochemical feature groups identified. The puncta formed by 17/19 FOs in Group 1 (89%), 7/8 FOs in Group 2 (88%), and 23/26 FOs in Group 3 (88%) localized within the nucleus, whereas 32/43 FOs in Group 4 (74%) localized within the cytoplasm (Fig. 4a, first column). These results suggest that the sub-cellular localization of puncta formed by FOs, a key aspect of their biological function(s), is encoded in the physicochemical properties that delineate the four feature groups for puncta(+) FOs.

### Puncta(-) FOs also cluster into distinct groups based on physicochemical features

We performed a similar 2D clustering analysis of feature/amino acid enrichments for the 53 puncta(-) FOs, which revealed three groups (Supplementary Fig. 2B). Most members were placed within those groups with high statistical significance (>90% confidence) (Supplementary Fig. 2C). Group clustering for puncta(-) FOs was also not a result of high sequence identities, which averaged <15% for all groups (Supplementary Fig. 2D).

Group 1′ (8 FOs) was enriched in three charge-related features [ABT density; δ, Charge (chrg.) pattern; and Fraction negative (neg.) AAs; Supplementary Fig. 2E] with corresponding enrichment of the two negatively charged amino acids (Supplementary Fig. 2F). In contrast, FOs in Group 3′ (10 FOs) displayed enrichment in two LLPS-associated features (PScore and Prion propensity 1) and weak enrichment of another feature (Fraction polar AAs, Supplementary Fig. 2E), with corresponding enrichment of glycine and proline amino acids (Supplementary Fig. 2F). Finally, Group 2′ (35 FOs) displayed diverse, weak enrichments that, on average, resulted in average Z-scores <1 (Supplementary Fig. 2E) and amino acid enrichments <25% (Supplementary Fig. 2F), indicating that FOs with physicochemical features similar to the average features of the human PDB reference set have a low propensity to form condensates.

Overall, these results show that the features and amino acid enrichments associated with FOs that do not form condensates are diverse, with two groups (Groups 1′ and 3′) exhibiting distinct enrichments and the third, the largest (Group 2′), being indistinct. Interestingly, the largest groups for puncta(+) and puncta(-) FOs, Groups 4 and 2′, respectively, exhibit similar average feature Z-score values and amino acid enrichments, suggesting that combinations of features and amino acid enrichments not revealed by our current analyses, and/or protein features we have not probed, determine the condensate-forming behavior of these FOs.

### Relationships between FO physicochemical features and condensate features

We next probed the properties of the condensates formed by 22 puncta(+) FOs using fluorescence recovery after photobleaching (FRAP). We randomly selected FOs from each of the four physicochemical feature groups (~20% of the members of each group) (Supplementary Fig. 3). The rate of recovery of mEGFP fluorescence after photobleaching reflects, at least in part, the rate of diffusion of the mEGFP-tagged FOs within, and into and out of, their respective condensates. However, quantitative analysis of FRAP data to obtain diffusion rates can be problematic[36], and we thus report average normalized FRAP curves (Supplementary Fig. 3A) and values of the average normalized fluorescence recovery between 40 and 50 seconds after bleaching (% recovery; Supplementary Fig. 3B). The results show that fluorescence recovery for FOs varied within each group, with notable differences between groups. None of the 10 FOs that formed cytoplasmic condensates recovered >50% after photobleaching, whereas 6/11 FOs that formed nuclear condensates exhibited recovery >50%; consequently, the average recoveries of cytoplasmic and nuclear FO condensates were different (Supplementary Fig. 3C). These findings recapitulate previous observations on the NUP98-HOXA9 FO, which forms hundreds of small nuclear condensates and exhibited FRAP;[8] and on EML4-ALK, which forms cytoplasmic condensates with variable and generally incomplete recovery[16]. Our results, together with the noted prior findings, suggest that the mobility of FOs into, out of, and within condensates may on average be higher for nuclear than cytoplasmic condensates, and can differ depending upon protein sequence and cellular context.

While performing time-lapse live cell fluorescence imaging for FRAP experiments, we noticed that puncta formed by 8 FOs, members of Groups 1, 3 and 4, underwent fusion events (*i.e.*, smaller condensates coalesce to form larger condensates) on variable timescales (Supplementary Fig. 3D). These results indicate that the condensates formed by this set of FOs display surface tension, suggesting that they have liquid-like properties.

### Predicting cellular condensation by FOs using physicochemical features and Machine Learning

With the experimentally validated puncta(+) and puncta(-) list of 149 FOs (excluding Nucleolar and Other FOs; Fig. 2b), we hypothesized that differences between the patterns of physicochemical features could be used to train a Machine Learning model to predict the cellular condensation behavior of the remaining FOs in FOdb-II. To this end, we applied supervised Machine Learning (ML) using the H₂O AutoML package[37] to develop a model to predict condensate formation propensity of FOs. To broadly represent physicochemical feature space, we used the 25 features with low mutual information (MI) (Fig. 3b) for all 96 puncta(+) and 53 puncta(-) FOs (termed the Training FOs; Fig. 5a) as training data. A Gradient Boosting Machines (GBM) model with 50 trees performed best amongst the 220 models tested (termed the FO-Puncta ML model). Using 25-fold cross validation with the Training FOs, performance metrics were: AUC [area under the ROC (receiver operating characteristic) curve], 0.88; AUCPR (area under the precision-recall curve), 0.94, and accuracy, 0.81 (Supplementary Dataset 6).

**a**

**39 physicochemical sequence features**

Charge (12)

Complexity (1)

Composition (3)

Conformation (1)

Disorder (2)

General (3)

Hydropathy (4)

Patterning (9)

Pi-pi, pi-cation (2)

Prion-like domain (2)

**b**

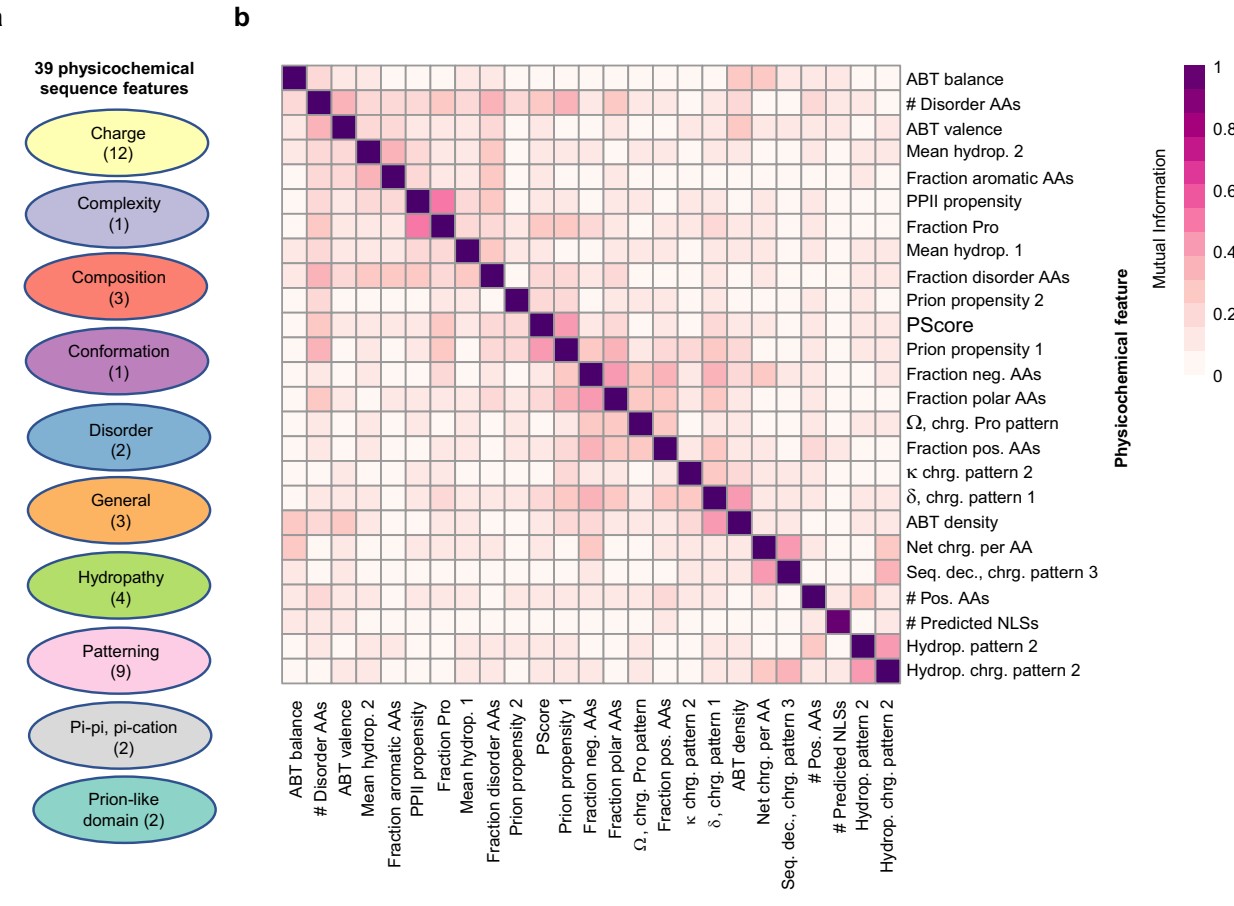

**c**

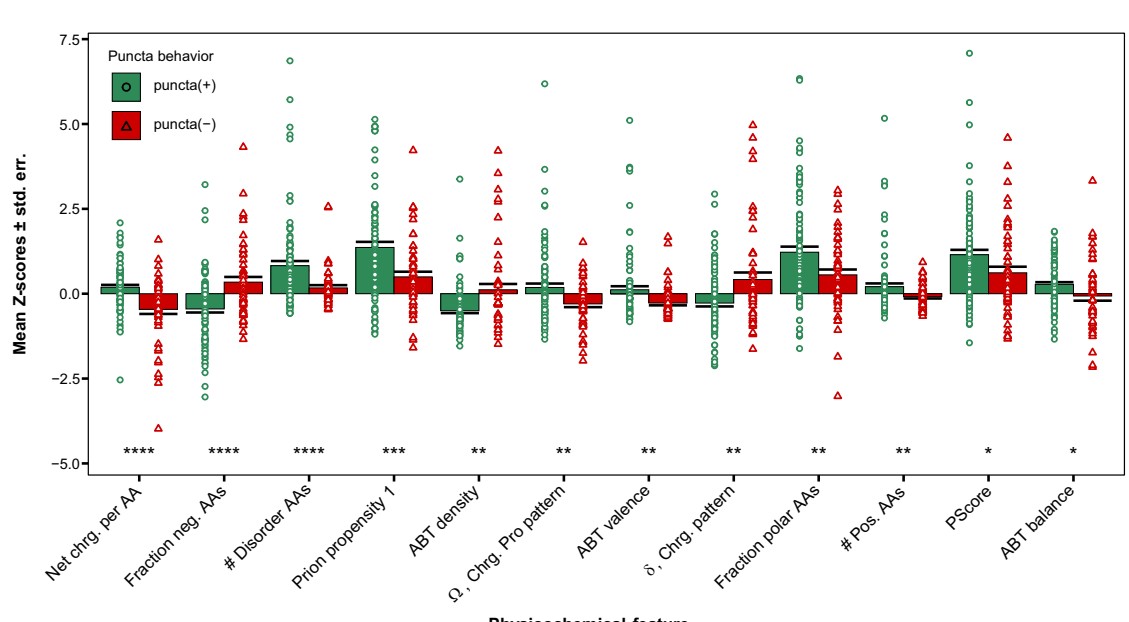

To verify FO-Puncta ML model performance, we applied it to 29 additional FOs (Verification FOs) with low sequence identity with the Expressed FOs (average sequence identity, 7.6 ± 0.0%), obtaining condensate formation probability values that ranged from -0.01 to 0.99. We experimentally tested the 29 Verification FOs for condensate formation in live cells and observed that 19 displayed puncta(+) and 10

puncta(-) behavior (Supplementary Fig. 4A–D, Supplementary Dataset 4). Using a probability value of 0.83 as the threshold for predicting puncta(+) behavior, the FO-Puncta ML model correctly predicted condensate formation for 17/19 puncta(+) FOs and diffuse localization for 6/10 puncta(-) FOs (Supplementary Fig. 4E). The performance metrics [AUC, 0.73; accuracy, 0.79 (Fig. 5b; Supplementary Dataset 6)],

**Fig. 3 | Physicochemical feature differences between the puncta(+) and puncta(-) Expressed FOs. a** The values of 39 physicochemical features, which fall into ten broad categories, were computed based on the amino acid sequences of 96 puncta(+) and 53 puncta(-) FOs. The numbers in parentheses indicate numbers of features in each category. See Supplementary Dataset 5 for physicochemical feature definitions. **b** Mutual information matrix assessing redundancy between the 39 physicochemical features. A mutual information cut-off of 0.5 or less was applied to reduce the number of features to 25. **c** Quantification of the enrichment or depletion of the 12 non-redundant and most significant physicochemical features (out of 25) for puncta(+) and puncta(-) FOs with respect to the human sequences within the Protein Data Bank (PDB). Values are reported as mean Z-scores ± standard error. The Z-scores values of the puncta(+) ($n = 96$) and puncta(-) ($n = 53$) FOs for each feature are shown in green circles and red triangles, respectively, along the

*y*-axis. Significance was assessed using two-sided *t*-test and no adjustment were made for multiple comparisons (*$p < 0.05$; **$p < 0.01$; ***$p < 0.001$; ****$p < 0.0001$). Features include: Net charge per amino acid (Net chrg. per AA); Fraction negative amino acids (Fraction neg. AAs); Number of disordered amino acids (# Disorder AAs); prion-like domain content (Prion propensity 1); Acidic/Basic Tract density, valence and balance (ABT density, ABT valence and ABT balance); Ω, Charged residue/proline patterning (Ω Chrg. Pro pattern); δ, Charged residue patterning (δ Chrg. Pattern); Fraction polar amino acids (Fraction polar AAs); Number of positive amino acids (# Pos. AAs); and pi-pi and pi-cation interaction score (PScore). See Supplementary Dataset 5 for additional information on these and other physicochemical features used in these analyses. All source data are provided as a Source Data File.

were similar to those of our cross validation testing (Supplementary Dataset 6).

The accuracy of the predictions indicates that the 25 low-MI sequence-derived features used to develop the FO-Puncta ML model capture physicochemical properties of FOs associated with condensate formation in cells. However, how the different physicochemical features contribute to the prediction of FO condensation behavior is inaccessible, an intrinsic limitation of certain types of ML approaches. To address this limitation, we determined the contributions of the 25 features to FO-Puncta predictions for the 29 Verification FOs using Shapley Additive exPlanations (SHAP; Fig. 5c)[38]. The SHAP contribution score enumerates the magnitude of the contribution of the normalized value of the individual features for a particular FO to the prediction of puncta(+) or puncta(-) behavior (positive and negative SHAP contribution values, respectively). These results indicated that the features that contribute most to FO-Puncta ML model predictions (located toward the top of Fig. 5c) quantify various charge-, disorder- and prion propensity-related properties of FO sequences (Fig. 5c, left) and that no single feature can correctly predict any particular FO's condensation behavior. Instead, different combinations of features contribute to predictions of condensation behavior (Fig. 5c, right). These results show that the charged residue content and patterning within FO sequences are important determinants of their condensation behavior.

We next compared the performance of the FO-Puncta ML model to three established protein phase separation predictors, catGranule[39], DeePhase[40] and FuzDrop[41] (Fig. 5d; Supplementary Fig. 5A), which yielded AUC values of 0.59, 0.65 and 0.65, respectively for 178 FOs (combined Training and Verification FO sets) (Supplementary Dataset 4 and 6). These results indicate that the sequence and physicochemical features of puncta(+) and puncta(-) FOs in our combined FO set are different from those of the various protein sets used to develop and/or test those other phase separation predictors. Indeed, catGranule takes into account disorder content and nucleic acid binding propensities together with sequence length and arginine, glycine, and phenylalanine content[39], whereas DeePhase uses integrated, trained neural network-based language and knowledge-based models[40]. In contrast, FuzDrop applies physical principles to predict the conformational entropy associated with nonspecific side-chain interactions, which are utilized by proteins that form condensates[41]. Thus, our FO-Puncta ML model is an accurate predictor of condensation behavior by FOs identified in human cancers and will certainly identify proteins in the human proteome with physicochemical features associated with condensate formation by FOs. However, it is not designed to recognize the broader sequence-based feature landscape of phase separation-prone human proteins.

### Using the FO-Puncta ML model to modulate FO condensation behavior

To further explore the predictive utility of our FO-Puncta ML model, we selected eight puncta(+) FOs spanning the four feature groups and

performed mutagenesis guided by analysis of changes in ML model parameters to weaken condensate formation behavior. The mutant FO sequences were then tested in HeLa cells for condensate formation. To guide our mutagenesis strategy, we first identified the physicochemical features with the largest SHAP contribution values for the ML model predictions for the eight puncta(+) FOs (Suppl. Supplementary Fig. 6A; Supplementary Dataset 7). We next analyzed the values for these highly predictive features, and also examined the corresponding amino acid enrichments within IDRs (Suppl. Supplementary Fig. 6B, C; Supplementary Dataset 7). To weaken condensation behavior, we performed mutagenesis of multiple, enriched residues within IDRs to modulate multivalent interactions[42]. After introducing mutations, we reassessed SHAP contribution, physicochemical feature, and amino acid enrichment values, and the condensation probabilities determined by the FO-Puncta ML model, to determine whether the mutations switched the FO-Puncta predictions from puncta(+) to puncta(-). This FO-Puncta ML model-guided mutagenesis process was iterated until the desired switch was achieved and is illustrated below for several FOs.

Analysis for the puncta(+) ML model prediction for SS18-SSX1 revealed that "ABT density", "ABT balance", and "Fraction neg. AAs" were the three features with the largest SHAP contribution values (Fig. 6a, top left). The Z-score values of the "ABT density" and "Fraction neg. AAs" features were lower than for the human PDB reference set, while that for "ABT balance" was similar to the reference set (Fig. 6a, bottom left). Further, analysis of amino acid enrichments within IDRs revealed that glutamine and methionine were the most enriched residues (Fig. 6b, left). To reverse the puncta(+) FO-Puncta ML model prediction, we mutated either 50% or 100% of all glutamine and methionine residues to the negatively charged residues, aspartic acid or glutamic acid. This resulted in mutant FO sequences with reduced enrichment of glutamine and methionine, and a reversal of the features, "ABT density", "ABT balance", and "Fraction neg. AAs", to negative SHAP contribution values, corresponding to prediction of puncta(-) behavior (Fig. 6a, b). These changes led to a decrease in the FO-Puncta ML model prediction from 0.99 for the unmutated SS18-SSX1 FO sequence to 0.19 and 0.07, respectively, for the two mutants (Fig. 6c; Supplementary Dataset 7).

Similarly, analysis of SHAP contribution values for NUP98-HOXD13 revealed that the features, "ABT density", "Fraction neg. AAs", and "Net chrg. per AA", were three of the four largest positive contributors to its puncta(+) prediction (Fig. 6a, top middle). The features, "ABT density" and "Fraction neg. AAs", had negative Z-score values for this FO, while that for "Net chrg. per AA" was slightly positive (Fig. 6a, bottom middle). Analysis of IDRs in this FO revealed an enrichment of threonine, phenylalanine, and glycine residues (Fig. 6B, middle). We therefore mutated all phenylalanine and glycine residues to negatively charged residues, aspartic acid or glutamic acid, which caused the SHAP contribution values for the three noted features to become negative (Fig. 6a, top middle). Acidic residues were introduced in part because they were depleted on average in

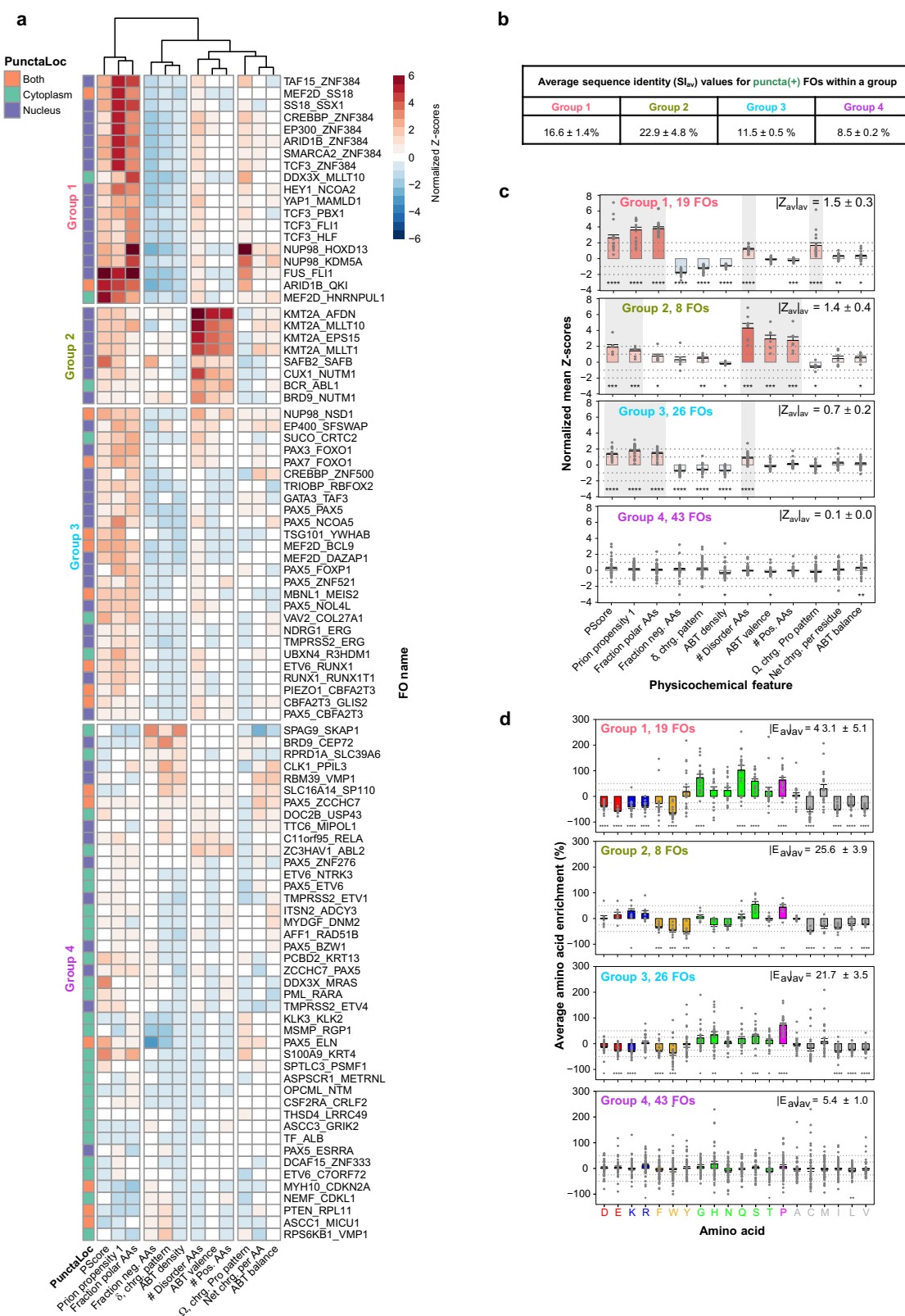

**b** Average sequence identity (SI$_{av}$) values for puncta(+) FOs within a group

| Group 1 | Group 2 | Group 3 | Group 4 |
|---------|---------|---------|---------|
| 16.6 ± 1.4% | 22.9 ± 4.8 % | 11.5 ± 0.5 % | 8.5 ± 0.2 % |

puncta(+) and enriched in puncta(-) FOs, respectively (Fig. 3c). The mutations caused the FO-Puncta condensation probability to decrease from 1 to 0.08 for the mutated sequence (Fig. 6c; Supplementary Dataset 7). For the DOC2B-USP43 FO, analysis of SHAP contribution values revealed that the feature, "Net chrg. per AA", was the largest positive contributor, and the additional features, "ABT balance", "ABT valence", and "Fraction pos. AAs" were also positive SHAP

contributions (Fig. 6a, top right). The Z-score values of these features were all positive for the DOC2B-USP43 FO sequence (Fig. 6a, bottom right). The IDRs within this FO are enriched in proline and arginine residues (Fig. 6b, right). To reduce enrichment of positive charge, we mutated all of the positive arginine residues to neutral alanine residues. This resulted in an FO-Puncta ML model condensation probability decrease from 1 for the unmutated sequence to 0.75 for the

**Fig. 4 | Physicochemical features of the puncta(+) Expressed FOs.**
**a** 2-dimensional (2D) hierarchical clustering of the puncta(+) FOs on the basis of the 12 most discriminatory physicochemical features. FO names are reported on the vertical axis. The values of features are reported on the horizontal axis. The first column (left) represents localization of the FO puncta (nucleus, purple; cytoplasm, green; or both, orange). FOs cluster into four groups (Groups 1–4) based on 2D hierarchical cluster analysis. The names of the physicochemical features used for clustering are given at the bottom. The significance of the different clusters/groups is given in Supplementary Fig. 2A. **b** Average sequence identity ± standard error for pairwise comparison of all FOs within each of the individual groups in (**d**). **c** Quantification of the mean enrichment or depletion values for the 12 physico-chemical features for Groups 1–4. Values are reported as mean Z-scores ± standard error and normalized to the human sequences in the PDB. The average values of the absolute mean Z-scores ± standard error are reported in the top right of each plot. The Z-scores values of the puncta(+) FOs for each feature are shown in solid gray circles along the y-axis for Groups 1–4. Gray boxes highlight the features with significant enrichments noted in the text (one standard deviation or greater above the mean Z-scores). **d** Quantification of the average amino acid enrichment or depletion ± standard error for FO sequences in Groups 1–4. The amino acid enrichment values of the puncta(+) FOs for each amino acid are shown in solid gray circles along the y-axis. The mean of the absolute average enrichments ± standard error are reported in the top right of each plot. In both (**c**) and (**d**), significance was calculated using two-sided t-test with respect to the human sequences in the PDB and no adjustment were made for multiple comparisons (*$p < 0.05$; **$p < 0.01$; ***$p < 0.001$; ****$p < 0.0001$). All source data are provided as a Source Data File.

mutant (Fig. 6c; Supplementary Dataset 7). A similar analytical process was applied to mutate five additional FOs, as described in Supplementary Dataset 7. The FO sequence analyses discussed above can be performed using the SAK web server available at https://sak.stjude.org.

Testing of the condensate formation behavior of the 11 mutant FO sequences in HeLa cells showed that seven were experimentally determined to be puncta(-) (Fig. 6c). The other mutated FOs (SLC16A14-SP110, PAX7-FOXO1, and SS18-SSX1) displayed condensates but in a smaller percentage of cells than the unmutated sequences (Fig. 6c). Highlighting the impact of the ML model-guided sequence modifications, the morphology of condensates formed by SS18-SSX1 were dramatically altered by the introduced mutations (Supplementary Fig. 7A). These results illustrate how our FO-Puncta ML model, together with analysis of SHAP contribution, physicochemical feature, and amino acid enrichment values, can be used to inform mutagenesis of FOs to reduce their condensate formation behavior.

## Relationships between condensate formation, sub-cellular localization and function of FOs

While the biological functions and contribution to oncogenic pheno-types are known for many of the FOs that we experimentally tested in cells (Supplementary Dataset 4), these data are lacking for others. We addressed this knowledge gap by annotating the functional features of all tested FOs using sequence analysis. The FOs we studied are comprised of both IDRs and folded domains. While assigning function based on the amino acid sequences of IDRs is challenging, it is straightforward for folded domains. To do this, we first identified conserved domains (CD) within the amino acid sequences of all 115 experimentally tested puncta(+) and 63 puncta(-) FOs (Training + Verification FOs) using the CDD/SPARCLE conserved domain database[43] and extracted functional terms from CD definitions. Because past studies have established that FOs known to form nuclear or cytoplasmic condensates through phase separation exhibit different biological functions (e.g., transcriptional regulation by nuclear FOs or regulation of cell signaling by cytoplasmic FOs)[6–12,16,19], we compiled functional terms separately for puncta(+) FOs that formed condensates in the nucleus, the cytoplasm, or both compartments (Supplementary Dataset 8). We also compiled functional terms for all puncta(-) FOs with those localization patterns (Supplementary Dataset 8).

A functional term that was significantly enriched for puncta(+) FOs in all compartments, as well as in puncta(-) FOs, was transcription ($p$-value < 0.001; Fig. 7a–c); other functional terms differed between nuclear and cytoplasmic puncta(+) FOs. For example, apart from transcription, the terms observed four or more times for nuclear puncta(+) FOs were chromatin (9 FOs, $p$-value < 0.01), RNA binding (6 FOs), oligomerization (4 FOs, $p$-value < 0.01) and ATPase (4 FOs, $p$-value < 0.01), while those for cytoplasmic puncta(+) FOs were protein binding (12 FOs, $p$-value < 0.0001), protein kinase (7 FOs, $p$-value < 0.0001), oligomerization (6 FOs, $p$-value < 0.001), and cell signaling (9 FOs, $p$-value < 0.05). The association of the terms transcription, chromatin and RNA binding with nuclear puncta(+) FOs suggested roles in regulation of gene expression, as has been demonstrated for NUP98 and other nuclear condensate-forming FOs[6–12,19]. While several cytoplasmic puncta(+) FOs displayed transcription as a functional term, others displayed terms associated with regulation of cell signaling (protein binding, protein kinase and cell signaling), which were rarely observed for nuclear puncta(+) FOs (Fig. 7a, b). The latter observation is consistent with previous findings showing that the EML4-ALK and CCDC6-RET FOs form cytoplasmic condensates that drive aberrant cell signaling[16–18].

Most puncta(-) FOs were localized within the cytoplasm (15 FOs) or both in the cytoplasm and nucleus (43 FOs), with very few (5 FOs) localized exclusively within the nucleus (Fig. 7d–f). Interestingly, cell signaling-related terms [e.g., protein kinase ($p$-value ≤ 0.001), protein binding ($p$-value ≤ 0.001), and cell signaling ($p$-value ≤ 0.05)] were enriched in the puncta(-) FOs that localized in the cytoplasm (Fig. 7e, f). While a few condensate-forming FOs were previously shown to drive aberrant cell signaling in the cytoplasm[16–18], most of the FOs that we tested that displayed cell signaling functional features within their amino acid sequences did not form condensates.

We performed an independent analysis of the functional properties of the puncta(+) and puncta(-) FOs using InterPro2GO domain annotations[44] to identify Gene Ontology-derived functional terms, and found generally similar associations between FO condensation behavior, sub-cellular localization, and function (Supplementary Fig. 8A–F and Supplementary Dataset 9). Together, these results and those above suggest that the cellular functions of FOs are aligned with their physicochemical features, condensate formation behavior, and sub-cellular localization.

## Predicting cellular condensation, localization, and function for ~3000 FOs using physicochemical features and Machine Learning

We applied the FO-Puncta ML model to all FOs in FOdb-II not included in the Training or Verification FO sets (2999 FOs, termed Untested FOs) and found that 67% are predicted to form condensates (Fig. 8a; Supplementary Fig. 9A), a percentage slightly larger than that for Expressed FOs that formed condensates in HeLa cells (58%; Fig. 2b). This result indicates that many FOs beyond those experimentally tested herein by us, and those characterized previously by us and others, are likely to form condensates in cells.

We next analyzed the predicted, untested puncta(+) and puncta(-) FOs (2999 FOs) based on the physicochemical features of their sequences. Among the 1999 predicted puncta(+) FOs, 41% (Fig. 8b) matched one of the four feature groups identified for the 96 Training puncta(+) FOs (Figs. 4a, and 8b, c). Such matches were not rooted in amino acid sequence identities, as the values of $SI_{av}$ for all pairs of FOs in matching groups for the two sets of FOs were ≤ 12% (Fig. 8d–g), and thus should reflect the presence of regions with physicochemical features that are likely to promote condensate formation by FOs. Moreover, these matches (35% in Groups 1-3 and 6% in Group 4) allow

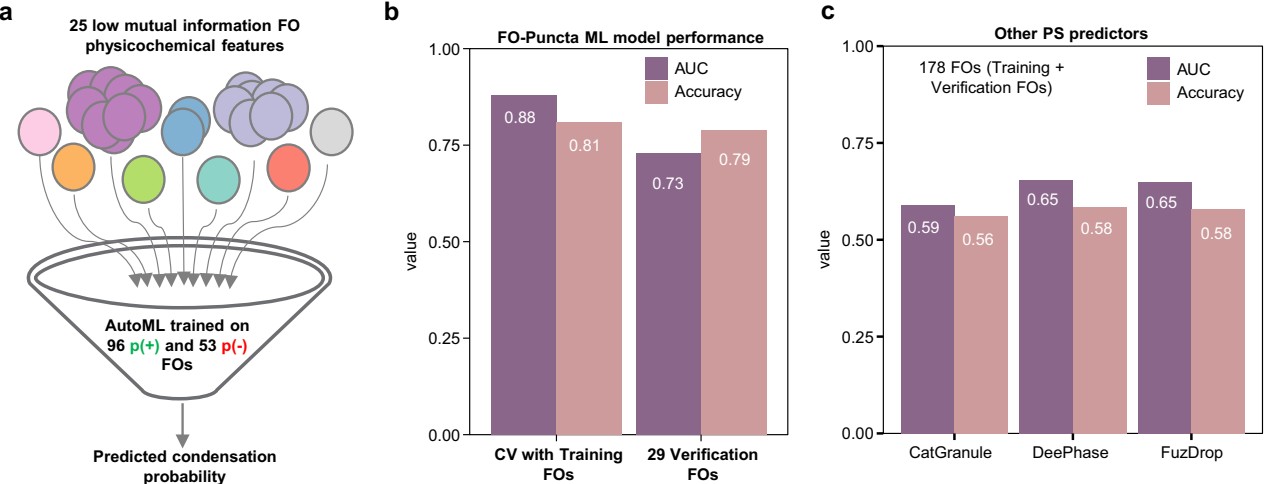

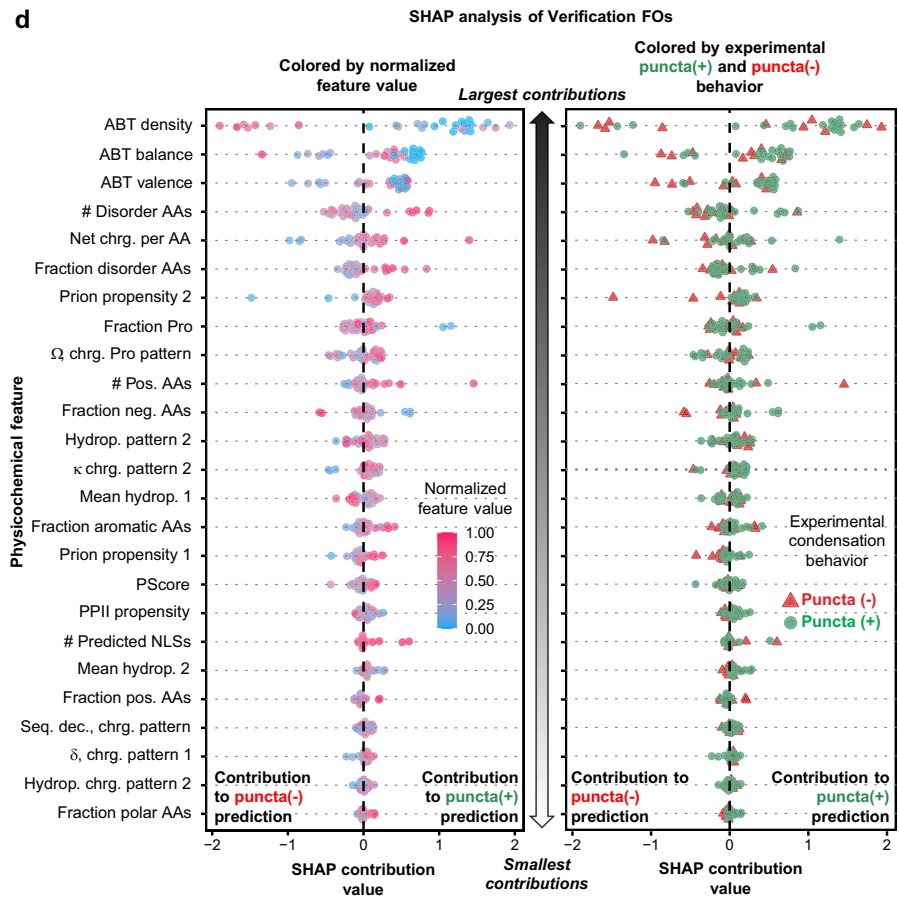

**Fig. 5 | A Machine Learning model for predicting condensate formation probability of FOs. a** Supervised Machine Learning was used to develop a Gradient Boosting Machine model (termed FO-Puncta ML model) trained using the 25 low mutual information physicochemical features for 96 puncta(+) [abbreviated p(+)] and 53 puncta(-) [abbreviated p(-)] FOs (termed Training FOs). **b** Performance metrics [area under the curve (AUC, purple) and accuracy (cream)] for the FO-Puncta ML model using cross validation (CV) with the Training FOs and independent testing of 29 Verification FOs. **c** SHapley Additive exPlanations (SHAP) analysis for the 29 Verification FOs colored by normalized physicochemical feature value

(left) or condensation behavior (right). The features are ranked by the magnitude of their relative SHAP contributions, with those with the largest contributions toward the top. Positive values of the SHAP contributions are for predictions of puncta(+) behavior and negative values for puncta(-) behavior. **d** Performance metrics [area under the curve (AUC, purple) and accuracy (cream)] for prediction of phase separation behavior for the combined Training and Verification FOs (reported herein) using three previously published phase separation predictors (catGranule, DeePhase, and FuzDrop). All source data are provided as a Source Data File.

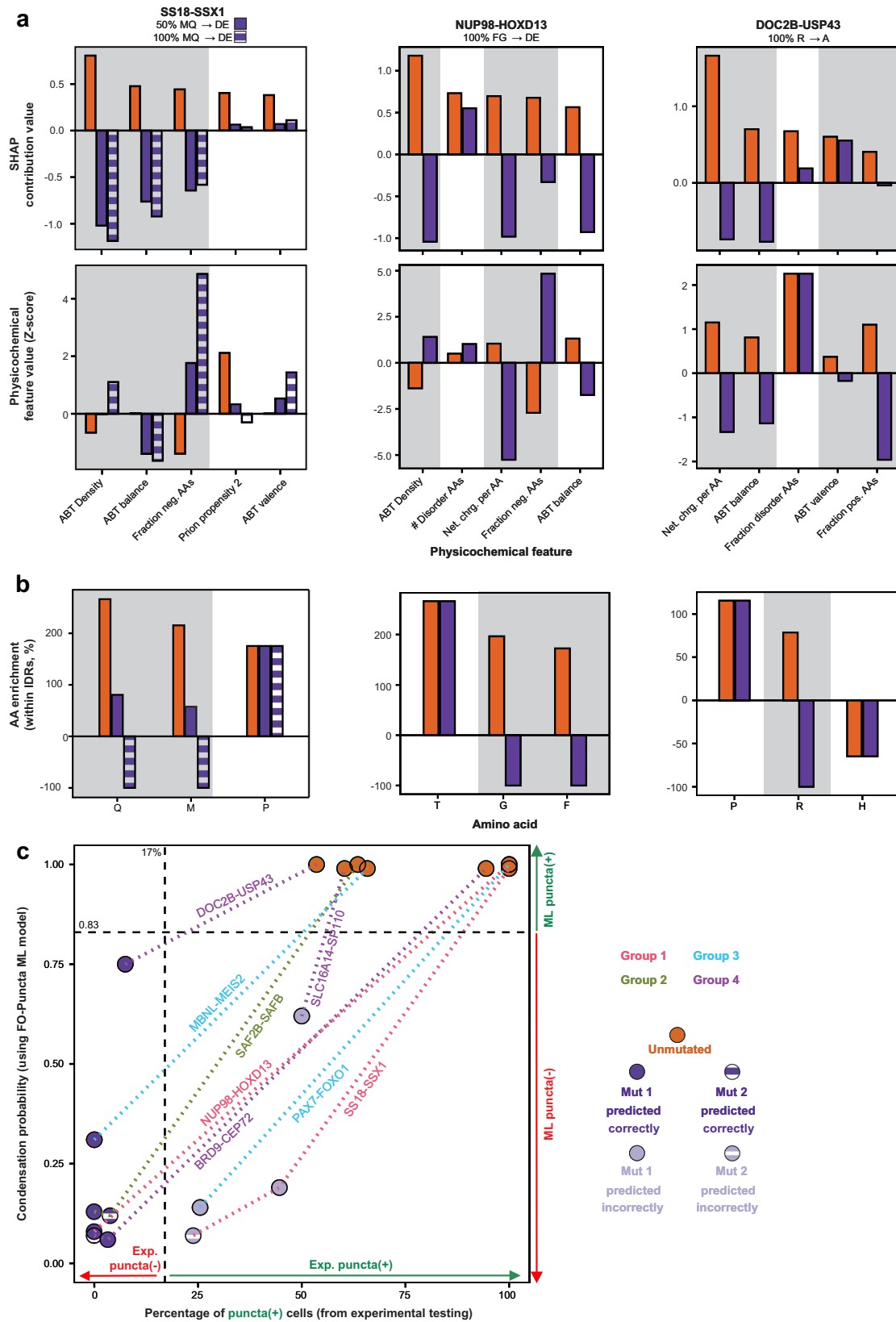

us to confidently assign the sub-cellular localization of their condensates and their biological function (Groups 1-3, nuclear condensates and regulation of gene expression; Group 4, cytoplasmic condensates and regulation of cell signaling). Among the predicted puncta(-) FOs (1000 FOs), 47% can be matched to one of the three original puncta(-) FO groups (Supplementary Figs. 2B and 9B, C), and

those matches were not due to amino acid sequence similarities (Supplementary Fig. 9D). We note that more than half of the predicted puncta(+) and puncta(-) FOs had physicochemical features that did not match those of the functionally annotated groups. This observation reflects the highly diverse physicochemical properties of FOs and indicates that much remains to be learned regarding their

**Fig. 6 | Mutagenesis of puncta(+) expressed FOs. a** SHapley Additive exPlanations (SHAP) (top) and feature value (bottom) analysis for unmutated (orange) and mutated (purple shades) FOs. Positive SHAP contribution values indicate the magnitude of contributions to puncta(+) predictions, while negative SHAP contribution values indicate the magnitude of contributions to puncta(-) predictions. The five features with the largest SHAP contributions based on absolute values are listed for each FO and those used in mutant design are highlighted in gray. See Figure S6 and Supplementary Dataset 7 for the full complement of features and values for all eight FOs that were mutated. See Supplementary Dataset 5 for additional information on the physicochemical features used in these analyses. **b** The top three amino acid enrichments or depletions within the intrinsically disordered regions (IDRs) of the specified unmutated (orange) and mutated (purple shades)

FOs. Those used in mutant design are highlighted in gray. See Figure S6 and Supplementary Dataset 7 for the full complement of IDR amino acid enrichments or depletions for the eight FOs that were mutated. **c** Plot of the FO-puncta ML model condensation probability prediction on the y-axis and experimentally determined percentage of puncta(+) cells on the x-axis. Unmutated FOs are in orange. Mutated FOs that were correctly predicted as puncta(-) are in purple. Mutant FOs that were puncta(+) are in gray. Lines connect unmutated FOs to their mutated counterparts and are color-coded based on the group (Group 1–4) from which the original FO was derived. FO names are indicated along the lines. The FO-Puncta ML model cut-off for puncta(-) classification is less than 0.83. The experimental puncta(-) cut-off is less than 17% of cells with puncta. See Fig. S7 for representative cell images of each unmutated and mutated FO. All source data are provided as a Source Data File.

condensation behavior and function in cells. Nevertheless, the insights we have gained for the FOs in FOdb-II that matched functionally annotated feature patterns represent a significant advance in our understanding of FO behavior.

### Visualizing the FO condensate landscape

Some parent proteins comprise many different FOs through fusion to multiple other parent proteins, some of which in turn are found in multiple additional FOs. To capture this complexity, we used Cytoscape[45] to create the FO condensate network, or landscape, with the network edges connecting nodes (one node for each unique parent protein and edges representing a fusion event) colored to indicate condensate formation behavior (true behavior for Training and Verification FOs and predicted behavior for Untested FOs; Fig. 9a and Supplementary Fig. 10A, respectively). In these landscape graphs, green edges represent puncta(+) FOs and red edges, puncta(-) FOs, respectively. As expected, many of the FOs we tested in cells appear within highly branched regions of the network (Fig. 9a), with green edges surrounding parent proteins with the greatest number of partners (termed the degree value). Analysis of the 15 parents with a degree value of ≥ 3 showed that all except JAK2 formed FOs that were largely puncta(+) (more than 67% puncta(+); Fig. 9b). The average degree (computed as the average number of green or red edges, respectively, associated with FO parents; see "Methods") for parents associated with puncta(+) FOs in this set (0.8 ± 0.1) was greater than that for puncta(-) FOs (0.5 ± 0.0, p-value, $6.1 \times 10^{-6}$), showing that FOs formed with high-degree parents are more likely to form condensates than those formed with low-degree parents. Additionally, FOs formed with a high-degree parent are more likely to function in the regulation of gene expression than cell signaling (Fig. 9b).

We also analyzed the Untested FOs (2999 FOs; 67% predicted to be puncta(+) and 33% to be puncta(-)) using Cytoscape. The prevalence of puncta(+) FOs can be appreciated in the context of the FO condensate network (Supplementary Fig. 10A). The puncta(+) versus degree visualization of the Untested FO condensate landscape (Fig. 9c) revealed a dichotomy between predicted FO condensation behavior, and likely sub-cellular localization and biological function. For example, FOs with one high-degree parent (degree ≥ 3) that are often predicted to form condensates (e.g., FOs represented within circles above the 50% puncta(+) value in Fig. 9c) are also often predicted to localize within the nucleus and regulate gene expression (purple shades in Fig. 9c). In contrast, FOs with high-degree parents that are often predicted to not form condensates (e.g, FOs within circles below the 50% puncta(+) value in Fig. 9c) are mostly predicted to localize within the cytoplasm and regulate cell signaling (blue shades in Fig. 9c). Interestingly, the color shading in Fig. 9c, which encodes the percentage of functionally annotated FOs within a circle predicted to be involved in either gene expression or cell signaling function, intensifies at the top and bottom of the graph, respectively, thus revealing a dichotomy between predicted condensation behavior and biological function. We note that the FOs represented in the different regions of Fig. 9c are associated with diverse cancer types and variable patient numbers.

Finally, we noted the striking juxtaposition of two high-degree FO parents in the network for the Training and Verification FOs, PAX5 and JAK2, that displayed almost completely opposite puncta formation behavior (PAX5, mostly puncta(+) FOs and JAK2, all FOs puncta(-); Fig. 9A). We leveraged the SHAP contribution values for the PAX5 and JAK2 FOs determined through analysis of the FO-Puncta ML model to identify physicochemical features whose values were different between the two groups of FOs. No single physicochemical feature explains the switch in condensation behavior between the two FO parent sub-networks. Instead, differences in the normalized Z-score values and SHAP contribution values for three features, ABT density, ABT balance, and Net charge per AA (Supplementary Fig. 10B) collectively provide insight into the switch. These features capture different types of charge characteristics of the FO amino acid sequences, with the greatest difference in values between the puncta(+) and puncta(-) FOs observed for the ABT density and Net charge per AA features. These observations reveal the complexity of relationships between the amino acid sequences, physicochemical features and condensation behavior of FOs and highlight the need for Machine Learning methods to identify patterns underlying these relationships.

## Discussion

We present databases, microscopy images, reagents, and computational tools to understand the relationships between the amino acid sequence-based physicochemical features of FOs and their cellular behavior, including condensate formation, sub-cellular localization and biological function (Fig. 10). We provide experimental results for condensate formation for 195 FOs, along with vectors for their mEGFP-tagged expression, and annotation of their physicochemical features, sub-cellular localization, and biological functions. Importantly, the FOs we tested are associated with diverse cancer types. The extensive data we report will be a valuable resource to investigate the molecular mechanisms, and involvement of condensate formation by FOs in the oncogenic mechanisms underlying their associations with diverse cancers.

We leveraged our findings and the physicochemical features of FOs to develop an accurate predictor of condensate formation behavior (the FO-Puncta ML model) and provide predictions for ~3000 additional, cancer-associated FOs. For 44% of these, we also provide annotation of predicted sub-cellular localization and biological function. These data are a resource for hypothesis-based research into the roles and mechanisms of condensate formation by FOs in cancer biology and beyond, by providing insight into patterns of protein physicochemical features and cellular condensate formation behavior. Further, we demonstrate how analysis of FO-Puncta ML model parameters, physicochemical features, and amino acid enrichments, can be leveraged to modulate the condensation behavior of FOs, providing opportunities to engineer synthetic, condensate-forming FOs in the future. Our FO sequence data, imaging Datasets, reagents, and computational tools will be valuable to members of the growing biomolecular condensate community.

**Fig. 7 | Conserved Domain and functional analysis of puncta(+) and puncta(-) Training and Verification FOs.** Functional terms identified from the Conserved Domain Database (CDD) are shown for the 115 puncta(+) Training and Verification FOs localized to the nucleus (**a**), cytoplasm (**b**), and both compartments (**c**). Functional terms identified from the Conserved Domain Database (CDD) are shown for the 63 puncta(-) Training and Verification FOs localized to nucleus (**d**), cytoplasm (**e**), and both compartments (**f**). The colors of the bars represent the three major functional classes, regulation of gene expression (including transcription, chromatin, and RNA binding Conserved Domain functional terms; purple), regulation of cell signaling (including protein kinase, protein binding, and cell signaling Conserved Domain functional terms; blue) and other functions (gray). The numbers in each bar indicate the number of unique FOs with the noted functional term, and asterisks indicate statistically significant over-representation based on $p$-value estimates from 100,000-fold one-sided resampling with replacement using identically-sized protein sets ($*p < 0.05$; $**p < 0.01$; $***p < 0.001$). All source data are provided as a Source Data File.

The physicochemical features of the experimentally tested, puncta(+) FOs are diverse, indicating that multiple mechanisms of multivalent interactions are operative during condensate formation. However, our analyses did reveal certain patterns of feature enrichment. For example, puncta(+) FOs are enriched in most charge-related features, with the exception being depletion in negative charge. Additionally, they are enriched in prion domain-like features and score highly for pi-pi and pi-cation interactions. Interestingly, these enrichments are not distributed uniformly amongst puncta(+) FOs, with Groups 1 and 3 enriched in the latter features and generally depleted

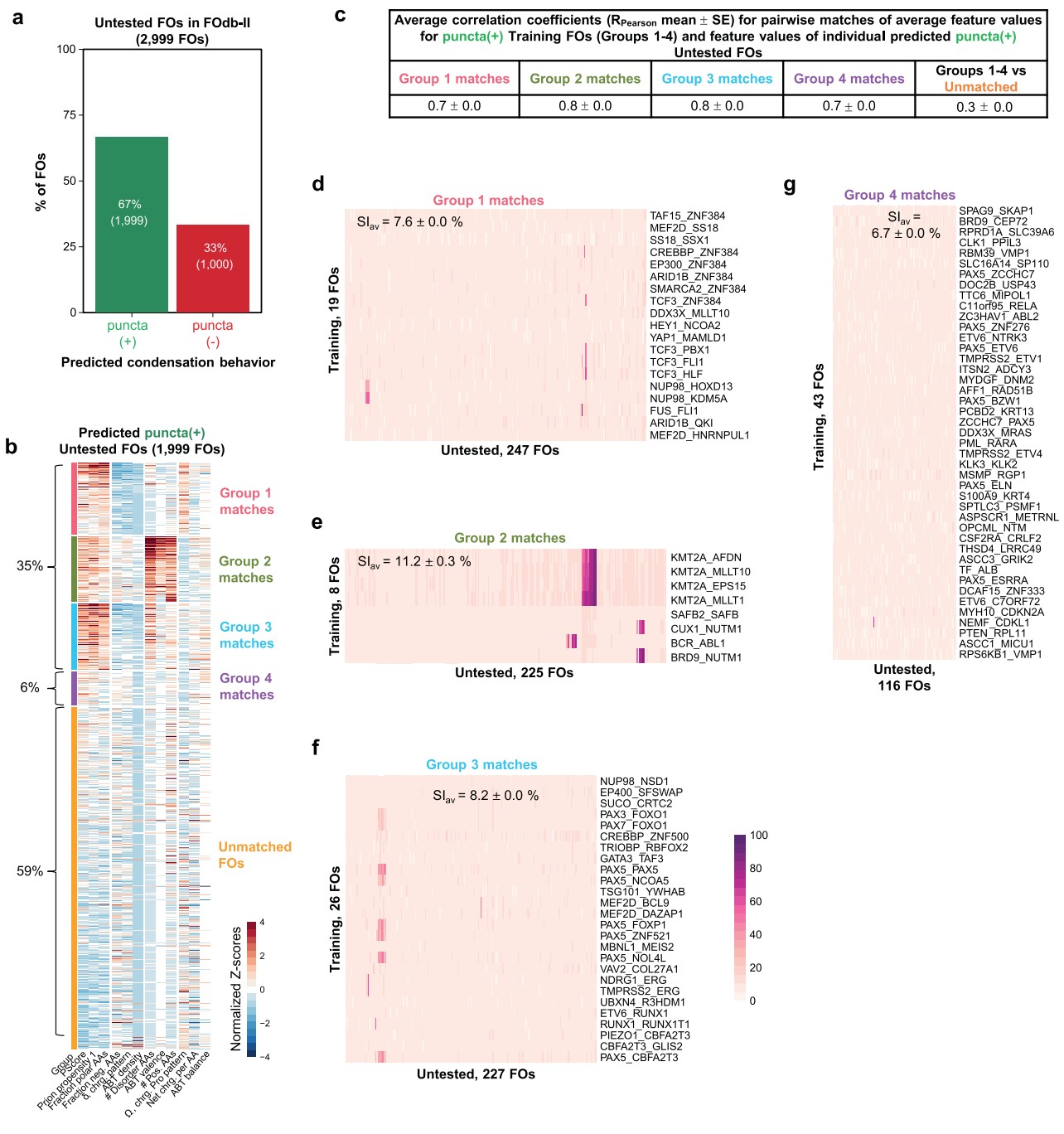

**Fig. 8 | Physicochemical features for predicted puncta(+) FOs. a** Results of predicted condensation behavior using the FO-Puncta ML model for all FOs in FOdb-II excluding the Expressed and Verification FOs (2999 FOs, in total; termed the Untested FOs). **b** Results of comparing the values of 12 physicochemical features (as performed for the Training FOs) for each predicted puncta(+) FO in the Untested FO set to the average feature values of Groups 1–4 of the puncta(+) Training FOs. The Untested FOs were matched to the feature groups with which they had the greatest and most significant ($p \leq 0.05$) pairwise positive correlation and data is presented as a clustered heatmap. 1184 FOs (59%) did not match any of the four feature groups and were placed in a separate group (Unmatched FOs, orange). See Supplementary Dataset 5 for additional information on the physicochemical features used in these analyses. **c** The average Pearson correlation coefficients for the feature group matches displayed in (A). Data is reported as $R_{Pearson}$ mean ± standard error. **d–g** Matrices comparing pairwise amino acid sequence identities between the matched groups (Training FOs versus Untested FOs in Groups 1–4). The average percent identity standard error is given at the top of each matrix. All source data are provided as a Source Data File.

or unenriched in most charge-related features. In contrast, puncta(+) Group 2 FOs are highly enriched in disorder content, ABT valence (a measure of the number of charged tracts), and positively charged residues and only weakly enriched in the features seen in Groups 1 and 3. Almost 90% of the Group 1-3 puncta(+) FOs are localized in the

nucleus, suggesting a role for interactions with nucleic acids, in addition to multivalent interactions between proteins mediated by the physicochemical feature enrichments discussed above, in condensate formation. The features of the Groups 1–3 FOs are generally reminiscent of those discussed for components of the transcriptional

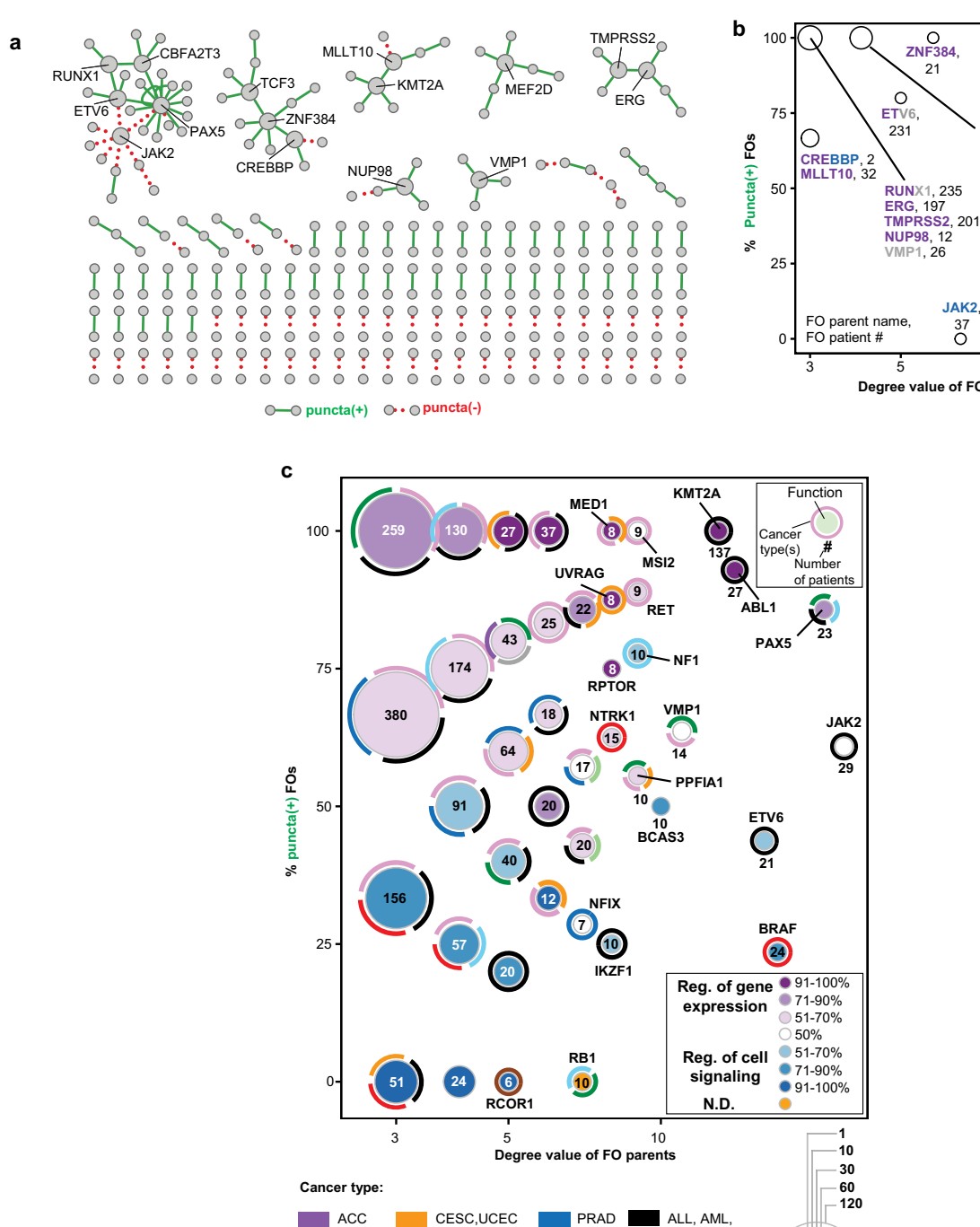

machinery[28,46,47], although the predominance of charges features, seen in some FOs, was more recently pointed out amongst certain transcriptional regulators[29]. It is interesting that features that quantify different properties of acidic and basic tracts (ABT density, valence and balance), developed in studies of nucleolar proteins[33], are amongst the 12 that are most deterministic of puncta(+) or puncta(-) behavior by FOs.

In contrast to FOs in Groups 1–3, those in Group 4 displayed indistinct feature enrichments, which further were indistinguishable from those of puncta(-) FO Group 2'. These findings may indicate that the 12 most discriminatory physicochemical features included in our analyses do not capture the properties associated with their

condensate formation behavior. However, expanding our analyses to 25 features and examination of how they contribute to condensation formation predictions by the FO-Puncta ML model revealed differences between the Group 4 puncta(+) and Group 2' puncta(-) FOs (Supplementary Fig. 11A). Unbiased 2D hierarchical clustering on the basis of SHAP contribution values revealed that differences in the values of up to six features (left feature columns, Supplementary Fig. 11A) naturally led to segregation into the puncta(+) and puncta(-) groups (Group 4 and Group 2', respectively). While it is difficult to discern differences between these two groups on the basis of physicochemical feature values (Supplementary Fig. 11B), the FO-Puncta ML model was able to identify such differences and correctly recapitulate

**Fig. 9 | The FO condensate landscape. a** Cytoscape network analysis of all FO parents (nodes) from the Training and Verification FO sets. Edges indicate a fusion event. Solid green edges reflect puncta(+) and dotted red edges puncta(-) cellular condensation behavior, respectively. **b** Analysis of the condensation behavior of all FO parents from (A) that are involved in ≥ 3 fusion events in our Training and Verification FO sets (degree value ≥ 3). The percent of puncta(+) FOs in which the parent is involved is plotted on the y-axis and the degree value of the parent is on the x-axis. Circles represent FO parents with the same puncta(+) percentage and degree values, and the size of the circle reflects the number of FO parents encompassed by that circle. Numbers following parent names indicate the total patient count for FOs associated with that parent. The parent names are color-coded to indicate the predominant functional associations of the FOs in which each parent is found as analyzed through the Conserved Domain Database (CDD). Purple indicates a predominant association with regulation of gene expression, blue indicates regulation of cell signaling, and gray indicates all other functions. See

Supplementary Dataset 7 for all terms. **c** Analysis of the condensation behavior of all Untested FO parents that are involved in ≥ 3 fusions (degree value ≥ 3). The percentage of puncta(+) FOs in which a parent is involved is plotted on the y-axis and the degree value of the parent is on the x-axis. Circles represent clusters of FO parents with the same puncta(+) percentage and degree values, and the size of a circle reflects the number of FO parents encompassed by that circle. The gradient coloring of circles reflects the dominance of the functional classification of the FOs in which the parents comprising each circle are involved, with 50% indicating that the two functional classes are equally represented. Functional assignment is based on matching FOs to Groups 1–4 of puncta(+) or Group 1'–3' of puncta(-) FOs. Orange circles indicate that FOs did not match any of the Groups 1–4. Circles containing a single parent are labeled with the parent's name. Cancer type abbreviations are defined in Supplementary Dataset 3. All source data are provided as a Source Data File.

FO condensation behavior. Interestingly, the SHAP contribution values that most highly discriminate between Groups 4 and 2' report on charge-related physicochemical features (Supplementary Fig. 11A). These results illustrate the utility of our FO-Puncta ML model and its use of physicochemical features in analysis of the condensation behavior of FOs.

Our analyses of conserved domains within the experimentally tested FOs provide insight into their possible oncogenic mechanisms. Puncta(+) FOs localized in the nucleus (Fig. 7a, c) are highly enriched in functional terms related to regulation of gene expression. Further, these terms are not enriched amongst puncta(-) FOs, very few of which are localized in the nucleus (Fig. 7a–c). Several condensate-forming FOs studied previously have similar functional domains and are known to promote oncogenesis by driving aberrant gene expression[48]. We speculate that this functional mechanism is common to many of the nuclear, puncta(+) FOs we identified in our studies, although definitive testing awaits future investigations.

Regulation of expression of certain genes involves formation of so-called super-enhancers, which involve the compaction of distal DNA regulatory elements and transcriptional regulatory proteins, within condensates that additionally recruit RNA polymerase II[46]. The emergent properties of phase separated condensates, including compaction of biopolymers such as DNA and concentration of multiple protein factors within them, are well-matched with mechanistic aspects of transcriptional regulation and may underlie why many nuclear FOs harboring domains associated with gene regulation form condensates. In contrast, the majority of FOs with functional terms related to regulation of cell signaling did not form condensates in our studies (Fig. 7). However, there are notable examples of condensate-forming FOs that aberrantly regulate cell signaling (reviewed in ref. 48), and we did observe cytoplasmic condensate formation by some FOs enriched in cell signaling-related terms (Fig. 7). However, the enrichment of these functional terms was greatest for FOs that exhibited diffuse localization in the cytoplasm or both the cytoplasm and nucleus (Fig. 7e, f). This apparent functional dichotomy for puncta(+) and puncta(-) FOs was recapitulated through analysis of 2999 additional FOs using physicochemical features and the FO-Puncta ML model (Fig. 9c), with FOs predicted to form condensates most frequently matching the features of Groups 1-3 FOs, which we propose encodes aberrant regulation of gene expression function in the nucleus. In contrast, FOs predicted to not form condensates most frequently matched the features of Groups 1'–3' FOs, which we align with aberrant regulation of cell signaling function in the cytoplasm. These observations, overall, reinforce the long-held idea that FOs fall into one of two general functional classes, those that drive oncogenesis by regulating transcription and others that promote oncogenesis by regulating cell signaling. However, the key insight from our studies is the alignment of condensate formation with the former class and not the latter. These results suggest that the emergent properties of

condensates are not generally required for regulation of cell signaling by FOs, but it is also possible that structures with emergent properties do in fact form and are too small to be detected using the confocal fluorescence microscopy methods employed in our studies.

The discovery of condensate formation by almost 58% of the FOs we tested represents a potential therapeutic vulnerability. Most FOs are comprised of both folded domains and IDRs, which together mediate condensate formation and aberrant biological functions. While small molecules have been shown to interact with specific regions within IDRs and modulate function[49–51], IDR-targeted small molecule drugs have not yet reached the clinic. In contrast, folded domains within condensate-forming FOs are potentially accessible for therapeutic targeting. For example, BCR-ABL, which possesses the constitutively active ABL tyrosine kinase domain and is a driver of chronic myelogenous leukemia[52], tested positive for condensate formation in our studies, suggesting that clinically effective kinase inhibitors such as imatinib[52] might function within condensates. Interestingly, BCR-ABL was previously shown to localize within stress granules; inhibition of kinase activity with imatinib released the FO from these granules[53]. Many of the experimentally tested FOs (Expressed and Verification FOs) contain kinase domains (29 FOs, in total), with eight shown to form condensates. Therefore, it is relevant to consider the accessibility of small molecules to the interior of condensates when seeking to target puncta(+), kinase domain-containing FOs. This issue has been addressed experimentally, with some FDA-approved drugs shown to preferentially partition into certain biomolecular condensates reconstituted in vitro[54,55]. A strategy for the future may be to develop small molecules to target puncta(+), kinase domain-containing FOs by optimizing both condensate partitioning and kinase inhibition. Many other puncta(+) FOs formed condensates in the nucleus and contain domains involved in regulation of gene expression (Fig. 7a), including many DNA-binding domains (Supplementary Dataset 8). Many of these FOs also contain IDRs, creating chimeric transcription factors that drive aberrant gene expression[48]. However, transcription factors have generally been considered undruggable[56], although some have expressed optimism about the potential of proteolysis-targeting chimeric (PROTAC) molecules to target them[57]. An alternative strategy for puncta(+) FOs that are aberrant transcription factors is to target critical interaction partners that have targetable folded domains. An example is the interaction of NUP98 FOs with the Menin-MLL1 complex, which is a molecular dependency in FO-driven pediatric AML[58]. The compound VTP50469 inhibits the Menin-MLL1 interaction, displacing these proteins from chromatin sites also occupied by NUP98 FOs, reversing the leukemogenic gene expression program[58]. This compound acted similarly in MLL FO-driven leukemias, which also depend on interactions with Menin[59]. Both the NUP98[48] and MLL FOs (reported herein) form nuclear condensates but the activity of VTP50469 has not been considered through this lens; perhaps consideration of condensate

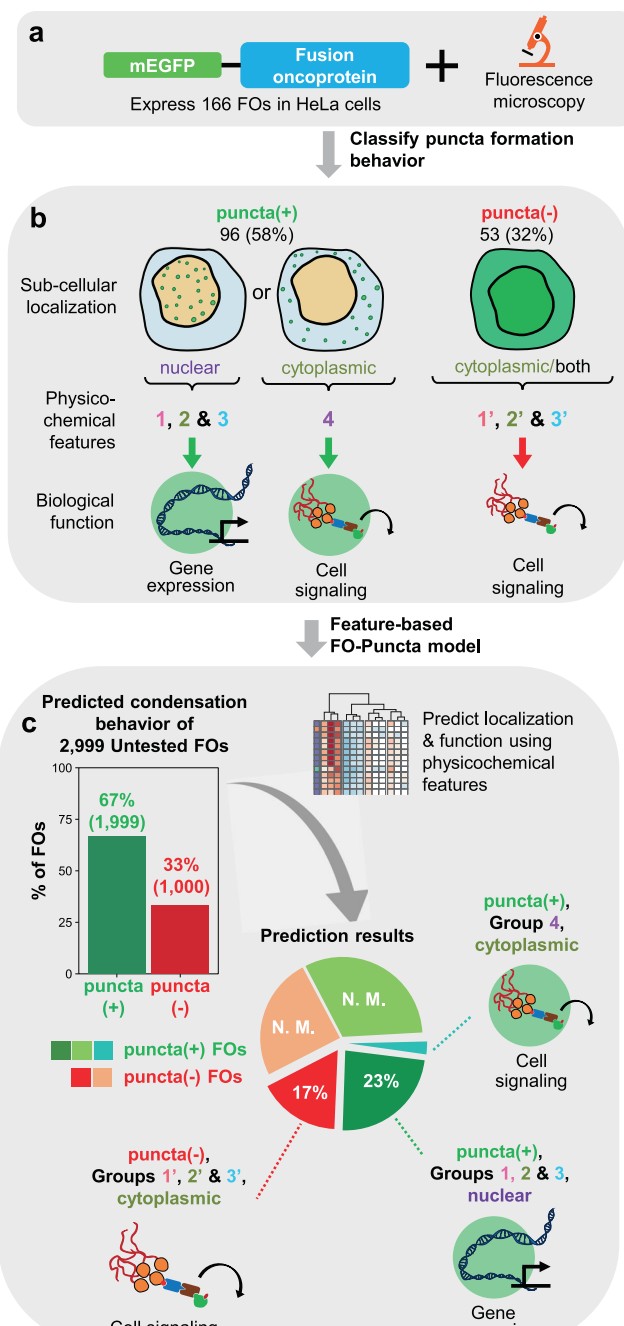

**Fig. 10 | Condensate formation by fusion oncoproteins; experimental workflow and major findings.** Summary schematic of the reported findings on the FO condensate landscape. Cellular imaging (**a**) and 2D-hierarchical clustering (**b**) of puncta(+) and puncta(-) FOs resulted in the identification of FO groups with distinct physicochemical features. The groups further correlated with sub-cellular localization and function. Most nuclear FOs function in the regulation of gene expression and are found in puncta(+) Groups 1–3, while most cytoplasmic FOs function in the regulation of cell signaling and are found in puncta(+) Group 4 and puncta(-) Groups 1'–3'. **c** Application of the FO-Puncta ML model to 2999 Untested FOs resulted in a prediction of 67% puncta(+) and 33% puncta(-) FOs. A portion of these FOs could be matched to the established puncta(+) and puncta(-) groups (23% and 17% of 2999 FOs, respectively) based on their physicochemical feature values, providing insight into their predicted sub-cellular localization and function. Another, larger portion could not be matched to established feature groups (N. M.).

partitioning may guide the development of improved analogs of this compound in the future. In conclusion, because a significant proportion of cancer-driving FOs form condensates in either the nucleus or cytoplasm, or in both compartments, consideration of the influence of the condensate physicochemical environment on small molecule partitioning and interactions may promote drug development against them in the future.

We wish to alert readers to certain considerations while interpreting the results of this study. Firstly, we tested FOs for condensate formation by overexpressing them exclusively in HeLa cells, not in cell types relevant to the diverse cancers associated with the studied FOs. However, we and others previously showed that the pediatric AML-associated NUP98 FOs form very similar nuclear condensates in a non-AML-relevant cell line (HEK293T) and in mouse hematopoietic stem and progenitor cells (HSPCs)[6,8], and that condensate formation was associated with the induction of leukemogenic phenotypes in HSPCs. We reason that our observation of condensate formation by >100 FOs in HeLa cells reflects the true behavior of these proteins driven by their intrinsic, sequence-based physicochemical features, and our results represent a valuable resource for guiding in-depth studies of links between condensate formation and FO-driven oncogenesis in cancer-relevant cell types. Secondly, while we identified statistically significant relationships between the physicochemical features of FOs and their condensate formation behavior, sub-cellular localization and biological function, these relationships are based on studies of a limited number of FOs (166 Expressed and 29 Verification FOs). We used the FO-Puncta ML model and physicochemical feature matching to predict the condensate formation behavior, sub-cellular localization, and biological function of ~3000 additional Untested FOs; such predictions are necessarily subject to some degree of uncertainty. Nonetheless, the general trends regarding the association of regulation of gene expression function with puncta(+) condensation behavior and regulation of cell signaling function with puncta(-) behavior are likely to be generally valid and warrant experimental testing in the future. Another limitation to this work is the infeasibility of directly assessing the functional consequences of condensate formation by the 96 puncta(+) FOs. Importantly, links between condensate formation and function have been established for a number of FOs, some of which were tested in our studies[6–8,10–12,16–19,60]. Also, 34 of our puncta(+) FOs have previously been shown to promote oncogenic cellular phenotypes. Further, we identified conserved domains with definitive functional annotations in the amino acid sequences of a large portion of both puncta(+) and puncta(-) FOs (166 Expressed and 29 Verification FOs; Fig. 7), enabling us to propose their biological functions. These hypotheses are a rich resource for us and others to pursue mechanistic studies into relationships between condensate formation, or not, and oncogenesis driven by FOs in the future.

Additionally, while we identified distinct patterns of physicochemical features associated with puncta(+) behavior by 96 of the 166 Expressed FOs, we have not explored how these patterns are associated with the conformational properties of these proteins or their propensities for multivalent interactions that underlie condensate formation. Nor have we probed interaction partners associated with puncta(+) FOs and their potential roles in condensate formation. FOs previously shown to form condensates (reviewed in ref. 48) have been shown to interact with multiple, additional proteins, but have also been shown to have an intrinsic propensity to form condensates through phase separation. Beyond interaction partners, we hypothesize that puncta(+) FOs displaying different patterns of physicochemical features and amino acid enrichments (e.g., FOs in Groups 1–3; Fig. 4a, c, d) will exhibit different conformational properties and different types of intra- and inter-polypeptide chain interactions that

promote their ability to form condensates. However, testing this hypothesis must await future investigation. In conclusion, while there are some limitations associated with our experimental and computational methods, we argue that our findings represent a valuable resource that defines the condensate landscape of FOs in terms of their physicochemical features, condensate formation behavior, subcellular localization, and biological functions.

## Methods

The research reported in this manuscript was approved by the Institutional Review Board of St. Jude Children's Research Hospital (SJCRH). Genomic data were obtained from samples collected from patients enrolled on studies approved by the SJCRH IRB. Samples were provided deidentified and research was deemed non-human. For tissues, the data are provided from deidentified samples from human subjects, they were enrolled on studies that included genomic analysis or had provision for sample banking and genomic analysis as a secondary use of material. Patients/representatives provided consent/assent on IRB approved protocols consistent with the Declaration of Helsinki.

### Database curation

FO amino acid sequences were created based on RNA and DNA sequencing results obtained from the Cancer Genome Atlas (TCGA) (RRID:SCR_003193) and RNA sequencing (RNA-seq) results from patient samples at St. Jude Children's Research Hospital.

To compose a set of human fusion genes across cancer types using the TCGA database, we retrieved all fusion events from ChimerDB 3.0 (RRID:SCR_007596)[61], which is a comprehensive database of fusion events from analysis of next-generation sequencing data and manual curation. ChimerDB 3.0 contains data from three sources (ChimerKB, ChimerPub and ChimerSeq). As the ChimerSeq module provides more systematic and unbiased information and carries many fusion genes from analyzing deep sequencing RNA-seq data from the TCGA project, we used the ChimerSeq module for our analyses. We started with 46,492 fusion events, with specific break points and cancer types specified for each cancer-associated fusion gene. After filtering redundant entries across cancer types and samples, we obtained 38,000 unique gene fusion events. These gene fusion events were identified by computational analysis of transcriptome sequencing data by TopHat-Fusion (RRID:SCR_011899)[62] and FusionScan[63]. As evidence has demonstrated that TopHat-Fusion's prediction is most reliable, we considered all the gene fusion events identified by TopHat-Fusion. In total, we retrieved 8175 gene fusion events across 2947 tumor samples in 23 cancer types. To get the frequency of fusion genes, we computed the instances of each fusion gene pair (such as "GeneA_GeneB" as an example), involving all possible break points for the same fusion gene pair that occur across cancer types as well as in each cancer type. To further obtain the sequences of fusion oncoproteins, we then filtered our list for "in-frame" fusion genes only, where the fusion event resulted in a fusion protein product [a chimera made of the N-terminal head protein (e.g., parent protein A) joined with C-terminal tail protein (e.g., parent protein B)]. Based on Ensembl 75, we obtained the sequences of each component protein (either head or tail gene) forming each fusion. Information on >4000 fusion oncoproteins identified using TCGA are provided in Supplementary Dataset 1, including head/tail gene names, full-length amino acid sequence, sequence source (e.g., TCGA for these sequences), and the number of patients identified with the FO in different cancer types (when this information was available).

For the St. Jude Children's Research Hospital (SJCRH)-derived fusion oncoprotein database entries, full-length fusion protein sequences were generated via an extension of the frame-checking component of the CICERO gene fusion caller[64], which determines whether the partner genes of the predicted fusions share the same reading frame. During this procedure, RNA fusion contigs assembled from RNA-seq reads spanning the fusion breakpoints were translated into either three putative coding frames or six frames (using Transeq (RRID:SCR_015647)[65] from the emboss suite (v6.6.0, https://www.ebi.ac.uk/Tools/emboss/), protein sequences were computed for each frame, and frames matching both partner genes were identified. Each end of the in-frame protein contig is then mapped to the full-length protein for the corresponding N-terminal or C-terminal partner gene and the longest peptide of the N-terminal of 5′-partner and longest peptide of the C-terminal of the 3′-partner were chosen when multiple RefSeq transcripts were present. Finally, the N-terminal and C-terminal gene protein sequences were joined with the contig protein to form a full-length fusion protein sequence. Reference protein sequences were extracted from the NCBI RefSeq distribution (RRID:SCR_003496)[66] using UCSC (RRID:SCR_005780) table browser. The process considers all potential combinations of isoforms in the RefSeq database[66] mapped to the genomic breakpoints of the fusion transcripts, so predicted full-length products can vary depending on the isoforms. Isoform annotations were documented so that the user may filter to select the most appropriate transcripts if desired; for example, when UTRs were used in the fusion protein, such as ZNF384r fusions, the inserted peptide translated from 5′-UTR of the 3′-partner were manually checked. Information on ~400 fusion oncoproteins identified at SJCRH are provided in Supplementary Dataset 1, including head/tail gene names, full-length amino acid sequence, sequence source (e.g., SJCRH for these sequences), and the number of patients identified with the FO in different cancer types (when this information was available).

### Validating fusion protein transcripts and their prevalence in cancer cohorts

We screened the publicly available tumor RNA-seq data sets to determine the prevalence of the fusion oncoprotein amino acid sequences used in this study, as described above. The adult cancer data set was from 9461 TCGA RNA-seq sets hosted on the Cancer Genomics Cloud (https://datacommons.cancer.gov/analytical-resource/seven-bridges-cancer-genomics-cloud). The pediatric cancer data set includes 921 RNA-seq Datasets from the Pediatric Cancer Genome Project (PCGP)[67] and 1650 RNA-seq Datasets from St. Jude Clinical Genomic data available on the St. Jude Cloud platform[68,69]. We also analyzed 9419 RNA-seq Datasets from GTEx (RRID:SCR_001618) (https://gtexportal.org/home/) which were generated from more than 40 normal tissues as a control (https://gtexportal.org/home/tissueSummaryPage). All GTEx data releases follow the NIH Genomic Data Sharing (GDS) Policy.

To determine the presence of the FO sequences in our TCGA- and St. Jude-derived FO databases in massively parallel sequencing data, we ran Fuzzion2 (https://github.com/stjude/fuzzion2/), a high-performance program for identifying fusions via fuzzy matching of fusion contig sequences to unaligned read pairs. For each targeted FO sequence, Fuzzion2 uses a pattern sequence which describes the fusion junction along with ±500 nt of flanking sequence in the context of the target RNA-seq Dataset. For this project, we developed a method to generate RNA patterns from amino acid sequences by aligning each given full-length protein sequence with its associated refSeq protein pair via BLAST (blastp, RRID:SCR_001010)[70], then using the underlying RNA sequences to create a pattern spanning the fusion breakpoint. Using this method, we were able to generate patterns for ~92% of FO sequences in the full database, and 94% of the subset of patterns evaluated for this manuscript. In some cases, patterns could not be automatically generated, for example when the full-length FO contained a long interstitial amino acid sequence between the regions attributable to the fused gene pair. Other processing exceptions were related to sequence ambiguity in fusions between gene families, internal tandem deletions (ITDs), or fusions involving genes without a coding RefSeq record (e.g., IGH). Because Fuzzion2 is highly sensitive, the possibility exists that it may detect false positive matches attributable to known sequencing artifacts such as barcode hopping. To set

a rigorous threshold, we required matches to have at least 3 "strong" matching read pairs, which are defined as having a match of at least 15 nucleotides to both sides of the junction, or to the breakpoint region. This approach captures supporting evidence for the fusion involving split or discordantly paired reads. The results from the cancer data sets were compared to those of the GTEx normal tissues to remove potential false positives with strong hits in GTEx that can arise due to read-through events, germline polymorphisms or mapping ambiguities caused by paralogous genes. Cancer types and patient counts were based on TCGA and St. Jude tumor sample bar codes to avoid double-counting. Results from this analysis were used to exclude FOs from FOdb-II that were identified to arise from sequencing errors or were found in non-cancerous tissues (Supplementary Dataset 2). Additionally, where evidence could be found in the literature, FOs arising from reciprocal gene fusion events associated with bona fide cancer-driving FOs were removed from FOdb-II (Supplementary Dataset 2).

### Cloning plasmids

All FOs were *Escherichia coli* codon-optimized and synthesized using Integrated DNA Technologies' gBlocks Gene Fragments. Fragments and the destination vector (CL20) were cut with Not1 and Xba1 restriction enzymes and ligated together New England BioLabs' Quick Ligation Kit per the manufacturer instructions. All plasmid sequences were confirmed by whole plasmid sequencing. Mutant FOs (Fig. 6) were synthesized in the same manner.

### Cell culture and transient transfections

HeLa cells (American Type Culture Collection, CCL-2; RRID:CVCL_0030) were cultured in Dulbecco's Modified Eagle Medium (DMEM) with high glucose (Gibco) and supplemented with 1× penicillin/streptomycin (Gibco), 10% fetal bovine serum (Hyclone), and 4 mM L-Glutamine (Gibco) and maintained at 37 °C in 5% $CO_2$. Cells were tested for Mycoplasma every 2 months using PCR (e-Myco™ plus, LiLiF). Cells were authenticated by STR-profiling (PowerPlex Fusion at the St. Jude Hartwell Center). Cells were transfected in a 96 well plate with 100 ng of plasmid DNA in the CL20 vector backbone[8] using FuGENE HD (Promega) per the manufacturer's instructions. All FOs were transfected under identical conditions (same amount of plasmid, same protocol, same incubation period). All Training and Verification FOs were N-terminally tagged with monomeric EGFP (A207K mutation in EGFP). Cells were used for a maximum of 25 passages post thawing.

### Confocal microscopy Imaging

All microscopy images were acquired on a 3i Marianas system (Denver, CO) configured with a Yokogawa CSU-W spinning disk confocal microscope utilizing a 100x Zeiss objective, 405 nm (Hoechst) and 488 nm (mEGFP) laser lines, and Slidebook (RRID:SCR_014300) 6.0 (3i). 3D images of cells were captured as z stacks with 0.2 mm spacing between planes, spanning 12.2 mm in total. Live HeLa cells were imaged at 37 °C in phenol red-free DMEM with high glucose (Gibco) supplemented with 1× penicillin/streptomycin, 10% fetal bovine serum, 4 mM L-Glutamine, and 25 mM HEPES.

### Classification of the FOs as puncta(+), puncta(-), Other, and Nucleolar

After collecting the cell images over multiple replicates (images of at least 50 cells were recorded for each FO, with at least two replicates of each), the total number of transfected cells (using mEGFP fluorescence as readout) were counted. A cell was considered non-expressing if the minimum mEGFP intensity of the expressing region was less than 50 raw intensity units above background. This cut-off was chosen by comparing the minimum mEGFP intensity per pixel (mEGFP-FO expression level) over the puncta(+) FO image Datasets. Amongst the transfected cells, the number of cells that showed circular puncta in

the cytoplasm and/or nucleus was counted. The FOs for which puncta were detected in at least 17% of transfected cells were classified as "puncta(+)", while those with less than 17% puncta-forming cells were classified as "puncta(-)". The threshold of 17% was chosen to give approximately 2/3 puncta(+) FOs and 1/3 puncta(-) FOs. Puncta(+) FOs were further classified as "nuclear", "cytoplasmic", or "both" (localized to both compartments) based on the localization of their puncta, while this classification was done on the basis of mEGFP signal localization for the puncta(-) FOs. For those FOs which were classified as "both", this includes those in which a single cell displayed both nuclear and cytoplasmic localization patterns and those in which some cells expressing the FO displayed nuclear localization, while other cells expressing the same FO displayed cytoplasmic localization. The FOs for which there was significantly higher nucleolar localization (either diffuse or punctate) compared to nuclear or cytoplasmic localization were categorized as "nucleolar". A final category of "other" was used for the FOs which showed non-diffuse mEGFP signal, but the structures formed were not reminiscent of round condensates. Rather, these appeared as clusters or tubular structures. "Nucleolar" and "other" FOs were excluded from analyses involving of physicochemical features. For all imaged FOs and proteins, at least 5 representative image files are provided in raw format on the Biostudies Database[71] with the accession number: S-BIAD863. All files consist of multiple Z stacks ranging at least 10 µm and two channels. C0 files refer to the 405 channel (Hoechst DNA) and C1 files refer to the 488 channel (mEGFP-FO).

### Calculation of amino acid sequence-derived physicochemical features of FOs

To understand why some FOs formed puncta and others did not, we selected physicochemical features that were calculated for each FO amino acid sequence (See Supplementary Dataset 2). The feature, "Urea $\Delta\Delta G$" is the residue-specific water to 1 M urea-group transfer free energy (kcal mol$^{-1}$ M$^{-1}$) predicting urea-dependent cooperative protein unfolding energetics and was calculated based on the method of Auton et al.[72]. The fraction of disordered amino acids was calculated with CIDER[24] based on the method of Campen et al.[73]. "$\delta$, chrg. pattern" defines the average squared deviation from the overall charge symmetry amongst all charged blobs, "Max $\delta$, chrg. pattern" is the maximum value of "$\delta$, chrg. pattern", "$\kappa$, chrg. pattern" describes the extent of charged amino acid mixing in a sequence and is defined as, "$\delta$, chrg. pattern"/ "Max $\delta$, chrg. pattern" ("$\kappa$, chrg. pattern" is normalized between 0 and 1), and "PPII propensity" is a propensity scale for type II polyproline helices; all of the latter features were calculated with CIDER based on the work of Das and Pappu[74]. Low-complexity domain (LCD) length was calculated as the number of residues within low Shannon entropy regions, defined as having a Shannon entropy[75] below 0.78 with a window of 12 residues used, with the cut-off of 0.78 being selected to return results similar to those obtained using the SEG algorithm[76] that is used by the Conserved Domain Database (RRID:SCR_002077)[43] webserver to identify low-complexity regions. The features "# Pos. AAs" (the number of positively charged residues (R/K) in a sequence), "# Neg. AAs" [the number of negatively charged residues (D/E) in a sequence], "Isoelectric point" (the pH at which an amino acid sequence has no net electrostatic charge), "Fraction expnd. AAs" [fraction of residues which are predicted to contribute to chain expansion (E/D/R/K/P)], "Fraction pos. AAs" (fraction of residues which are positively charged in a sequence), "Fraction neg. AAs" (fraction of residues which are negatively charged in a sequence), "Fraction polar AAs" (fraction of polar residues (Q/N/S/T/G/H) in a sequence), "Fraction neutral AAs" (fraction of uncharged residues at neutral pH), "Fraction Pro" (fraction of proline residues), "Seq. Length" (number of amino acids in a sequence), "Mol. Weight" (total molecular weight of the amino acid sequence), "# Neutral AAs" (number of uncharged residues at neutral pH), "Fraction hphobic AAs" [fraction of

hydrophobic residues (A/I/L/V/M)], and "Fraction aromatic AAs" [(fraction of aromatic residues (F/Y/W)) were calculated using CIDER[24]. The feature, "Mean hydrop. 1", is the mean normalized hydropathy determined using the method of Kyte and Doolittle[77] and calculated using CIDER. The feature, "Mean hydrop. 2", is the mean (normalized to 1) hydropathy calculated based on coarse-grained simulations on a large data set of IDRs[78]. The feature, "Prion propensity 1", is the normalized log-likelihood ratio of a sequence being prion-like calculated with PLAAC[23]. The features, "Net chrg. Per AA"[79] is the mean net charge per residue and "Ω, chrg. Pro pattern"[80] describes the patterning of charged and proline residues and was calculated with CIDER. The feature, "# Disorder AAs", is the total number of residues within identified IDRs. For this study, IDRs are defined as regions of at least 60 residues in length in which the 11-residue moving average IUPred2-calculated (RRID:SCR_014632)[81] disorder propensity was > 0.45, with the added tolerance that regions <12 residues apart are joined into single regions including the up-to-11 additional residues. The feature, "# Predicted NLSs", is the number of nuclear localization signals predicted by the algorithm, NLStradamus[82]. The feature, "Seq. dec., chrg. Pattern 3", describes charge patterning in a sequence that can be used to estimate the scaling exponent for an arbitrary amino acid sequence enriched in charged residues and was calculated based on the method of Sawle and Ghosh[83]. The features, "ABT valence" is the sum of the absolute charge of all acidic and basic tracts within disordered regions of at least 60 residues in length, "ABT density" is the ABT valence normalized by the number of residues, which were calculated based on the work of Somjee, Mitrea, and Kriwacki and assesses all amino acids except for the first and last residue in IDRs as defined by IUPred2[33]. The "ABT balance" for a disordered region is the total number of residues within charged tracts within a region divided by the ABT valence for that region. The feature, "Prion propensity 2", is the PAPAprop parameter calculated with the PLAAC implementation of the algorithm, PAPA[84]. The feature, "Fraction charge AAs" (the fraction of charged residues in a sequence) was calculated using CIDER based on the work of Uversky[85]. The feature, "PScore", describes aggregated propensity for pi-pi and pi-cation interactions in a sequence and was calculated based on the algorithm of Vernon, et al.[22]. The features "Hydrop. Pattern 1" and "Hydrop. charge pattern 1" describe hydropathy and hydropathy/charge patterning, respectively, and were calculated based on the hydropathy scale and formalism of Zheng et al.[86]. Similarly, the features "Hydrop. pattern 2" and "Hydrop. charge pattern 2" were calculated based on the formalism of Zheng et al.[86] using the hydropathy scale developed by Dannenhoffer-Lafage and Best[78] derived from simulations to help explain the conformational states of LLPS-prone IDPs.

## Calculation of amino acid enrichment for FO sequences

Amino acid enrichment for each FO sequence was calculated using Eq. 1,

$$AA_{Enrichment} = \frac{100 \times Percent\ composition\ of\ sequence}{Percent\ composition\ off\ olded\ human\ proteome} - 100\%$$

(1)

where "Percent composition of sequence" is the percent composition of a particular amino acid in the sequence being evaluated and "Percent composition of folded human proteome" is the percent composition of a particular amino acid in a database of human protein sequences found in the PDB sourced from Swiss-Prot (RRID:SCR_021164) on 05-08-2022 with sequences smaller than 5 amino acids excluded.

## Analysis of sequence-derived physicochemical features

First, we identified that for the 96 puncta(+) and 53 puncta(-) Expressed FOs (collectively termed the Training FOs; 149 FOs, in total) values of the features "ABT valence", "ABT balance", and "ABT density" were

missing for ~20% of FOs, whereas "PScore" and "PAPAprop" values were missing in <2% of FOs, due to the lack of identified IDRs due to the limitation of the IDR length requirement for the calculation of these features. Therefore, we replaced all the missing values using the non-missing median value of these features from the puncta(+) and puncta(-) Expressed FOs, separately. Next, to identify interdependence of the 39 sequence-based physicochemical features for the 149 Training FOs, we computed mutual information (MI) among these features using *infotheo* (version: 1.2.0) package in R (version: 4.1.0) package. Features that displayed strong mutual dependence with others were filtered to remove the redundant features, which resulted in 25 features with low MI (≤ 0.5). We next performed the two-sided t-test for these 25 features to identify those that showed significant differences between the 96 puncta(+) and 53 puncta(-) FOs using the *rstatix* package (version: 0.7.0) in R. Effect size was calculated using the *effsize* package (version: 0.8.1) (https://zenodo.org/record/196082) in R. In this way, we identified 12 physicochemical features whose values were significantly different between the puncta(+) and puncta(-) groups with *p*-values ≤ 0.05. The values of these 12 features were converted to z-scores using the *scale* function in base R, with respect to the human protein sequences in the PDB sourced from UniprotKB/Swiss-Prot version 2022 01 (see above). We next performed hierarchical clustering based on Manhattan distance and Ward's minimum variance method[87] ("ward.D2"), as implemented in the *pheatmap* package (version: 1.0.12) in R, using the z-scores for the noted 12 features to identify groups of FOs in the puncta(+) and puncta(-) sets with related physicochemical features. To determine the significance of the clustered FO groups from the heatmaps of the 96 puncta(+) and 53 puncta(-) FOs, respectively, we applied the *pvclust* package (version 2.2-0) in R that assesses the uncertainty in hierarchical cluster analysis. Using *pvclust*, Approximately Unbiased (AU) *p*-values were calculated via multiscale bootstrap resampling for each cluster in hierarchical clustering. The AU *p*-value (%) of a cluster is a value between 0 and 100, which indicates the significance of cluster membership[34]. We considered cluster members with AU *p*-values ≥ 90% to be significant.

## Calculation of FO sequence identity

To obtain sequence identity (SI) values between the 166 Expressed FOs and 29 Verification FOs, we performed Multiple Sequence Alignment by submitting amino acid sequences of the FOs in FASTA format to the MUSCLE (MUltiple Sequence Comparison by Log-Expectation, RRID:SCR_011812) server (https://www.ebi.ac.uk/Tools/msa/muscle/)[65]. Average sequence identity (SI_av) values were computed from the values in the percent identity matrix (PIM) for the 166 Expressed FOs and 29 Verification FOs. To obtain SI values between the matching groups of the 1999 predicted puncta(+) FOs and 96 puncta(+) Expressed FOs, we used the MUSCLE server[65]. The MUSCLE server was also used to obtain SI values between the matching groups of the 1000 predicted puncta(-) FOs and 53 puncta(-) Expressed FOs, independently.

## Overlap with proteins known to undergo phase separation

To quantify the overlap of parent proteins of FOs in FOdb (in Fig. 1E) with those proteins known to undergo LLPS, we compiled LLPS-prone proteins from four sources: PhaSepDB 1.0[88], DrLLPS 1.0[89], PhaSePro[90], and LLPSDB[91] (all retrieved October 4, 2020). For PhaSepDB, we removed proteins with a "Source" of "High Throughput", keeping only reviewed proteins. For DrLLPS, we removed proteins with an "LLPS Type" of "Client", keeping regulators and scaffolds. Proteins from all sources were merged by UniProt accession, and any with evidence from at least one source were retained. To represent the human proteome, we obtained all human Swiss-Prot (reviewed) proteins from UniProt release 2022_03[92]. Gene symbols for all three sets (FOdb, known LLPS proteins, and the human proteome) were updated to the latest HGNC (RRID:SCR_002827)[93] gene symbols using unambiguous

aliases where available, and were then compared and plotted using the *eulerr*[94] R package. Fisher's exact test was used to assess set overlap.

## Identification of Conserved Domains within and assignment of biological functions to Training and Verification FOs

We identified functional domains within the puncta(+) and puncta(·) Training and Verification FOs by submitting their amino acid sequences in FASTA format to NCBI's Conserved Domain Database (CDD) search tool (https://www.ncbi.nlm.nih.gov/Structure/bwrpsb/bwrpsb.cgi), as a batch job using default parameters. This tool returned Conserved Domain assignments and definitions. The functional terms presented in Fig. 6 and Supplementary Dataset 6 were manually extracted from Conserved Domain definitions. The definitions for many assigned Conserved Domains were uninformative regarding biological function; in these cases, a function was not assigned. Resampling *p*-values were estimated using 100,000 resamples (with replacement) of equally sized sets of FOs.

## GO term analysis

InterProScan 5.52 (RRID:SCR_005829)[95] was used to identify InterPro 86.0 (RRID:SCR_006695) domains[96] in the FO sequences, using default parameters. Manually curated Gene Ontology (GO) annotations for these domains (InterPro2GO, dated 2021/05/26[97]) were also obtained from InterPro. To address the differing level of detail in GO annotation between different domains, direct ancestor GO slim terms were identified using the *mapslim* function of the GOATOOLS Python library[98] (Supplementary Dataset 9). Resampling *p*-values were estimated using 100,000 resamples (with replacement) of equally sized sets of FOs.

## Fluorescence recovery after photobleaching (FRAP)

FRAP experiments were performed on a Marianas spinning disk confocal microscope with SlideBook 6.0 software (3i, Denver, CO), using 100× oil immersion objective, numerical aperture 1.45. For all the FO puncta, excluding SS18-SSX1, a single punctum was photobleached to ~50% intensity by illuminating the punctum with an appropriate laser power with 75 ms exposure. Fluorescence recovery after photobleaching was measured using the Trackmate plugin for FIJI[99–101]. Original data was exported as a stacked tiff file from the Slidebook format, then reformatted such that each frame was a separate timepoint with the interval set as the average time between acquisitions. A region of interest (ROI) was set around the photobleached area encompassing the area the punctum remained in during the entire acquisition. Puncta were detected using the Laplacian of Gaussian (LoG) detector. For the LoG detector, the estimated object diameter was set as the apparent punctum size pre-photobleach and a quality threshold was set to 0. Initial thresholding was set so only the photobleached puncta were assigned as spots for each frame. The Intersection-over-Union (IoU) overlap tracker was used to link spots together, with a minimum IoU set at 0.3. Intensity values over time were exported by plotting the intensity features of spots, then exporting to a.csv file. For the SS18-SSX1 FO, the condensates were large enough that an ROI within a punctum could be photobleached, rather than an entire punctum. A 0.5 µm diameter circular region of interest (ROI) was used to bleach the center of a single punctum and photobleached to 50% intensity using appropriate laser intensity. Csv files were imported to R version 3.6.0, which was used to make plots and calculate recovery and statistical parameters. Analysis code is available upon request.

## Supervised machine learning for puncta classification

We employed the automatic machine learning (AutoML) tool within the *h2o.ai*[37] (version: 3.36.1.1) package in R to classify the puncta (+) and puncta (·) FOs, and predicting the probability of condensate formation using data for the 149 Training FOs (consisting 96 puncta(+) and 53 puncta(·) FOs from the Expressed FOs). Using the 25 sequence-based physicochemical features with low MI for the 149 Training FOs, we set

*nfolds* = 25 for 25-fold cross validation and generated 220 models from AutoML using otherwise default parameters. A Gradient Boosting Machines (GBM) model with 50 trees performed the best amongst the 220 models tested (termed the FO-Puncta ML model) based on the metrics AUC and AUCPR for the 149 Training FOs, and 25-fold cross validation with 149 Training FOs and 29 Verification FOs. Next, we applied the FO-Puncta ML model to the 29 Verification FOs and determined that a threshold value for the condensate formation probability of 0.83 gave the best performance [based on maximizing the F1 score (Supplementary Dataset 5 and Supplementary Fig. 4E)] for classifying the puncta(+) and puncta(·) FOs in this set. For the phase separation predictors, catGranule[39], DeePhase[40] and FuzDrop[41], performance was evaluated for 178 FOs (combined Training and Verification FO sets; 149 and 29 FOs, respectively). The threshold or cut-off values for each of the phase separation predictors were as follows: 0.75 for catGranule[102], 0.5 for DeePhase[40] and 0.61 for FuzDrop[41]. We used the packages *PRROC* (version 1.3.1)[103] and *MLmetrics* (version 1.1.1) in R to compute the performance metrics for catGranule, DeePhase and FuzDrop.

## SHAP analysis of the FO-Puncta predictor

SHAP (Shapley Additive exPlanations)[38] is a method to explain individual predictions, which is based on game theory optimal Shapley values. We applied this method to determine the contributions of the 25 physicochemical features to FO-Puncta ML model predictions. Specifically, we computed the Shapely values separately for the 149 Training FOs and 29 Verification FOs using the *h2o.predcit_contribution* function in the *h2o.ai* package in *R*.

## Analysis of the 2999 untested FOs

We applied the FO-Puncta ML model to predict the condensate formation behavior of 2999 Patient-derived Untested FOs. Using a the condensate formation probability threshold of 0.83, the FO-Puncta ML model predicted 1999 puncta(+) and 1000 puncta(·) FOs. After this classification was performed, missing physicochemical feature values were replaced using the non-missing median value for the same features for the puncta(+) and puncta(·) Expressed FOs. Next, for each of the 1999 predicted puncta(+) Untested FOs, we compared their z-score values (with respect to the human sequences in the PDB, as discussed above) for the 12 physicochemical features used for classification of the Training FOs with the average z-score values for the 4 distinct feature groups (Groups 1–4) identified for the 96 puncta(+) FOs and computed the Pearson correlation coefficient ($R_{Pearson}$) value and the asymptotic *p*-value using the *rcorr* function from *Hmisc* package (version 4.7.1) in R. A predicted puncta(+) Untested FOs was matched to one of the groups identified for 96 puncta(+) FOs (Groups 1–4) based upon the largest positive value of the pairwise $R_{Pearson}$ coefficient and if the asymptotic *p*-value was ≤0.05. pairwise positive $R_{Pearson}$ value. Further, based upon our findings that most FOs in Groups 1-3 formed condensates in the nucleus and displayed functional terms associated with regulation of gene expression, the predicted puncta(+) Untested FOs that were matched to these feature groups were assigned the same properties. Similarly, the predicted puncta(+) Untested FOs that were matched to Group 4 were assigned cytoplasmic localization and regulation of cell signaling function. A similar feature matching analysis was performed for the predicted puncta(·) Untested FOs, with those that matched the features of Groups 1'–3' assigned cytoplasmic localization and regulation of cell signaling function. We also determined pairwise SI values for the matched, predicted puncta(+) and puncta(·) Untested FOs, and the Training FO groups using the MUSCLE server, as discussed above (e.g., Group 1 Training FOs *versus* matched predicted puncta(+) Untested FOs, etc.).

## Network analysis of FO parent protein condensate landscape

We visualized the condensate landscape of the 178 experimentally tested FOs (combining the 149 Training FOs and 29 Verification FOs)

using the *Analyze Network* option in Cytoscape (version 3.9.1, RRID:SCR_003032)[45]. After generating the FO parent network using Cytoscape, we next identified each of the parent proteins with degree value ≥3 (*e.g.*, those parents observed in ≥3 different FOs), grouped FOs associated with parents with the same degree value and then determine the percentage of them that were experimentally puncta(+). The FO condensate landscape was visualized as a two-dimensional (2D) plot of the percentage of puncta(+) FOs associated with each FO parent with degree value ≥3 *versus* the degree value for each FO parent. The 2D plot was annotated with information about the biological functions (derived from the CDD analysis) of the FOs associated with the high-degree FO parents.

We performed a similar network analysis for the 2999 Untested FOs using Cytoscape and identified the FO parent proteins with degree value ≥3 and the associated FOs. The high-degree FO parents were grouped based on degree value and the percentage of associated FOs predicted to be puncta(+) for each group determined based upon the FO-Puncta ML model predictions. These results we plotted as discussed above for the Training and Verification FOs. The 2D plot was annotated with information on predicted biological function for each FO based upon feature group matches (see section above), and the cancer types and patient numbers associated with FOs, defining the FO condensate landscape.

### SAK web server (sak.stjude.org)

The SAK web server allows the submission of one or more amino acid sequences in Fasta format and provides the following outputs to the user by email: graphical sequence analyses [including sequence complexity, conserved domains, secondary structure and disorder predictions, pi-pi and pi-cation interaction (PScore), PLAAC prion-like domain and net-charge-per-residue (NCPR) analyses, and amino acid distributions within the sequence], bar plots of Z-scores for 12 physicochemical features and amino acid enrichments for the full sequence and all identified IDRs, and, finally, a condensate prediction using the FO-Puncta ML model ("YES" for puncta(+) or "NO" for puncta(-); see Fig. 5) and feature group, if identified [Groups 1, 2, 3 or 4 if puncta(+); Group 1', 2', or 3' if puncta(-); see Figs. 8 and S9]. The SAK server also provides a spreadsheet (SAKreport.xlsx) that includes the full set of parameters used and generated by the FO-Puncta ML model, including physicochemical feature Z-score values and SHAP contribution values. These outputs will enable users to determine whether an FO of interest is likely to form condensates in cells, or not, and to analyze the sequence-based physicochemical features associated with the predicted behavior. Also, if the test FO's sequence-derived features match those of a known group, the SAKreport.xlsx file will include prediction of sub-cellular localization and function (either regulation of gene expression or cell signaling).

### Reporting summary

Further information on research design is available in the Nature Portfolio Reporting Summary linked to this article.

## Data availability

All plasmids generated in this study have been deposited with Addgene (https://www.addgene.org/). No new genomic or RNA sequencing data were generated in the studies reported in this manuscript. Gene fusions were identified using publicly available genomic and tumor RNA sequencing data deposited on: St. Jude Cloud [https://www.stjude.cloud/;[68] some of the leukemia-derived gene fusion data were previously reported in refs. 4,104,105] or The Cancer Genome Atlas (TCGA) (RRID:SCR_003193). The data deposited on St. Jude Cloud were generated through research that was approved by the Institutional Review Board (IRB) of St. Jude Children's Research Hospital (SJCRH). Genomic and RNA sequencing data were obtained from samples collected from patients enrolled on studies approved by the

SJCRH IRB. Patients/representatives provided consent/assent on IRB approved protocols consistent with the Declaration of Helsinki. Samples were provided deidentified and research was deemed non-human. Representative microscopy images have been deposited to the Biostudies Database[71] with the accession number: S-BIAD863. Source data are provided with this paper. This includes all of the data analyses reported in this study. The raw data used in this study are available in the Zenodo database under accession code https://doi.org/10.5281/zenodo.7114870. Source data are provided with this paper.

## Code availability

All original code has been deposited on Zenodo: (https://doi.org/10.5281/zenodo.7114870) and is available for download from the provided link. Any additional information required to reanalyze the data reported in this manuscript is available from the lead contact upon request.

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

## Acknowledgements

We thank Drs. Ines Chen for critical review of the manuscript and Swetha Joswala for technical support. We also thank Drs. J. Paul Taylor and Ben

Sabari for gifting plasmids that served as positive controls for condensate formation in cells. This work is supported by National Institutes of Health grants R35 GM137836 (to N.S.) and R35 GM133658 (to S.Y.), Komen Foundation grants CCR19609287 (to S.Y.) and PDF17483544 (to D.J.M.), an NCI Cancer Center Support Grant (P30 CA021765, to St. Jude Children's Research Hospital), NCI Outstanding Investigator Award R35 CA197695 (to C.G.M.), NCI R01 CA246125 (to R.W.K. and M.M.B.), NCI FusOnC2 U54 CA243124 (to C.G.M. and R.W.K.), NCI R01 CA216391 (to J.Z.), NIGMS F32 GM143847 (to S.D.G.), NCI T32 CA236748 (to D.W.B.), a St. Jude Children's Research Hospital Chromatin Collaborative award (to C.G.M.), a Neoma Boadway Fellowship from St. Jude Children's Research Hospital (to B.C.), a SummerPlus Program Fellowship from Rhodes College (to R.S.), and ALSAC. We are grateful for the support of Core Facilities used in this study by NCI P30 CA021765, including the Cell and Tissue Imaging Center, with technical support from Aaron Taylor, Aaron Pitre, and George Campbell, and the Hartwell Center. N.S. is a CPRIT Scholar in Cancer Research with funding from the Cancer Prevention and Research Institute of Texas (CPRIT) New Investigator Grant RR160021 and Research Grant RP220292. D.J.M. is also supported by National Cancer Institute grant K99 CA240689. This research content is solely the responsibility of the authors and does not necessarily represent the official views of the National Institutes of Health.

## Author contributions

Conceptualization, S.S.Y., M.M.B., C.G.M., J.Z., N.S., and R.W.K.; Methodology, S.T., R.S., B.L., S.M.H.H., Y.L., Q.G., M.N.E., S.V.R., X.Z., and J.B.; Software, S.T., S.D.G., R.S., B.L., S.M.H.H., M.N.E., X.Z., S.V.R., and J.B.; Formal analysis, S.T., D.W.B., B.L., and S.M.H.H.; Investigation, S.T., H.K.S., S.D.G., B.C., D.W.B., D.M.M., and M.R.W.; Resources, C.G.P., I.I., M.N.E., X.Z., S.V.R., D.J.M., D.F.J., S.S.Y., C.G.M., J.Z., and N.S.; Data Curation, S.T., D.W.B., M.N.E., X.Z., S.V.R., and J.Z.; Writing - Original Draft, S.D.G., and R.W.K.; Writing – Review & Editing, S.T., H.K.S., B.C., D.W.B., B.L., B.J.P., M.N.E., D.M.M., D.J.M., D.F.J., S.S.Y., M.M.B., C.G.M., J.Z., and N.S.; Visualization, S.T., H.K.S., S.D.G., B.C., D.W.B., B.L., S.M.H.H., and B.J.P.; Supervision, B.L., S.S.Y., M.M.B., C.G.M., J.Z., N.S., and R.W.K.; Funding acquisition, M.M.B., C.G.M., J.Z., and R.W.K.

## Competing interests

S.D.G. is currently employed by Arrakis Therapeutics but his authorship role occurred while he was employed at St. Jude Children's Research Hospital (SJCRH). I.I. has received honoraria from Amgen and Mission Bio. D.M.M. is currently employed by Dewpoint Therapeutics but her authorship role occurred while she was employed at SJCRH. M.R.W. is currently employed by IDEXX Laboratories, Inc. but his authorship role occurred while he was employed at SJCRH. D.F.J. reports personal fees from Transition Bio outside the submitted work. C.G.M. has received consulting and speaking fees from Illumina and Amgen, and research support from Loxo Oncology, Pfizer and Abbvie. R.W.K. reports personal fees from Dewpoint Therapeutics, GLG Consulting, and New Equilibrium Biosciences outside the submitted work. No disclosures were reported by the other authors.

## Additional information

[1]Department of Structural Biology, St. Jude Children's Research Hospital, Memphis, TN, USA. [2]Rhodes College, Memphis, TN, USA. [3]Center of Excellence for Data-Driven Discovery, Department of Structural Biology, St. Jude Children's Research Hospital, Memphis, TN, USA. [4]Livestrong Cancer Institutes, Department of Oncology, Dell Medical School, The University of Texas at Austin, Austin, TX 78712, USA. [5]Department of Pathology, St. Jude Children's Research Hospital, Memphis, TN, USA. [6]Department of Computational Biology, St. Jude Children's Research Hospital, Memphis, TN, USA. [7]Center for Immunotherapy and Precision Immuno-Oncology, Cleveland Clinic, Cleveland, OH, USA. [8]Lerner Research Institute, Cleveland Clinic, Cleveland, OH, USA. [9]Department of Chemical and Systems Biology, Stanford University School of Medicine, Stanford, CA, USA. [10]Department of Developmental Biology, Stanford University School of Medicine, Stanford, CA, USA. [11]Department of Biomedical Engineering, and Oden Institute for Computational Engineering and Sciences, The University of Texas at Austin, Austin, TX, USA. [12]Department of Epigenetics and Molecular Carcinogenesis, The University of Texas MD Anderson Cancer Center, Houston, TX, USA. [13]Department of Bioinformatics and Computational Biology, The University of Texas MD Anderson Cancer Center, Houston, TX, USA. [14]Program in Quantitative and Computational Biosciences, Baylor College of Medicine, Houston, TX, USA. [15]Department of Microbiology, Immunology and Biochemistry, University of Tennessee Health Sciences Center, Memphis, TN, USA. [16]Present address: Arrakis Therapeutics, 830 Winter St, Waltham, MA 02451, USA. [17]Present address: Washington University School of Medicine, 660 South Euclid Avenue, St. Louis, MO 63110, USA. [18]Present address: Dewpoint Therapeutics, 451 D Street, Suite 104, Boston, MA 02210, USA. [19]Present address: IDEXX Laboratories, Inc., One IDEXX Drive, Westbrook, ME 04092, USA. [20]These authors contributed equally: Swarnendu Tripathi, Hazheen K. Shirnekhi, Scott D. Gorman. [21]These authors jointly supervised this work: S. Stephen Yi, M. Madan Babu, Charles G. Mullighan, Jinghui Zhang, Nidhi Sahni, Richard W. Kriwacki. ✉e-mail: richard.kriwacki@stjude.org

