## [Peer Review File · Nature Communications]

REVIEWER COMMENTS

Reviewer #1 (Remarks to the Author):

Tripathi et al generate a large resource on the ability of fusion oncoproteins (FOs) to form condensates in HeLa cells when overexpressed by transient transfection. While several key examples of FOs have been shown to operate through condensate formation, this current study analyzes 166 FOs using high-throughput imaging assays. This rich dataset enables the authors to investigate sequence features of FOs in general and then of the FOs that do or do not form puncta. The authors use their experimental dataset to train a machine learning algorithm to predict whether the remaining FOs in their database would form puncta in cells.

While transient over-expression in HeLa cells is not the ideal method or cell type to investigate many of these FOs, the breadth of the dataset benefits from all FOs being compared in the same background. The benefits of this dataset outweigh the inherent technical caveat which the authors clearly address in the discussion and in the limitation section. The authors do a good job at highlighting the limitations of their study in the final section of the text. A major caveat that is not addressed in the text is that there is no experimental data addressing the potential function consequence of whether FOs do or do not form condensates. It is interesting enough to focus on the presence or absence of condensates, but whether or not condensate formation has anything to do with the oncogenic mechanism of these FOs is not addressed.

Overall, this will be a useful resource for the condensate community and the cancer/FO community. This paper will be of interest to the Nature Communications audience. I provide a few comments and questions below that once addressed should improve the impact of the study.

Major comments:

1) For a resource of this type to be most useful to the community, the ML algorithm puncta prediction and other useful outputs should be set up as an online resource so that the community can query their favorite fusion protein not found in the database. This will increase the impact of the study and will enable new studies on potentially novel FOs to utilize the work described in this study.

2) Does the ML algorithm or 2D clustering used identify the critical physicochemical features necessary for puncta formation? Could the authors pick examples from each of the 4 cluster groups, generate

mutations that lead to no puncta prediction and test those mutants in HeLa cells? This would serve to validate both the 2d clustering and ML algorithm.

Minor: Page 10, line 9. Please define ABT or any other acronym the first time it is used in the text.

Reviewer #2 (Remarks to the Author):

The study investigates aberrant biomolecular condensates formed by fusion oncoproteins (FOs) undergoing liquid-liquid phase separation, in oncogenesis. This work expands the current knowledge on FO and provides with useful tools and resources for FO further (experimental) characterization in human cancer.

Major comments

Live imaging (Fig.2)

By cell imaging, the authors analyze the ability of mEGFP-tagged FOs to form condensates (detected as puncta) in live cell. The authors should also analyze effect of CL20 plasmid when mEGFP is not tagged to any FO (e.g. EGFP empty vector). In addition, to confirm that the puncta correspond to phase separated condensates the authors could include as a control mEGFP tagged to a known protein mediating phase separation.

The authors should clarify what the "FOs were expressed at variable levels" mean (line 20). Was each plasmid expressing mEGFP-tagged FO transfected at the same amount? Or the plasmid concentration was different depending on the FO?

Fig.2, lines 12-13

The authors should comment on the fact the largest group of FOs exhibit weak feature enrichments. Is perhaps their ability to form condensates linked to their interactors?

In the same context the authors should comment more on the fact the largest groups of FOs forming and not forming pucta, respectively (Fig.3 and Fig.S2 lines 15-18) exhibit similar properties.

FO physiochemical features (Fig.S3)

Perhaps the authors should clarify on the FRAP experiment results and condensate properties. FRAP was used to study the mobility of mEGFP-tagged FOs and showed that FOs have different fluorescent recovery. While cytoplasmic FO forming condensates exhibits low fluorescent recovery (i.e. they are less mobile), nuclear FOs have a higher recovery (i.e. they are more dynamic). This seems in contrast with the fact that some FOs (for instance ETV6-RUNX1) although have very low fluorescent recovery (Fig. S3 A) form liquid-like condensates, with small condensates that coalesce to form larger ones (Fig. S3B).

Fig 4.

Is it contradictory that they see that FOs are enriched in the features that associate with LLPS, but their model is not a good predictor of LLPS-prone proteins but only of the FOs identified in human cancer? This is probably because the majority of the FOs that they analyzed they have weak enrichments (Fig. 3F, 3G)? So, FOs have some characteristics that made them different from any other LLPS protein, which ones? And more importantly it means that these characteristics were not taken into account in the study.

Discussion

I believe that the discussion should be expanded and not only re-stating the results provided or presenting the limitations. Indeed, the authors could elaborate a bit on the functions (in light of their functional FO analysis Fig 5) of these FO-condensates and how they promote oncogenesis by undergoing liquid liquid phase separation (LLPS), speculating on how phase separation is responsible for the FO oncogenicity. In addition, the authors could also comment on the chemical properties of the FO-condensates and maybe do a comparison with other known cellular condensates.

The authors could conclude on the importance of studying the aberrant formation of biomolecular condensates for possible drug development.

Minor comments

It would be clearer if S3 is cited together with S4 instead of separated

Adjust Figure 3, some parts are too small the not readable.

To keep the story linear and guide the reader it would easier if all the functional features are presented before the prediction and Machine learning part (i.e. Figure 5 presented before Figure 4) as indeed is depicted in the Summary schematic of the FO condensate landscape Fig 7 D.

Lines 24-25, complex sentence, to rephrase.

Reviewer #3 (Remarks to the Author):

In this manuscript, authors first generated a complete list of fusion oncoproteins (FOs) and examined the potential of puncta production with 166 of them following overexpression in Hela cells. Features of those puncta(+) and puncta(-) subgroups were then studied, which were used for building a machine learning model to predict condensation by a broad list of FOs. Overall, the manuscript is generally well

written and figures are properly presented. Predicting condensation behaviors of thousands of FOs is also a rather ambitious and exciting task, since the detailed investigation of condensation/LLPS property of a single FO often requires a full set of assays. However, I also feel that this manuscript comes in short at key places.

Fig 1 used the available databases, part of which I believe was published.

The main concern of Fig 2-3 is that, authors only used very limited assays for studying FOs (that is, overexpression and condensate production in Hela cells). Often, assaying puncta formation in this cell line isn't sufficient to demonstrate LLPS. Many described features associated with puncta+ and puncta- FOs are partly same or similar; authors also mentioned several limitations of the useful approaches such as the over-expression levels, cellular contexts (not mimicking those FO-related cancer contexts), availability of cofactors, etc. Evidence strikes to be thin and seems that in vitro assays shall be used as a complementary approach.

It remains unclear how reliable the presented results would be in determining LLPs behavior or true biological scenarios; how useful would the machine learning predictions be to the field?

As the title indicated, the work focuses on defining the landscape of condensates by FOs. To me, a far more important question is, how and whether or not the condensation by FOs would affect biology, which certainly isn't the focus.

Due to these main weaknesses, I would recommend this Resource paper to a more specialized journal.

Please note: Reviewer comments are in **blue text** and author responses in **black text**. Sections from the main text are in **purple text**.

We greatly appreciate the reviewers' time and effort to carefully consider our manuscript. We have endeavored to address all their comments, suggestions and concerns, as detailed in the accompanying point-by-point response. In many cases, we performed new experiments to address reviewer comments and include new data in the revised manuscript. In other cases, we clarify points included in the original manuscript.

Briefly, major points of revision include: 1) clarification of our usage of the term “condensates”; 2) experimental testing of predictions made by our FO-Puncta ML model through mutagenesis; 3) expanded testing of the FO-Puncta ML model through experiments with 17 additional Verification FOs (bringing the total to 29 FOs); 4) development of an open-access, on-line resource (termed SAK) for performing the computational analyses reported in the manuscript, including condensate formation predictions using FO-Puncta ML model; 5) clarification of physicochemical feature differences between puncta(+) Group 4 and puncta(-) Group 2' FOs; 6) addition of both negative (using GFP only) and positive (using literature-reported, condensate-forming proteins) cell imaging controls; 7) strengthening of analyses of functional features of puncta(+) and puncta(-) FOs and links with oncogenic mechanisms; and 8) mention of the biological relevance of our findings in the Discussion section.

We feel that the manuscript is significantly improved through these revisions and hope the reviewers find it suitable for further consideration at *Nature Communications*.

Responses for all Reviewers

1. Author comment: We wish to clarify our usage of the term “condensates”, which we use to describe the dense, round structures that we observed in cells through expression of many GFP-tagged FOs. We regret any confusion we caused by using this term. It was an oversight on our part to not explicitly define our usage of the term. By using this term, we do not intend to imply formation of these structures through the formal process of phase separation. While the appearance of dense, round structures in cells for many FOs is highly suggestive of formation through phase separation, our current data for most of the FOs studied do not support this claim. To strengthen the suggestion that phase separation is operative for some FOs, we performed FRAP experiments, which showed that most of the GFP-FO molecules were mobile within condensates. Also, in several cases, we observed what appear to be condensate fusion events, indicating that the condensates exhibit surface tension. We feel that these data, for a small subset of the tested FOs, gives us liberty to suggest, in general, that the condensates we observe form through phase separation. We note that the physical features and behavior of the FO condensates we observe are similar to those observed by us and others for a variety other FOs, for which rigorous evidence of formation through phase separation was provided (reviewed in Shirnekhi, Chandra, Kriwacki, *Ann. Rev. Cancer Biology*, 2023). Further, while we wish we had the resources to do so, it is simply not feasible to perform the complement of experiments needed to establish phase separation as the mechanism of formation for over 100 FOs with our current manpower and resources. To address this issue, we have clarified our usage of the term “condensates” in several place in the text, as follows.

Introduction, Pg. 5, Lines 14 – 18

“We note that while the observation of FO-induced condensates is suggestive of formation through phase separation, other tests beyond the scope of the current investigations would be needed to establish this assembly mechanism. Therefore, our use of the term “condensates” is agnostic as to formation mechanism.”

and Results, Pg. 8, Lines 10 - 12:

“We reiterate, however, that our observation of condensate formation by a large proportion of the Expressed FOs is insufficient to establish whether formation occurs through phase separation.”

2. Author comment: To enhance the rigor of our verification of the FO-Puncta ML model, we expanded the Verification FO set from 12 to 29 FOs. ML model performance with this expanded Verification FOs set was AUC, 0.73, and Accuracy, 0.79 (Fig. 5B; also see Supplementary Fig. 4), which we feel is more realistic than the previous results with only 12 Verification FOs (e.g., accuracy of 0.92). We note that the FO-Puncta ML model changed slightly due to the expanded Verification FO set; please see Methods for details. This revision has the added benefit of increasing the number of FOs used in the analysis of conserved domains and functional features, which has enhanced the robustness of relationships between FO physicochemical features, subcellular localization and biological function (Fig. 7).

The expanded verification analyses are discussed under Results on page 13, as follows.

“To verify FO-Puncta ML model performance, we applied it to 29 additional FOs (Verification FOs) with low sequence identity with the Expressed FOs (average sequence identity, 7.6% □ 0.0%), obtaining condensate formation probability values that ranged from ~0.01 to 0.99. We experimentally tested the 29 Verification FOs for condensate formation in live cells and observed that 19 displayed puncta(+) and 10 puncta(-) behavior (Supplementary Fig. 4A-D, Supplementary Table 4). Using a probability value of 0.83 as the threshold for predicting puncta(+) behavior, the FO-Puncta ML model correctly predicted condensate formation for 17/19 puncta(+) FOs and diffuse localization for 6/10 puncta(-) FOs (Supplementary Fig. 4E). The performance metrics [AUC, 0.73; accuracy, 0.79 (Fig. 5B; Supplementary Table 6)], were similar to those of our cross-validation testing (Supplementary Table 6).”

The expanded analyses of conserved domains and functional features are presented under Results on pages 17-19 and in Fig. 7; please see the revised manuscript for these sections.

Point-by-point responses to Reviewer Comments

Reviewer #1 (Remarks to the Author):

Tripathi et al generate a large resource on the ability of fusion oncoproteins (FOs) to form condensates in HeLa cells when overexpressed by transient transfection. While several key examples of FOs have been shown to operate through condensate formation, this current study analyzes 166 FOs using high-throughput imaging assays. This rich dataset enables the authors to investigate sequence features of FOs in general and then of the FOs that do or do not form puncta. The authors use their experimental dataset to train a machine learning algorithm to predict whether the remaining FOs in their database would form puncta in cells.

While transient over-expression in HeLa cells is not the ideal method or cell type to investigate many of these FOs, the breadth of the dataset benefits from all FOs being compared in the same background. The benefits of this dataset outweigh the inherent technical caveat which the authors clearly address in the discussion and in the limitation section. The authors do a good job at highlighting the limitations of their study in the final section of the text. A major caveat that is not addressed in the text is that there is no experimental data addressing the potential function consequence of whether FOs do or do not form condensates. It is interesting enough to focus on the presence or absence of condensates, but whether or not condensate formation has anything to do with the oncogenic mechanism of these FOs is not addressed.

Overall, this will be a useful resource for the condensate community and the cancer/FO community. This paper will be of interest to the Nature Communications audience. I provide a few comments and questions below that once addressed should improve the impact of the study.

Author Comment: We appreciate the reviewer's comments regarding the connection between condensate formation by FOs and oncogenic mechanisms, which were also offered by another reviewer. It is very important to understand the functional mechanism(s) underlying the oncogenic functions of FOs. Importantly, we and others previously established rigorous relationships between condensate formation through phase separation by a select group of FOs and oncogenic mechanism (reviewed in Shirnekhi, Chandra, Kriwacki, *Ann. Rev. Cancer Biology*, 2023). For each FO, the work required one or more years of effort by teams of investigators. We wish we could perform these types of studies for >100 puncta(+) FOs, but doing this is beyond what is feasible given our current resources. We hope the reviewers understand this limitation.

However, while large-scale experimental testing is not feasible, we have extensively mined the literature to strengthen links between condensate formation and oncogenic function. While such data were included in the original manuscript, we performed additional literature searches and now report data related to oncogenic phenotypes for 36 puncta-positive FOs (and 6 puncta(-) FOs) in Supplementary Table 4. These data include evidence of cell transformation by an FO, data showing that an FO drives an oncogenic transcriptional program, and/or in vivo data showing that an FO promotes oncogenesis. While these data do not establish that condensate formation by these 36 FOs is causally linked with oncogenesis, they do support the idea that many FOs that promote cancer phenotypes form condensates in cells. We argue that these additional literature-derived data strengthen the biological and cancer relevance of our studies. Furthermore, our analyses of conserved domains (Fig. 7) indicate possible oncogenic mechanisms of some puncta(+) FOs, although further experimentation beyond the scope of the current manuscript would be needed in the future to clarify these indications. We have modified the text, as follows, to address these points.

Results, Pg. 7, Lines 14-17:

“The 166 FOs spanned both adult and pediatric cancers and included 77 of the 110 FOs with a patient count of ≥ 3 and 36 FOs previously demonstrated to drive oncogenic phenotypes in cancer-relevant cell types (Supplementary Table 4).”

Discussion, Pg. 25, Line 5 to Pg. 26, Line 13:

“Our analyses of conserved domains within the experimentally tested FOs provide insight into their possible oncogenic mechanisms. Puncta(+) FOs localized in the nucleus (Fig. 7A, C) are highly enriched in functional terms related to regulation of gene expression. Further, these terms are not enriched amongst puncta(-) FOs, very few of which are localized in the nucleus (Fig. 7A-C). Several condensate-forming FOs studied previously have similar functional domains and are known to promote oncogenesis by driving aberrant gene expression¹. We speculate that this functional mechanism is common to many of the nuclear, puncta(+) FOs we identified in our studies, although definitive testing awaits future investigations.

Regulation of expression of certain genes involves formation of so-called super-enhancers, which involve the compaction of distal DNA regulatory elements and transcriptional regulatory proteins, within condensates that additionally recruit RNA polymerase II². The emergent properties of phase separated condensates, including compaction of biopolymers such as DNA and concentration of multiple protein factors within them, are well-matched with mechanistic aspects of transcriptional regulation and may underlie why many nuclear FOs harboring domains associated with gene regulation form condensates. In contrast, the majority of FOs with functional terms related to regulation of cell signaling did not form condensates in our studies (Fig. 7). However, there are notable examples of condensate-forming FOs that aberrantly regulate cell signaling (reviewed in ref. ¹), and we did observe cytoplasmic condensate formation by some FOs enriched in cell signaling-related terms (Fig. 7). However, the enrichment of these functional terms was greatest for FOs that exhibited diffuse localization in the cytoplasm or both the cytoplasm and nucleus (Fig. 7E, F). This apparent functional dichotomy for puncta(+) and puncta(-) FOs was recapitulated through analysis of 2,999 additional FOs using physicochemical features and the FO-Puncta ML model (Fig. 9C), with FOs predicted to form condensates most frequently matching the features of Groups 1-3 FOs, which we propose encodes aberrant regulation of gene expression function in the nucleus. In contrast, FOs predicted to not form condensates most frequently matched the features of Groups 1'-3' FOs, which we align with aberrant regulation of cell signaling function in the cytoplasm. These observations, overall, reinforce the long-held idea that FOs fall into one of two general functional classes, those that drive oncogenesis by regulating transcription and others that promote oncogenesis by regulating cell signaling. However, the novel insight from our studies is the alignment of condensate formation with the former class and not the latter. These results suggest that the emergent properties of condensates are not generally required for regulation of cell signaling by FOs, but it is also possible that structures with emergent properties do in fact form and are too small to be detected using the confocal fluorescence microscopy methods employed in our studies.”

Pg. 28, Line 22 to Pg. 29, Line 5:

“Another limitation to this work is the infeasibility of directly assessing the functional consequences of condensate formation by the 96 puncta(+) FOs. Importantly, links between condensate formation and function have been established for a number of FOs, some of which were tested in our studies³⁻¹³. Also, 34 of our puncta(+) FOs have previously been shown to promote oncogenic cellular phenotypes. Further, we identified conserved domains with definitive functional annotations in the amino acid sequences of a large portion of both puncta(+) and puncta(-) FOs (166 Expressed and 29 Verification FOs; Fig. 7), enabling us to propose their biological functions. These hypotheses are a rich resource for us and others to pursue mechanistic studies into relationships between condensate formation, or not, and oncogenesis driven by FOs in the future.”

Major comments:

1) For a resource of this type to be most useful to the community, the ML algorithm puncta prediction and other useful outputs should be set up as an online resource so that the community can query their favorite fusion protein not found in the database. This will increase the impact of the study and will enable new studies on potentially novel FOs to utilize the work described in this study.

Author Comment: We thank the reviewer for noting the potential significance of our studies and how to maximize their impact. To this end, we now provide our computational analysis pipeline and FO-Puncta ML model to the community through the SAK online resource (sak.stjude.org). This site allows the submission of one or more amino acid sequences in Fasta format and provides the following outputs to the user by email: graphical sequence analyses (including sequence complexity, conserved domains, secondary structure and disorder predictions, PScore, PLAAC and NCPR analyses, and amino acid distributions within the sequence), bar plots of Z-scores for 12 physicochemical features and amino acid enrichment for the full sequence and all identified IDRs, and, finally, a condensate prediction using the FO-Puncta ML model (“YES” or “NO”; see Fig. 5) and feature group, if identified [Groups 1, 2, 3 or 4 if puncta(+); Group 1’, 2’, or 3’ if puncta(-); see Figs. 8 and S9]. The SAK server also provides spreadsheets that include the full set of parameters used and generated by the FO-Puncta ML model, including physicochemical feature values and SHAP contribution values. These outputs will enable users to determine whether an FO of interest is likely to form condensates in cells, or not, and to analyze the sequence-based features associated with the predicted behavior. Also, if the test FO’s sequence-derived features match those of a known group, the SAKreport.xlsx file will include prediction of sub-cellular localization and function (either regulation of gene expression or cell signaling).

The SAK web server is mentioned in the Results on page 17 as follows.

“The FO sequence analyses discussed above can be performed using the SAK web server available at <https://sak.stjude.org>.”

A description of SAK’s output is now included in the Methods section on pages 45-46.

“SAK web server (sak.stjude.org)

The SAK web server allows the submission of one or more amino acid sequences in Fasta format and provides the following outputs to the user by email: graphical sequence analyses [including sequence complexity, conserved domains, secondary structure and disorder predictions, pi-pi and pi-cation interaction (PScore), PLAAC prion-like domain and net-charge-per-residue (NCPR) analyses, and amino acid distributions within the sequence], bar plots of Z-scores for 12 physicochemical features and amino acid enrichments for the full sequence and all identified IDRs, and, finally, a condensate prediction using the FO-Puncta ML model (“YES” for puncta(+) or “NO” for puncta(-); see Fig. 5) and feature group, if identified [Groups 1, 2, 3 or 4 if puncta(+); Group 1’, 2’, or 3’ if puncta(-); see Figs. 8 and S9]. The SAK server also provides a spreadsheet (SAKreport.xlsx) that includes the full set of parameters used and generated by the FO-Puncta ML model, including physicochemical feature Z-score values and SHAP contribution values. These outputs will enable users to determine whether an FO of interest is likely to form condensates in cells, or not, and to analyze the sequence-based physicochemical features associated with the predicted behavior. Also, if the test FO’s sequence-derived features

match those of a known group, the SAKreport.xlsx file will include prediction of sub-cellular localization and function (either regulation of gene expression or cell signaling).”

2) Does the ML algorithm or 2D clustering used identify the critical physicochemical features necessary for puncta formation? Could the authors pick examples from each of the 4 cluster groups, generate mutations that lead to no puncta prediction and test those mutants in HeLa cells? This would serve to validate both the 2d clustering and ML algorithm.

Author Comment: We thank the reviewer for these important questions. To demonstrate the usefulness of our FO-Puncta ML model, we selected 8 puncta(+) FOs representing all four physicochemical feature groups and performed mutagenesis guided by parameters used and generated by the ML model to switch their condensation behavior from puncta(+) to puncta(-). We expressed the mutant FOs in HeLa cells and determining their condensation formation behavior. The results of these experiments have been added in Figs. 6, S6, S7, and Supplementary Table 7. We appreciate the reviewer’s suggestion and feel that the addition of these new data strengthen the manuscript by demonstrating the robustness of the physicochemical feature-based FO-Puncta ML model.

These new data are included under Results on Pg.15, Line 11 to Page 17, Line 20:

“Using the FO-Puncta ML model to modulate FO condensation behavior

To further explore the predictive utility of our FO-Puncta ML model, we selected eight puncta(+) FOs spanning the four feature groups and performed mutagenesis guided by analysis of changes in ML model parameters to weaken condensate formation behavior. The mutant FO sequences were then tested in HeLa cells for condensate formation. To guide our mutagenesis strategy, we first identified the physicochemical features with the largest SHAP contribution values for the ML model predictions for the eight puncta(+) FOs (Suppl. Supplementary Fig. 6A; Supplementary Table 7). We next analyzed the values for these highly predictive features, and also examined the corresponding amino acid enrichments within IDRs (Suppl. Supplementary Fig. 6B, C; Supplementary Table 7). To weaken condensation behavior, we performed mutagenesis of multiple, enriched residues within IDRs to modulate multivalent interactions¹⁴. After introducing mutations, we reassessed SHAP contribution, physicochemical feature, and amino acid enrichment values, and the condensation probabilities determined by the FO-Puncta ML model, to determine whether the mutations switched the FO-Puncta predictions from puncta(+) to puncta(-). This FO-Puncta ML model-guided mutagenesis process was iterated until the desired switch was achieved and is illustrated below for several FOs.

Analysis for the puncta(+) ML model prediction for SS18-SSX1 revealed that “ABT density”, “ABT balance”, and “Fraction neg. AAs” were the three features with the largest SHAP contribution values (Fig. 6A, top left). The Z-score values of the “ABT density” and “Fraction neg. AAs” features were lower than for the human PDB reference set, while that for “ABT balance” was similar to the reference set (Fig. 6A, bottom left). Further, analysis of amino acid enrichments within IDRs revealed that glutamine and methionine were the most enriched residues (Fig. 6B, left). To reverse the puncta(+) FO-Puncta ML model prediction, we mutated either 50% or 100% of all glutamine and methionine residues to the negatively charged residues, aspartic acid or glutamic acid. This resulted in mutant FO sequences with reduced enrichment of glutamine and methionine, and a reversal of the features, “ABT density”, “ABT balance”, and “Fraction neg. AAs”, to negative SHAP contribution values, corresponding to prediction of puncta(-) behavior (Fig. 6A, B). These changes led to a decrease in the FO-Puncta

ML model prediction from 0.99 for the unmutated SS18-SSX1 FO sequence to 0.19 and 0.07, respectively, for the two mutants (Fig. 6C; Supplementary Table 7).

Similarly, analysis of SHAP contribution values for NUP98-HOXD13 revealed that the features, “ABT density”, “Fraction neg. AAs”, and “Net chrg. per AA”, were three of the four largest positive contributors to its puncta(+) prediction (Fig. 6A, top middle). The features, “ABT density” and “Fraction neg. AAs”, had negative Z-score values for this FO, while that for “Net chrg. per AA” was slightly positive (Fig. 6A, bottom middle). Analysis of IDRs in this FO revealed an enrichment of threonine, phenylalanine, and glycine residues (Fig. 6B, middle). We therefore mutated all phenylalanine and glycine residues to negatively charged residues, aspartic acid or glutamic acid, which caused the SHAP contribution values for the three noted features to become negative (Fig. 6A, top middle). Acidic residues were introduced in part because they were depleted on average in puncta(+) and enriched in puncta(-) FOs, respectively (Fig. 3C). The mutations caused the FO-Puncta condensation probability to decrease from 1 to 0.08 for the mutated sequence (Fig. 6C; Supplementary Table 7). For the DOC2B-USP43 FO, analysis of SHAP contribution values revealed that the feature, “Net chrg. per AA”, was the largest positive contributor, and the additional features, “ABT balance”, “ABT valence”, and “Fraction pos. AAs” were also positive SHAP contributions (Fig. 6A, top right). The Z-score values of these features were all positive for the DOC2B-USP43 FO sequence (Fig. 6A, bottom right). The IDRs within this FO are enriched in proline and arginine residues (Fig. 6B, right). To reduce enrichment of positive charge, we mutated all of the positive arginine residues to neutral alanine residues. This resulted in an FO-Puncta ML model condensation probability decrease from 1 for the unmutated sequence to 0.75 for the mutant (Fig. 6C; Supplementary Table 7). A similar analytical process was applied to mutate five additional FOs, as described in Supplementary Table 7. The FO sequence analyses discussed above can be performed using the SAK web server available at <https://sak.stjude.org>.

Testing of the condensate formation behavior of the 11 mutant FO sequences in HeLa cells showed that seven were experimentally determined to be puncta(-) (Fig. 6C). The other mutated FOs (SLC16A14-SP110, PAX7-FOXO1, and SS18-SSX1) displayed condensates but in a smaller percentage of cells than the unmutated sequences (Fig. 6C). Highlighting the impact of the ML model-guided sequence modifications, the morphology of condensates formed by SS18-SSX1 were dramatically altered by the introduced mutations (Supplementary Fig. 7A). These results illustrate how our FO-Puncta ML model, together with analysis of SHAP contribution, physicochemical feature, and amino acid enrichment values, can be used to inform mutagenesis of FOs to reduce their condensate formation behavior.”

Minor: Page 10, line 9. Please define ABT or any other acronym the first time it is used in the text.

Author Comment: We apologize for this oversight and have now defined ABT in the text on Page 9, which is when the acronym is first used and in the abbreviations list at the beginning of the manuscript. ABT is an acronym for “Acidic/Basic Tract” used in a previous manuscript¹⁵. We have also confirmed that all other acronyms have similarly been defined.

Results, Pg. 10, Lines 3 – 8:

“Group 2, comprised of 8 FOs (Fig. 4C), showed enrichment of disorder and charge-related features [# Disorder AAs, Acidic/Basic Tract (ABT) valence, # Positive (Pos.) AAs] and weak enrichment of two additional features, PScore and Prion propensity 1 (Fig. 4C, Group 2).”

Reviewer #2 (Remarks to the Author):

The study investigates aberrant biomolecular condensates formed by fusion oncoproteins (FOs) undergoing liquid-liquid phase separation, in oncogenesis. This work expands the current knowledge on FO and provides with useful tools and resources for FO further (experimental) characterization in human cancer.

Major comments

Live imaging (Fig.2)

By cell imaging, the authors analyze the ability of mEGFP-tagged FOs to form condensates (detected as puncta) in live cell. The authors should also analyze effect of CL20 plasmid when mEGFP is not tagged to any FO (e.g. EGFP empty vector). In addition, to confirm that the puncta correspond to phase separated condensates the authors could include as a control mEGFP tagged to a known protein mediating phase separation.

Author Comment: We thank the reviewer for this important comment and apologize for our oversight. mEGFP empty vector controls were performed every time a new set of FOs were imaged, but we failed to include a representative image of this negative control in the figures. We have now added representative control images to Fig. 2G. We have also included this information in the Methods section under the sub-heading of “Classification of the FOs as puncta(+), puncta(-), Other, and Nucleolar”. Regarding the reviewer’s second comment, we have performed live cell imaging of HeLa cells expressing a protein that forms condensates in both the cytoplasm and nucleus [mEGFP-NUP98-N (Chandra, *et al.*, *Cancer Discovery*, 2022)], and three proteins that form condensates in the nucleus (mCherry-TDP43, mCherry-EWSR1-IDR, and mCherry-DDX4-IDR). These proteins have previously been shown by us and others to form condensates in cells. These data are included in Supplementary Fig. 1C. The main text has been modified to discuss these data:

Results, Pg. 8, Lines 8-15:

“Their morphologies were similar to those of several proteins shown previously to form condensates through phase separation and used here as positive controls ^{3,7,16-21} (Supplementary Fig. 1C). We reiterate, however, that our observation of condensate formation by a large proportion of the Expressed FOs is insufficient to establish whether formation occurs through phase separation. 53 FOs did not form puncta (termed puncta(-) FOs) and exhibited varied sub-cellular localization (Fig. 2B, right panel; Supplementary Table 4) and a diffuse fluorescence appearance consistent with an mEGFP empty vector negative control (Figs. 2F, 2G).”

The authors should clarify what the “FOs were expressed at variable levels” mean (line 20). Was each plasmid expressing mEGFP-tagged FO transfected at the same amount? Or the plasmid concentration was different depending on the FO?

Author Comment: We thank the reviewer for this comment. All FOs were transfected at the same plasmid concentration and using the exact same protocol/conditions. However, due to natural variability in transient transfections, we routinely saw cells in a single well that expressed

varying amounts of the mEGFP-tagged FO. We have clarified this in the text and in the Methods section under the sub-heading “Cell culture and transient transfections”.

Results, Pg. 7, Lines 18-23:

“Live HeLa cells were transfected with mEGFP-tagged FOs under identical conditions (e.g., protocol, DNA concentration, number of cells) and imaged 24 hours post-transfection to test for condensate formation. Due to the nature of transient transfections, the mEGFP-tagged FOs were expressed at variable levels, both for a given FO within a cell population and between different FOs, and we recorded fluorescence microscopy images of □ 50 cells for each FO to broadly sample their expression profiles.”

Fig. 2, lines 12-13

The authors should comment on the fact the largest group of FOs exhibit weak feature enrichments. Is perhaps their ability to form condensates linked to their interactors?

In the same context the authors should comment more on the fact the largest groups of FOs forming and not forming puncta, respectively (Fig.3 and Fig.S2 lines 15-18) exhibit similar properties.

Author Comment: We thank the reviewer for this comment. While we cannot rule out that interaction partners influence the condensation behavior of the FOs we tested, past studies from us and others have shown through in vitro and cellular studies that some FOs have an intrinsic ability to form condensates (reviewed in Shirnekhi, *et al.*, Shirnekhi, Chandra, Kriwacki, *Ann. Rev. Cancer Biology*, 2023). While by qualitative inspection it is difficult to identify distinguishable feature patterns between the puncta(+) Group 4 and puncta(-) Group 2' FOs, our FO-Puncta Machine Learning (ML) model is able to do so. The physicochemical feature heatmaps shown in Fig. 4 and Supplementary Fig. 2 display Z-scores for only the 12 most discriminatory features, but the ML model used 25 features in making predictions. Examination of the weight of the contribution of each feature to the ML predictions for the two groups, as reported by SHAP contribution values, revealed feature differences between the positive (Group 4) and negative (Group 2') FOs and results in unbiased, separate clustering of the two groups. We have included these data as a new supplemental figure, Supplementary Fig. 11 and modified the text to include this information. These additions illustrate the basis for the ML model's ability to discriminate between these two groups. We have modified the text to address these points, as follows.

Discussion, Pg. 24, Line 14 to Pg. 25, Line 4:

“In contrast to FOs in Groups 1-3, those in Group 4 displayed indistinct feature enrichments, which further were indistinguishable from those of puncta(-) FO Group 2'. These findings may indicate that the 12 most discriminatory physicochemical features included in our analyses do not capture the properties associated with their condensate formation behavior. However, expanding our analyses to 25 features and examination of how they contribute to condensation formation predictions by the FO-Puncta ML model revealed differences between the Group 4 puncta(+) and Group 2' puncta(-) FOs (Supplementary Fig. 11A). Unbiased 2D hierarchical clustering on the basis of SHAP contribution values revealed that differences in the values of up to six features (left feature columns, Supplementary Fig. 11A) naturally led to segregation into

the puncta(+) and puncta(-) groups (Group 4 and Group 2', respectively). While it is difficult to discern differences between these two groups on the basis of physicochemical feature values (Supplementary Fig. 11B), the FO-Puncta ML model was able to identify such differences and correctly recapitulate FO condensation behavior. Interestingly, the SHAP contribution values that most highly discriminate between Groups 4 and 2' report on charge-related physicochemical features (Supplementary Fig. 11A). These results illustrate the utility of our FO-Puncta ML model and its use of physicochemical features in analysis of the condensation behavior of FOs."

Discussion, Pg. 29, Lines 6 - 17:

"Additionally, while we identified distinct patterns of physicochemical features associated with puncta(+) behavior by 96 of the 166 Expressed FOs, we have not explored how these patterns are associated with the conformational properties of these proteins or their propensities for multivalent interactions that underlie condensate formation. Nor have we probed interaction partners associated with puncta(+) FOs and their potential roles in condensate formation. FOs previously shown to form condensates (reviewed in ref. 49) have been shown to interact with multiple, additional proteins, but have also been shown to have an intrinsic propensity to form condensates through phase separation. Beyond interaction partners, we hypothesize that puncta(+) FOs displaying different patterns of physicochemical features and amino acid enrichments (e.g., FOs in Groups 1-3; Fig. 4A, C, D) will exhibit different conformational properties and different types of intra- and inter-polypeptide chain interactions that promote their ability to form condensates. However, testing this hypothesis must await future investigation."

FO physiochemical features (Fig.S3)

Perhaps the authors should clarify on the FRAP experiment results and condensate properties. FRAP was used to study the mobility of mEGFP-tagged FOs and showed that FOs have different fluorescent recovery. While cytoplasmic FO forming condensates exhibits low fluorescent recovery (i.e. they are less mobile), nuclear FOs have a higher recovery (i.e. there are more dynamic). This seems in contrast with the fact that some FOs (for instance ETV6-RUNX1) although have very low fluorescent recovery (Supplementary Fig. 3 A) form liquid-like condensates, with small condensates that coalesce to form larger ones (Supplementary Fig. 3B).

Author Comment: We appreciate the reviewer's attention to detail in noting these seemingly contradictory data. We repeated FRAP and fusion analyses with ETV6-RUNX1 and confirmed that the original results are true. While this FO on average exhibits very limited recovery after bleaching, some of the condensates it forms are still able to fuse, suggesting that the condensates have liquid-like properties. Due to the small size of these puncta, we performed the FRAP experiments by photobleaching entire puncta. The recovery in these instances depends on how fast the protein molecules diffuse into the condensates from the surrounding light phase (outside condensates). Although molecules within the dense phase can be mobile, which can lead to fusion events, there are several reasons why the diffusion of the light phase protein molecules into the puncta can be slow. For example, factors such as the ratio of fluorescent material inside and outside of puncta (the partition coefficient) and interface resistance, which imposes a barrier for molecules to enter or exit condensates based on conformational requirements, can affect the rate of recovery after photobleaching and are not reflective of the mobility of molecules inside of condensates²²⁻²⁵. We appreciate the reviewer's

thoughtfulness in raising this point; in the interest of general readability, we have elected to not discuss these details in the text of the manuscript.

Fig 4.

Is it contradictory that they see that FOs are enriched in the features that associate with LLPS, but their model is not a good predictor of LLPS-prone proteins but only of the FOs identified in human cancer? This is probably because the majority of the FOs that they analyzed they have weak enrichments (Fig. 3F, 3G)? So, FOs have some characteristics that made them different from any other LLPS protein, which ones? And more importantly it means that these characteristics were not taken into account in the study.

Author Comment: We thank the reviewer for these questions. We first want to clarify that our FO-Puncta ML model was not applied to other, non-FO proteins. Rather, we tested the performance of other published phase separation predictors with our set of experimentally tested FOs. The results show that the other predictors perform less well than our FO-Puncta ML model in predicting the condensation behavior of the 178 Training and Verification FOs (Fig. 5D). The other phase separation predictors were developed using data for large numbers of proteins from different species based upon phase separation data obtained using varied experimental approaches both in vitro and in cells. Therefore, these data are highly heterogeneous. In contrast, the data on condensate formation by FOs used to develop our FO-Puncta ML model were obtained under highly standardized conditions. FOs are a special protein family due to their origins through selection via the process of oncogenesis. They need to promote cell proliferation in order to be selected and, therefore, need to retain certain types of functional domains (often those associated with regulation of gene expression or cell signaling). These specialized functional features are reflected in the physicochemical features of the FOs, and provide a basis for distinguishing between puncta(+) and puncta(-) FOs using the FO-Puncta ML model. In general, the proteins used to train the other phase separation predictors we tested do not exhibit the specialized features of FOs and, therefore, performed less well than our ML model in predicting the condensation behavior of the tested FOs. We reviewed the text in the original manuscript describing these results and, respectfully, feel that it accurately reports our findings. We note that we have done our best to avoid being judgmental regarding the other predictors we tested. They were developed using different sets of proteins and methods, and perform less well than our FO-Puncta ML model when tasked with predicting the phase separation of FOs.

Discussion

I believe that the discussion should be expanded and not only re-stating the results provided or presenting the limitations. Indeed, the authors could elaborate a bit on the functions (in light of their functional FO analysis Fig 5) of these FO-condensates and how they promote oncogenesis by undergoing liquid liquid phase separation (LLPS), speculating on how phase separation is responsible for the FO oncogenicity. In addition, the authors could also comment on the chemical properties of the FO- condensates and maybe do a comparison with other known cellular condensates.

Author Comment: We thank the reviewer for these suggestions. Regarding the chemical properties of the puncta(+) FO condensates and relating these to other known cellular condensates, we have now expanded the discussion to include additional insights from our

physicochemical feature analysis and how these inform on the types of multivalent interactions that might lead to condensate formation. The text was modified as follows:

Discussion, Pg. 23, Line 20 to Pg. 25, Line 4:

“The physicochemical features of the experimentally tested, puncta(+) FOs are diverse, indicating that multiple mechanisms of multivalent interactions are operative during condensate formation. However, our analyses did reveal certain patterns of feature enrichment. For example, puncta(+) FOs are enriched in most charge-related features, with the exception being depletion in negative charge. Additionally, they are enriched in prion domain-like features and score highly for pi-pi and pi-cation interactions. Interestingly, these enrichments are not distributed uniformly amongst puncta(+) FOs, with Groups 1 and 3 enriched in the latter features and generally depleted or unenriched in most charge-related features. In contrast, puncta(+) Group 2 FOs are highly enriched in disorder content, ABT valence (a measure of the number of charged tracts), and positively charged residues and only weakly enriched in the features seen in Groups 1 and 3. Almost 90% of the Group 1-3 puncta(+) FOs are localized in the nucleus, suggesting a role for interactions with nucleic acids, in addition to multivalent interactions between proteins mediated by the physicochemical feature enrichments discussed above, in condensate formation. The features of the Group 1-3 FOs are generally reminiscent of those discussed for components of the transcriptional machinery^{2,26,27}, although the predominance of charge features, seen in some FOs, was more recently pointed out amongst certain transcriptional regulators²⁸. It is interesting that features that quantify different properties of acidic and basic tracts (ABT density, valence and balance), developed in studies of nucleolar proteins¹⁵, are amongst the 12 that are most deterministic of puncta(+) or puncta(-) behavior by FOs.

In contrast to FOs in Groups 1-3, those in Group 4 displayed indistinct feature enrichments, which further were indistinguishable from those of puncta(-) FO Group 2'. These findings may indicate that the 12 most discriminatory physicochemical features included in our analyses do not capture the properties associated with their condensate formation behavior. However, expanding our analyses to 25 features and examination of how they contribute to condensation formation predictions by the FO-Puncta ML model revealed differences between the Group 4 puncta(+) and Group 2' puncta(-) FOs (Supplementary Fig. 11A). Unbiased 2D hierarchical clustering on the basis of SHAP contribution values revealed that differences in the values of up to six features (left feature columns, Supplementary Fig. 11A) naturally led to segregation into the puncta(+) and puncta(-) groups (Group 4 and Group 2', respectively). While it is difficult to discern differences between these two groups on the basis of physicochemical feature values (Supplementary Fig. 11B), the FO-Puncta ML model was able to identify such differences and correctly recapitulate FO condensation behavior. Interestingly, the SHAP contribution values that most highly discriminate between Groups 4 and 2' report on charge-related physicochemical features (Supplementary Fig. 11A). These results illustrate the utility of our FO-Puncta ML model and its use of physicochemical features in analysis of the condensation behavior of FOs.”

We addressed the first point about the function of FO condensates in our reply to Reviewer 1, above, which is repeated here:

Author Comment: We appreciate the reviewer's comments regarding the connection between condensate formation by FOs and oncogenic mechanisms, which were also offered by another reviewer. It is very important to understand the functional mechanism(s) underlying the

oncogenic functions of FOs. Importantly, we and others previously established rigorous relationships between condensate formation through phase separation by a select group of FOs and oncogenic mechanism (reviewed in Shirnekhi, Chandra, Kriwacki, *Ann. Rev. Cancer Biology*, 2023). For each FO, the work required one or more years of effort by teams of investigators. We wish we could perform these types of studies for >100 puncta(+) FOs, but doing this is beyond what is feasible given our current resources. We hope the reviewers understand this limitation.

However, while large-scale experimental testing is not feasible, we have extensively mined the literature to strengthen links between condensate formation and oncogenic function. While such data were included in the original manuscript, we performed additional literature searches and now report data related to oncogenic phenotypes for 36 puncta-positive FOs (and 6 puncta(-) FOs) in Supplementary Table 4. These data include evidence of cell transformation by an FO, data showing that an FO drives an oncogenic transcriptional program, and/or in vivo data showing that an FO promotes oncogenesis. While these data do not establish that condensate formation by these 36 FOs is causally linked with oncogenesis, they do support the idea that many FOs that promote cancer phenotypes form condensates in cells. We argue that these additional literature-derived data strengthen the biological and cancer relevance of our studies. Furthermore, our analyses of conserved domains (Fig. 7) indicate possible oncogenic mechanisms of some puncta(+) FOs, although further experimentation beyond the scope of the current manuscript would be needed in the future to clarify these indications. We have modified the text, as follows, to address these points.

Results, Pg. 7, Lines 14-17:

“The 166 FOs spanned both adult and pediatric cancers and included 77 of the 110 FOs with a patient count of ≥ 3 and 36 FOs previously demonstrated to drive oncogenic phenotypes in cancer-relevant cell types (Supplementary Table 4).”

Discussion, Pg. 25, Line 5 to Pg. 26, Line 3:

“Our analyses of conserved domains within the experimentally tested FOs provide insight into their possible oncogenic mechanisms. Puncta(+) FOs localized in the nucleus (Fig. 7A, C) are highly enriched in functional terms related to regulation of gene expression. Further, these terms are not enriched amongst puncta(-) FOs, very few of which are localized in the nucleus (Fig. 7A-C). Several condensate-forming FOs studied previously have similar functional domains and are known to promote oncogenesis by driving aberrant gene expression¹. We speculate that this functional mechanism is common to many of the nuclear, puncta(+) FOs we identified in our studies, although definitive testing awaits future investigations.

Regulation of expression of certain genes involves formation of so-called super-enhancers, which involve the compaction of distal DNA regulatory elements and transcriptional regulatory proteins, within condensates that additionally recruit RNA polymerase II². The emergent properties of phase separated condensates, including compaction of biopolymers such as DNA and concentration of multiple protein factors within them, are well-matched with mechanistic aspects of transcriptional regulation and may underlie why many nuclear FOs harboring domains associated with gene regulation form condensates. In contrast, the majority of FOs with functional terms related to regulation of cell signaling did not form condensates in our studies (Fig. 7). However, there are notable examples of condensate-forming FOs that aberrantly regulate cell signaling (reviewed in ref. ¹), and we did observe cytoplasmic condensate formation by some FOs enriched in cell signaling-related terms (Fig. 7). However,

the enrichment of these functional terms was greatest for FOs that exhibited diffuse localization in the cytoplasm or both the cytoplasm and nucleus (Fig. 7E, F). This apparent functional dichotomy for puncta(+) and puncta(-) FOs was recapitulated through analysis of 2,999 additional FOs using physicochemical features and the FO-Puncta ML model (Fig. 9C), with FOs predicted to form condensates most frequently matching the features of Groups 1-3 FOs, which we propose encodes aberrant regulation of gene expression function in the nucleus. In contrast, FOs predicted to not form condensates most frequently matched the features of Groups 1'-3' FOs, which we align with aberrant regulation of cell signaling function in the cytoplasm. These observations, overall, reinforce the long-held idea that FOs fall into one of two general functional classes, those that drive oncogenesis by regulating transcription and others that promote oncogenesis by regulating cell signaling. However, the novel insight from our studies is the alignment of condensate formation with the former class and not the latter. These results suggest that the emergent properties of condensates are not generally required for regulation of cell signaling by FOs, but it is also possible that structures with emergent properties do in fact form and are too small to be detected using the confocal fluorescence microscopy methods employed in our studies.”

Discussion, Pg. 28, Line 22 to Pg. 29, Line 5:

“Another limitation to this work is the infeasibility of directly assessing the functional consequences of condensate formation by the 96 puncta(+) FOs. Importantly, links between condensate formation and function have been established for a number of FOs, some of which were tested in our studies³⁻¹³. Also, 34 of our puncta(+) FOs have previously been shown to promote oncogenic cellular phenotypes. Further, we identified conserved domains with definitive functional annotations in the amino acid sequences of a large portion of both puncta(+) and puncta(-) FOs (166 Expressed and 29 Verification FOs; Fig. 7), enabling us to propose their biological functions. These hypotheses are a rich resource for us and others to pursue mechanistic studies into relationships between condensate formation, or not, and oncogenesis driven by FOs in the future.”

The authors could conclude on the importance of studying the aberrant formation of biomolecular condensates for possible drug development.

Author Comment: We appreciate the reviewer’s enthusiasm for the impact of this work and have expanded the discussion to emphasize the relevance of condensate formation in pharmaceutical efforts. We note that therapeutically relevant proteins are known to phase separate, and these proteins have properties similar to puncta-positive FO groups presented in this work. Furthermore, recent reports show preferential partitioning of small molecules into therapeutically relevant condensates. These points can be found in the Discussion, as follows.

Discussion, Pg. 26, Line 14 to Pg. 27, Line 25:

“The discovery of condensate formation by almost 58% of the FOs we tested represents a potential therapeutic vulnerability. Most FOs are comprised of both folded domains and IDRs, which together mediate condensate formation and aberrant biological functions. While small molecules have been shown to interact with specific regions within IDRs and modulate function²⁹⁻³¹, IDR-targeted small molecule drugs have not yet reached the clinic. In contrast, folded domains within condensate-forming FOs are potentially accessible for therapeutic targeting. For example, BCR-ABL, which possesses the constitutively active ABL tyrosine kinase domain and is a driver of chronic myelogenous leukemia³², tested positive for

condensate formation in our studies, suggesting that clinically effective kinase inhibitors such as imatinib³² might function within condensates. Interestingly, BCR-ABL was previously shown to localize within stress granules; inhibition of kinase activity with imatinib released the FO from these granules³³. Many of the experimentally tested FOs (Expressed and Verification FOs) contain kinase domains (29 FOs, in total), with eight shown to form condensates. Therefore, it is relevant to consider the accessibility of small molecules to the interior of condensates when seeking to target puncta(+), kinase domain-containing FOs. This issue has been addressed experimentally, with some FDA-approved drugs shown to preferentially partition into certain biomolecular condensates reconstituted *in vitro*^{34,35}. A strategy for the future may be to develop small molecules to target puncta(+), kinase domain-containing FOs by optimizing both condensate partitioning and kinase inhibition. Many other puncta(+) FOs formed condensates in the nucleus and contain domains involved in regulation of gene expression (Fig. 7A), including many DNA binding domains (Supplementary Table 8). Many of these FOs also contain IDRs, creating chimeric transcription factors that drive aberrant gene expression¹. However, transcription factors have generally been considered undruggable³⁶, although some have expressed optimism about the potential of proteolysis-targeting chimeric (PROTAC) molecules to target them³⁷. An alternative strategy for puncta(+) FOs that are aberrant transcription factors is to target critical interaction partners that have targetable folded domains. An example is the interaction of NUP98 FOs with the Menin-MLL1 complex, which is a molecular dependency in FO-driven pediatric AML³⁸. The compound VTP50469 inhibits the Menin-MLL1 interaction, displacing these proteins from chromatin sites also occupied by NUP98 FOs, reversing the leukemogenic gene expression program³⁸. This compound acted similarly in MLL FO-driven leukemias, which also depend on interactions with Menin³⁹. Both the NUP98¹ and MLL FOs (reported herein) form nuclear condensates but the activity of VTP50469 has not been considered through this lens; perhaps consideration of condensate partitioning may guide the development of improved analogs of this compound in the future. In conclusion, because a significant proportion of cancer-driving FOs form condensates in either the nucleus or cytoplasm, or in both compartments, consideration of the influence of the condensate physicochemical environment on small molecule partitioning and interactions may promote drug development against them in the future.”

Minor comments

It would be clearer if S3 is cited together with S4 instead of separated. Adjust Figure 3, some parts are too small the not readable.

Author Comment: We thank the reviewer for these comments. The original Figure S3 reported on the FRAP and fusion data of a subset of our FOs, while Figure S4 addressed the Verification of FO-Puncta ML model performance and comparison with other phase separation prediction models. Therefore, respectfully, it does not seem appropriate to combine the two. However, the suggestion regarding Figure 3 was very helpful. Figure 3 has been divided into two new figures (Fig. 3 and Fig. 4) to allow for larger panels and improve clarity.

To keep the story linear and guide the reader it would easier if all the functional features are presented before the prediction and Machine learning part (i.e. Figure 5 presented before Figure 4) as indeed is depicted in the Summary schematic of the FO condensate landscape Fig 7 D.

Author Comment: We thank the reviewer for this suggestion, and we initially intended to make this change. However, we perform the conserved domain/functional analyses on both the Training and Verification FOs, which means that the results must be presented after discussion of the ML model. Therefore, while we appreciate the suggestion, we have decided to present

the conserved domain/functional analyses (Fig. 7) after presentation of the ML model (Figs. 5 and 6). We hope the reviewer appreciates the basis for this decision, which is rooted in wanting to enhance the rigor of the conserved domain/functional analyses through analysis of both the Training and Verification FOs.

Lines 24-25, complex sentence, to rephrase.

Author Comment: The reviewer did not clarify the page they are referring to, but we assume it was original Page 19 in the Discussion section and the following sentence:

“We present databases, imaging datasets, reagents, and tools to understand the relationships between the amino acid sequence-based physicochemical features of FOs and their cellular behavior and function (Fig. 7D).”

We have reworded this sentence, as follows, in the revised manuscript:

Pg. 22, Line 24 to Pg. 23, Line 1:

“We present databases, microscopy images, reagents, and computational tools to understand the relationships between the amino acid sequence-based physicochemical features of FOs and their cellular behavior, including condensate formation, sub-cellular localization and biological function (Fig. 9D).”

Reviewer #3 (Remarks to the Author):

In this manuscript, authors first generated a complete list of fusion oncoproteins (FOs) and examined the potential of puncta production with 166 of them following overexpression in Hela cells. Features of those puncta(+) and puncta(-) subgroups were then studied, which were used for building a machine learning model to predict condensation by a broad list of FOs. Overall, the manuscript is generally well written and figures are properly presented. Predicting condensation behaviors of thousands of FOs is also a rather ambitious and exciting task, since the detailed investigation of condensation/LLPS property of a single FO often requires a full set of assays. However, I also feel that this manuscript comes in short at key places.

Author Comment: We thank the reviewer for their enthusiasm for our manuscript, “Predicting condensation behaviors of thousands of FOs is also a rather ambitious and exciting task ...”. We also appreciate their constructive comments and suggestions, which have contributed to improving the manuscript.

Fig 1 used the available databases, part of which I believe was published.

Author Comment: While some of the FOs included in our FOdb database have been included in previous published works from our colleagues, Jinghui Zhang and Charles Mullighan (e.g., refs. 40-46), those included in FOdb from St. Jude Children’s Research Hospital sources were identified de novo using the CICERO gene fusion caller⁴³. Other FOs included in FOdb were identified de novo in TCGA using the sequence analysis approaches discussed under Methods. Therefore, while there is likely extensive overlap between FOs in our FOdb dataset and others published previously, the FOdb dataset was generated de novo using recently developed sequence analysis methodologies.

The main concern of Fig 2-3 is that, authors only used very limited assays for studying FOs (that is, overexpression and condensate production in Hela cells). Often, assaying puncta formation in this cell line isn't sufficient to demonstrate LLPS.

Author Comment: We appreciate the concern of the reviewer and want to clarify, respectfully, that we do not explicitly state that the condensates we observed for 96 FOs form through the process of phase separation. We do note (on page 12, line 22-23) that a few we tested exhibit liquid-like features (e.g., rapid fluorescence recovery after photobleaching and/or observation of fusion events) suggestive of formation through phase separation. However, nowhere in the manuscript do we state unequivocally that the condensates we observe form through phase separation. We hope the reviewers is accepting of our usage of the term “condensates” and the associated cautious language.

We refer the reviewer to our comment at the beginning of this document on this point, which is repeated below.

Author Comment: We wish to clarify our usage of the term “condensates”, which we use to describe the dense, round structures that we observed in cells through expression of many GFP-tagged FOs. We regret any confusion we caused by using this term. It was an oversight on our part to not explicitly define our usage of the term. By using this term, we do not intend to imply formation of these structures through the formal process of phase separation. While the appearance of dense, round structures in cells for many FOs is highly suggestive of formation through phase separation, our current data for most of the FOs studied do not support this claim. To strengthen the suggestion that phase separation is operative for some FOs, we performed FRAP experiments, which showed that most of the GFP-FO molecules were mobile within condensates. Also, in several cases, we observed what appear to be condensate fusion events, indicating that the condensates exhibit surface tension. We feel that these data, for a small subset of the tested FOs, gives us liberty to suggest, in general, that the condensates we observe form through phase separation. We note that the physical features and behavior of the FO condensates we observe are similar to those observed by us and others for a variety other FOs, for which rigorous evidence of formation through phase separation was provided (reviewed in Shirnekhi, *et al.*, *Annual Reviews of Cancer Biology*, 2023). Further, while we wish we had the resources to do so, it is simply not feasible to perform the complement of experiments needed to establish phase separation as the mechanism of formation for over 100 FOs with our current manpower and resources. To address this issue, we have clarified our usage of the term “condensates” in several place in the text, as follows.

Introduction, Pg. 5, Lines 14 – 18

“We note that while the observation of FO-induced condensates is suggestive of formation through phase separation, other tests beyond the scope of the current investigations would be needed to establish this assembly mechanism. Therefore, our use of the term “condensates” is agnostic as to formation mechanism.”

and Results, Pg. 8, Lines 10 - 12:

“We reiterate, however, that our observation of condensate formation by a large proportion of the Expressed FOs is insufficient to establish whether formation occurs through phase separation.”

Many described features associated with puncta+ and puncta- FOs are partly same or similar; authors also mentioned several limitations of the useful approaches such as the over-expression levels, cellular contexts (not mimicking those FO-related cancer contexts), availability of cofactors, etc. Evidence strikes to be thin and seems that in vitro assays shall be used as a complementary approach.

Author Comment: We thank the reviewer for pointing out aspects of our manuscript that could be improved through revision. Regarding the latter comment about evidence pertaining to FO phase separation being thin, we want to refer the reviewer back to our response to their previous comment about puncta formation being insufficient to demonstrate LLPS. The point regarding similarity of features for some puncta(+) and puncta(-) FOs was addressed above in our response to Reviewer 2 and is repeated below:

Author Comment: We thank the reviewer for this comment. While we cannot rule out that interaction partners influence the condensation behavior of the FOs we tested, past studies from us and others have shown through in vitro and cellular studies that some FOs have an intrinsic ability to form condensates (reviewed in Shirnekhi, *et al.*, Shirnekhi, Chandra, Kriwacki, *Ann. Rev. Cancer Biology*, 2023). While by qualitative inspection it is difficult to identify distinguishable feature patterns between the puncta(+) Group 4 and puncta(-) Group 2' FOs, our FO-Puncta Machine Learning (ML) model is able to do so. The physicochemical feature heatmaps shown in Fig. 4 and Supplementary Fig. 2 display Z-scores for only the 12 most discriminatory features, but the ML model used 25 features in making predictions. Examination of the weight of the contribution of each feature to the ML predictions for the two groups, as reported by SHAP contribution values, revealed feature differences between the positive (Group 4) and negative (Group 2') FOs and results in unbiased, separate clustering of the two groups. We have included these data as a new supplemental figure, Supplementary Fig. 11 and modified the text to include this information. These additions illustrate the basis for the ML model's ability to discriminate between these two groups. We have modified the text to address these points, as follows.

Discussion, Pg. 24, Line 14 to Pg. 25, Line 4:

“In contrast to FOs in Groups 1-3, those in Group 4 displayed indistinct feature enrichments, which further were indistinguishable from those of puncta(-) FO Group 2'. These findings may indicate that the 12 most discriminatory physicochemical features included in our analyses do not capture the properties associated with their condensate formation behavior. However, expanding our analyses to 25 features and examination of how they contribute to condensation formation predictions by the FO-Puncta ML model revealed differences between the Group 4 puncta(+) and Group 2' puncta(-) FOs (Supplementary Fig. 11A). Unbiased 2D hierarchical clustering on the basis of SHAP contribution values revealed that differences in the values of up to six features (left feature columns, Supplementary Fig. 11A) naturally led to segregation into the puncta(+) and puncta(-) groups (Group 4 and Group 2', respectively). While it is difficult to discern differences between these two groups on the basis of physicochemical feature values (Supplementary Fig. 11B), the FO-Puncta ML model was able to identify such differences and correctly recapitulate FO condensation behavior. Interestingly, the SHAP contribution values that most highly discriminate between Groups 4 and 2' report on charge-related

physicochemical features (Supplementary Fig. 11A). These results illustrate the utility of our FO-Puncta ML model and its use of physicochemical features in analysis of the condensation behavior of FOs.”

Discussion, Pg. 29, Lines 6 - 17:

“Additionally, while we identified distinct patterns of physicochemical features associated with puncta(+) behavior by 96 of the 166 Expressed FOs, we have not explored how these patterns are associated with the conformational properties of these proteins or their propensities for multivalent interactions that underlie condensate formation. Nor have we probed interaction partners associated with puncta(+) FOs and their potential roles in condensate formation. FOs previously shown to form condensates (reviewed in ref. 52) have been shown to interact with multiple, additional proteins, but have also been shown to have an intrinsic propensity to form condensates through phase separation. Beyond interaction partners, we hypothesize that puncta(+) FOs displaying different patterns of physicochemical features and amino acid enrichments (e.g., FOs in Groups 1-3; Fig. 4A, C, D) will exhibit different conformational properties and different types of intra- and inter-polypeptide chain interactions that promote their ability to form condensates. However, testing this hypothesis must await future investigation.”

It remains unclear how reliable the presented results would be in determining LLPs behavior or true biological scenarios; how useful would the machine learning predictions be to the field? As the title indicated, the work focuses on defining the landscape of condensates by FOs. To me, a far more important question is, how and whether or not the condensation by FOs would affect biology, which certainly isn't the focus. Due to these main weaknesses, I would recommend this Resource paper to a more specialized journal.

Author Comment: We thank the reviewer for raising these questions and comments, which are similar to points raised by Reviewer 1, above. We repeat our response to Reviewer 1 below.

Author Comment: We appreciate the reviewer's comments regarding the connection between condensate formation by FOs and oncogenic mechanisms, which were also offered by another reviewer. It is very important to understand the functional mechanism(s) underlying the oncogenic functions of FOs. Importantly, we and others previously established rigorous relationships between condensate formation through phase separation by a select group of FOs and oncogenic mechanism (reviewed in Shirnekhi, Chandra, Kriwacki, *Ann. Rev. Cancer Biology*, 2023). For each FO, the work required one or more years of effort by teams of investigators. We wish we could perform these types of studies for >100 puncta(+) FOs, but doing this is beyond what is feasible given our current resources. We hope the reviewers understand this limitation.

However, while large-scale experimental testing is not feasible, we have extensively mined the literature to strengthen links between condensate formation and oncogenic function. While such data were included in the original manuscript, we performed additional literature searches and now report data related to oncogenic phenotypes for 36 puncta-positive FOs (and 6 puncta(-) FOs) in Supplementary Table 4. These data include evidence of cell transformation by an FO, data showing that an FO drives an oncogenic transcriptional program, and/or in vivo data showing that an FO promotes oncogenesis. While these data do not establish that condensate formation by these 36 FOs is causally linked with oncogenesis, they do support the idea that many FOs that promote cancer phenotypes form condensates in cells. We argue that these additional literature-derived data strengthen the biological and cancer relevance of our studies.

Furthermore, our analyses of conserved domains (Fig. 7) indicate possible oncogenic mechanisms of some puncta(+) FOs, although further experimentation beyond the scope of the current manuscript would be needed in the future to clarify these indications. We have modified the text, as follows, to address these points.

Results, Pg. 7, Lines 14 - 17:

“The 166 FOs spanned both adult and pediatric cancers and included 77 of the 110 FOs with a patient count of ≥ 3 and 36 FOs previously demonstrated to drive oncogenic phenotypes in cancer-relevant cell types (Supplementary Table 4).”

Discussion, Pg. 25, Line 5 to Pg. 26, Line 13:

“Our analyses of conserved domains within the experimentally tested FOs provide insight into their possible oncogenic mechanisms. Puncta(+) FOs localized in the nucleus (Fig. 7A, C) are highly enriched in functional terms related to regulation of gene expression. Further, these terms are not enriched amongst puncta(-) FOs, very few of which are localized in the nucleus (Fig. 7A-C). Several condensate-forming FOs studied previously have similar functional domains and are known to promote oncogenesis by driving aberrant gene expression¹. We speculate that this functional mechanism is common to many of the nuclear, puncta(+) FOs we identified in our studies, although definitive testing awaits future investigations.

Regulation of expression of certain genes involves formation of so-called super-enhancers, which involve the compaction of distal DNA regulatory elements and transcriptional regulatory proteins, within condensates that additionally recruit RNA polymerase II². The emergent properties of phase separated condensates, including compaction of biopolymers such as DNA and concentration of multiple protein factors within them, are well-matched with mechanistic aspects of transcriptional regulation and may underlie why many nuclear FOs harboring domains associated with gene regulation form condensates. In contrast, the majority of FOs with functional terms related to regulation of cell signaling did not form condensates in our studies (Fig. 7). However, there are notable examples of condensate-forming FOs that aberrantly regulate cell signaling (reviewed in ref. ¹), and we did observe cytoplasmic condensate formation by some FOs enriched in cell signaling-related terms (Fig. 7). However, the enrichment of these functional terms was greatest for FOs that exhibited diffuse localization in the cytoplasm or both the cytoplasm and nucleus (Fig. 7E, F). This apparent functional dichotomy for puncta(+) and puncta(-) FOs was recapitulated through analysis of 2,999 additional FOs using physicochemical features and the FO-Puncta ML model (Fig. 9C), with FOs predicted to form condensates most frequently matching the features of Groups 1-3 FOs, which we propose encodes aberrant regulation of gene expression function in the nucleus. In contrast, FOs predicted to not form condensates most frequently matched the features of Groups 1'-3' FOs, which we align with aberrant regulation of cell signaling function in the cytoplasm. These observations, overall, reinforce the long-held idea that FOs fall into one of two general functional classes, those that drive oncogenesis by regulating transcription and others that promote oncogenesis by regulating cell signaling. However, the novel insight from our studies is the alignment of condensate formation with the former class and not the latter. These results suggest that the emergent properties of condensates are not generally required for regulation of cell signaling by FOs, but it is also possible that structures with emergent properties do in fact form and are too small to be detected using the confocal fluorescence microscopy methods employed in our studies.”

Discussion, Pg. 28, Line 22 to Pg. 29, Line 5:

“Another limitation to this work is the infeasibility of directly assessing the functional consequences of condensate formation by the 96 puncta(+) FOs. Importantly, links between condensate formation and function have been established for a number of FOs, some of which were tested in our studies³⁻¹³. Also, 34 of our puncta(+) FOs have previously been shown to promote oncogenic cellular phenotypes. Further, we identified conserved domains with definitive functional annotations in the amino acid sequences of a large portion of both puncta(+) and puncta(-) FOs (166 Expressed and 29 Verification FOs; Fig. 7), enabling us to propose their biological functions. These hypotheses are a rich resource for us and others to pursue mechanistic studies into relationships between condensate formation, or not, and oncogenesis driven by FOs in the future.”

References cited in author responses:

- 1 Shirnekhi, H. K. C., B.; Kriwacki, R. The Role of Phase Separated Condensates in Fusion Oncoprotein Driven Cancers. *Annual Reviews of Cancer Biology* **in press** (2023).
- 2 Sabari, B. R. *et al.* Coactivator condensation at super-enhancers links phase separation and gene control. *Science* **361**, doi:10.1126/science.aar3958 (2018).
- 3 Boulay, G. *et al.* Cancer-Specific Retargeting of BAF Complexes by a Prion-like Domain. *Cell* **171**, 163-178 e119, doi:10.1016/j.cell.2017.07.036 (2017).
- 4 Davis, R. B., Kaur, T., Moosa, M. M. & Banerjee, P. R. FUS oncofusion protein condensates recruit mSWI/SNF chromatin remodeler via heterotypic interactions between prion-like domains. *Protein Sci* **30**, 1454-1466, doi:10.1002/pro.4127 (2021).
- 5 Owen, I. *et al.* The oncogenic transcription factor FUS-CHOP can undergo nuclear liquid-liquid phase separation. *J Cell Sci* **134**, doi:10.1242/jcs.258578 (2021).
- 6 Ahn, J. H. *et al.* Phase separation drives aberrant chromatin looping and cancer development. *Nature* **595**, 591-595, doi:10.1038/s41586-021-03662-5 (2021).
- 7 Chandra, B. *et al.* Phase Separation mediates NUP98 Fusion Oncoprotein Leukemic Transformation. *Cancer Discov*, 1152–1169, doi:10.1158/2159-8290.CD-21-0674 (2021).
- 8 Jevtic, Z. *et al.* SMARCA5 interacts with NUP98-NSD1 oncofusion protein and sustains hematopoietic cells transformation. *J Exp Clin Cancer Res* **41**, 34, doi:10.1186/s13046-022-02248-x (2022).
- 9 Terlecki-Zaniewicz, S. *et al.* Biomolecular condensation of NUP98 fusion proteins drives leukemogenic gene expression. *Nat Struct Mol Biol* **28**, 190-201, doi:10.1038/s41594-020-00550-w (2021).
- 10 Qin, Z. *et al.* Phase separation of EML4-ALK in firing downstream signaling and promoting lung tumorigenesis. *Cell Discov* **7**, 33, doi:10.1038/s41421-021-00270-5 (2021).
- 11 Sampson, J., Richards, M. W., Choi, J., Fry, A. M. & Bayliss, R. Phase-separated foci of EML4-ALK facilitate signalling and depend upon an active kinase conformation. *EMBO Rep* **22**, e53693, doi:10.15252/embr.202153693 (2021).
- 12 Tulpule, A. *et al.* Kinase-mediated RAS signaling via membraneless cytoplasmic protein granules. *Cell* **184**, 2649-2664 e2618, doi:10.1016/j.cell.2021.03.031 (2021).
- 13 Shao, X. *et al.* Deneddylation of PML/RARalpha reconstructs functional PML nuclear bodies via orchestrating phase separation to eradicate APL. *Cell Death Differ*, 1654-1668, doi:10.1038/s41418-022-00955-8 (2022).
- 14 Mitrea, D. M. & Kriwacki, R. W. Phase separation in biology; functional organization of a higher order. *Cell Commun Signal* **14**, 1, doi:10.1186/s12964-015-0125-7 (2016).

- 15 Somjee, R., Mitrea, D. M. & Kriwacki, R. W. Exploring Relationships between the Density of Charged Tracts within Disordered Regions and Phase Separation. *Pac Symp Biocomput* **25**, 207218 (2020).
- 16 Conicella, A. E., Zerze, G. H., Mittal, J. & Fawzi, N. L. ALS Mutations Disrupt Phase Separation Mediated by alpha-Helical Structure in the TDP-43 Low-Complexity C-Terminal Domain. *Structure* **24**, 1537-1549, doi:10.1016/j.str.2016.07.007 (2016).
- 17 Schmidt, H. B. & Rohatgi, R. In Vivo Formation of Vacuolated Multi-phase Compartments Lacking Membranes. *Cell Rep* **16**, 1228-1236, doi:10.1016/j.celrep.2016.06.088 (2016).
- 18 Ahmed, N. S. *et al.* Fusion protein EWS-FLI1 is incorporated into a protein granule in cells. *Rna* **27**, 920-932, doi:10.1261/rna.078827. 121 (2021).
- 19 Chong, S. *et al.* Imaging dynamic and selective low-complexity domain interactions that control gene transcription. *Science* **361**, eaar2555, doi:doi:10.1126/science.aar2555 (2018).
- 20 Lyons, H. *et al.* Functional partitioning of transcriptional regulators by patterned charge blocks. *Cell* **186**, 327-345 e328, doi:10.1016/j.cell.2022.12.013 (2023).
- 21 Nott, Timothy J. *et al.* Phase Transition of a Disordered Nuage Protein Generates Environmentally Responsive Membraneless Organelles. *Molecular Cell* **57**, 936-947, doi:<https://doi.org/10.1016/j.molcel.2015.01.013> (2015).
- 22 Zhang, Y. *et al.* Interface resistance of biomolecular condensates. *bioRxiv*, doi:10.1101/2022.03.16.484641 (2022).
- 23 Strom, A. R. *et al.* Phase separation drives heterochromatin domain formation. *Nature* **547**, 241245, doi:10.1038/nature22989 (2017).
- 24 Taylor, N. O., Wei, M. T., Stone, H. A. & Brangwynne, C. P. Quantifying Dynamics in Phase-Separated Condensates Using Fluorescence Recovery after Photobleaching. *Biophys J* **117**, 12851300, doi:10.1016/j.bpj.2019.08.030 (2019).
- 25 Farag, M. *et al.* Condensates formed by prion-like low-complexity domains have small-world network structures and interfaces defined by expanded conformations. *Nature communications* **13**, 7722, doi:10.1038/s41467-022-35370-7 (2022).
- 26 Boija, A. *et al.* Transcription Factors Activate Genes through the Phase-Separation Capacity of Their Activation Domains. *Cell*, doi:10.1016/j.cell.2018.10.042 (2018).
- 27 Chong, S. *et al.* Imaging dynamic and selective low-complexity domain interactions that control gene transcription. *Science* **361**, doi:10.1126/science.aar2555 (2018).
- 28 Lyons, H. *et al.* Functional partitioning of transcriptional regulators by patterned charge blocks. *Cell* **186**, 327-345.e328, doi: 10.1016/j.cell.2022.12.013 (2023).
- 29 Iconaru, L. I. *et al.* Discovery of Small Molecules that Inhibit the Disordered Protein, p27(Kip1). *Scientific reports* **5**, 15686, doi:10.1038/srep15686 (2015).
- 30 Ban, D., Iconaru, L. I., Ramanathan, A., Zuo, J. & Kriwacki, R. W. A Small Molecule Causes a Population Shift in the Conformational Landscape of an Intrinsically Disordered Protein. *J Am Chem Soc* **139**, 13692-13700, doi:10.1021/jacs.7b01380 (2017).
- 31 Heller, G. T. *et al.* Small-molecule sequestration of amyloid- β as a drug discovery strategy for Alzheimer's disease. *Sci Adv* **6**, doi:10.1126/sciadv.abb5924 (2020).
- 32 Ren, R. Mechanisms of BCR-ABL in the pathogenesis of chronic myelogenous leukaemia. *Nat Rev Cancer* **5**, 172-183, doi:10.1038/nrc1567 (2005).
- 33 Kashiwagi, S. *et al.* Localization of BCR-ABL to Stress Granules Contributes to Its Oncogenic Function. *Cell Struct Funct* **44**, 195-204, doi:10.1247/csf. 19033 (2019).
- 34 Thody, S. A. *et al.* Small Molecule Properties Define Partitioning into Biomolecular Condensates. *bioRxiv*, 2022.2012.2019.521099, doi:10.1101/2022.12.19.521099 (2022).

- 35 Klein, I. A. *et al.* Partitioning of cancer therapeutics in nuclear condensates. *Science* **368**, 13861392, doi:10.1126/science.aaz4427 (2020).
- 36 Dang, C. V., Reddy, E. P., Shokat, K. M. & Soucek, L. Drugging the 'undruggable' cancer targets. *Nat Rev Cancer* **17**, 502-508, doi:10.1038/nrc.2017.36 (2017).
- 37 Békés, M., Langley, D. R. & Crews, C. M. PROTAC targeted protein degraders: the past is prologue. *Nat Rev Drug Discov* **21**, 181-200, doi:10.1038/s41573-021-00371-6 (2022).
- 38 Heikamp, E. B. *et al.* The menin-MLL1 interaction is a molecular dependency in NUP98-rearranged AML. *Blood* **139**, 894-906, doi:10.1182/blood.2021012806 (2022).
- 39 Krivtsov, A. V. *et al.* A Menin-MLL Inhibitor Induces Specific Chromatin Changes and Eradicates Disease in Models of MLL-Rearranged Leukemia. *Cancer Cell* **36**, 660-673.e611, doi:10.1016/j.ccell.2019.11.001 (2019).
- 40 Roberts, K. G. *et al.* Targetable kinase-activating lesions in Ph-like acute lymphoblastic leukemia. *N Engl J Med* **371**, 1005-1015, doi:10.1056/NEJMoa1403088 (2014).
- 41 Liu, Y. *et al.* Etiology of oncogenic fusions in 5,190 childhood cancers and its clinical and therapeutic implication. *Nature communications* **14**, 1739, doi:10.1038/s41467-023-37438-4 (2023).
- 42 Zhao, X. *et al.* Epigenetic activation of the FLT3 gene by ZNF384 fusion confers a therapeutic susceptibility in acute lymphoblastic leukemia. *Nature communications* **13**, 5401, doi:10.1038/s41467-022-33143-w (2022).
- 43 Tian, L. *et al.* CICERO: a versatile method for detecting complex and diverse driver fusions using cancer RNA sequencing data. *Genome Biol* **21**, 126, doi:10.1186/s13059-020-02043-x (2020).
- 44 Ma, X. *et al.* Pan-cancer genome and transcriptome analyses of 1,699 paediatric leukaemias and solid tumours. *Nature* **555**, 371-376, doi:10.1038/nature25795 (2018).
- 45 Gao, Q. *et al.* The genomic landscape of acute lymphoblastic leukemia with intrachromosomal amplification of chromosome 21. *Blood*, doi:10.1182/blood.2022019094 (2023).
- 46 Brady, S. W. *et al.* The genomic landscape of pediatric acute lymphoblastic leukemia. *Nat Genet* **54**, 1376-1389, doi:10.1038/s41588-022-01159-z (2022).

REVIEWERS' COMMENTS

Reviewer #2 (Remarks to the Author):

The authors have replied to all previous comments and I am quite satisfied. It is a very nice piece of work, congrats!

Reviewer #3 (Remarks to the Author):

Authors have adequately addressed my prior comments and therefore i suggest the publication.